# Sinkhorn Treatment Effects: A Causal Optimal Transport Measure

**Medha Agarwal** [1]    **Alex Luedtke** [2]

## Abstract

We introduce the Sinkhorn treatment effect, an entropic optimal transport measure of divergence between counterfactual outcome distributions. Unlike classical quantities such as the average treatment effect, it captures differences across entire distributions. We show that this estimand can be written as a smooth transformation of counterfactual mean embeddings with an appropriate kernel. This characterization allows us to establish first-order pathwise differentiability in general, and second-order pathwise differentiability under the null hypothesis of equal counterfactual distributions. Leveraging this smoothness, we construct debiased estimators and asymptotically valid tests for distributional treatment effects at a fixed entropic regularization parameter. Because the power of the test depends on this unknown parameter, we propose an aggregated test that combines evidence across a grid of regularization choices. Experiments on simulated and image data demonstrate the practical advantages of our estimator and testing procedure.

## 1. Introduction

Estimating causal treatment effects is a central goal of causal inference (Rubin, 1974; 2005; Rosenbaum & Rubin, 1983; Chernozhukov et al., 2017). In the canonical binary treatment setting, the scientific goal is the discrepancy between the counterfactual outcome distributions under treatment and control, denoted by $Y_1 \sim P_1$ (treatment) and $Y_0 \sim P_0$ (control). However, $P_1$ and $P_0$ are never observed via independent samples. Instead, we observe independent copies of $(X, A, Y) \sim P$, where treatment $A$ and observed outcome $Y$ may depend on confounders $X$. Under standard causal identifiability conditions (see Sec. 2), $P_1$ and $P_0$ can

[1] Department of Statistics, University of Washington, Seattle, WA, USA [2] Department of Health Care Policy, Harvard Medical School, Boston, MA, USA. Correspondence to: Medha Agarwal <medhaaga@uw.edu>.

*Proceedings of the 43rd International Conference on Machine Learning*, Seoul, South Korea. PMLR 306, 2026. Copyright 2026 by the author(s).

be learned from $P$. Most empirical work summarizes treatment effects through the average treatment effect (ATE), $\mathbb{E}_{P_1}[Y_1] - \mathbb{E}_{P_0}[Y_0]$ (Imbens, 2004). While convenient for computation and inference, mean-based summaries can mask important effects that occur away from the center of the distribution. Empirical studies have shown that interventions may induce large distributional shifts despite a negligible ATE (Bitler et al., 2006).

These limitations motivate inference on the entire counterfactual outcome distributions, commonly referred to as distributional treatment effects (DTEs) (Abadie, 2002). Classical approaches target specific functionals of $P_1$ and $P_0$, including quantiles (Firpo, 2007), cumulative distribution functions (Chernozhukov et al., 2013)—which imply results for the 1-Wasserstein distance in univariate outcome settings (Lin et al., 2023; Balakrishnan et al., 2025)—and probability density functions (Robins & Rotnitzky, 2001; Kennedy et al., 2023). More recently, kernel-based methods estimate counterfactual distributions via kernel mean embeddings (KME) into characteristic reproducing kernel Hilbert spaces (RKHS) (Muandet et al., 2017) and quantify DTE using maximum mean discrepancy (MMD) (Fawkes et al., 2024; Martinez Taboada et al., 2023; Luedtke & Chung, 2024). Although MMD metrizes weak convergence, it induces a flat geometry on the space of probability measures and may fail to capture meaningful geometric discrepancies between counterfactual distributions, especially when their supports weakly overlap or separate. Related approaches based on $f$-divergence further require mutual absolute continuity, limiting applicability (Kennedy et al., 2023). Unlike $f$-divergences, MMD is finite for disjoint distributions; however, MMD saturates quickly as this disjointness increases, a phenomenon visualized in Fig. 1.

To overcome these limitations, we propose measuring DTE via the Sinkhorn divergence (Feydy et al., 2019; Ramdas et al., 2017), a centered entropy-regularized optimal transport (EOT) cost, that respects the intrinsic geometry of the outcome space. Under identifiability, we define the *Sinkhorn treatment effect* (STE) as

$$\mathcal{S}(P) = \mathrm{OT}_\varepsilon\left(P_1, P_0\right) - \frac{1}{2}\left[\mathrm{OT}_\varepsilon\left(P_1, P_1\right) + \mathrm{OT}_\varepsilon\left(P_0, P_0\right)\right]. \tag{1}$$

For measures $\mu$ and $\nu$, the EOT cost with quadratic cost

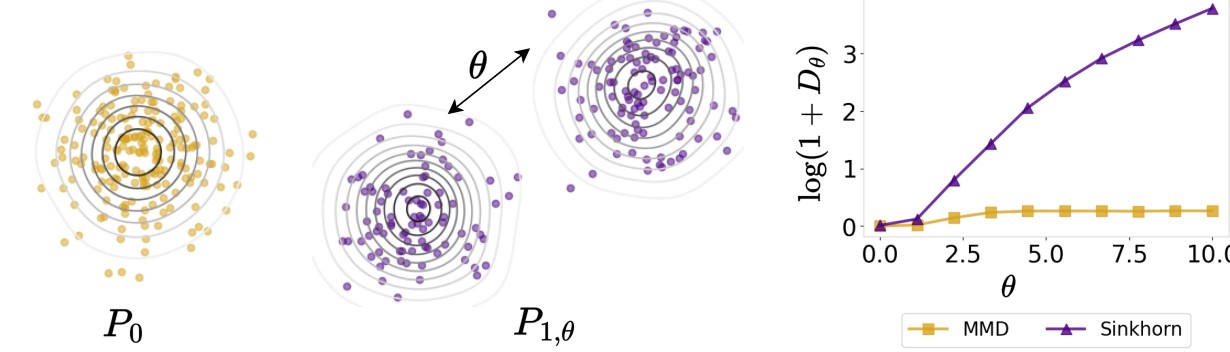

*Figure 1.* MMD vs. Sinkhorn divergence across increasing separation $\theta$ between counterfactual outcome distribution under control $P_0 = \mathcal{N}(\mathbf{0}_2, I_2)$ and under treatment $P_{1,\theta} = \frac{1}{2}\mathcal{N}(-\theta\mathbf{1}_2, I_2) + \frac{1}{2}\mathcal{N}(\theta\mathbf{1}_2, I_2)$. The average treatment effect is zero for all $\theta > 0$. Here $D_\theta$ denotes either MMD or Sinkhorn divergence between $P_0$ and $P_{1,\theta}$. As $\theta$ increases, the distributions diverge and **MMD saturates, failing to distinguish between 'far' and 'very far' distributions**, whereas the Sinkhorn divergence continues to grow.

function $c(x, y) = \frac{1}{2}\|x - y\|^2$ and $\varepsilon > 0$ regularization is

$$\mathrm{OT}_\varepsilon(\mu, \nu) = \inf_{\pi \in \Pi(\mu, \nu)} \int c\, d\pi + \varepsilon \mathrm{KL}(\pi \mid \mu \otimes \nu), \quad (2)$$

where $\Pi(\mu, \nu)$ denotes the set of couplings of $(\mu, \nu)$, and $\mathrm{KL}(\cdot \mid \cdot)$ is the Kullback–Leibler divergence. Here $c$ reflects the cost of moving mass between two outcome values, thereby encoding domain-specific notions of discrepancy between outcomes, while $\varepsilon$ controls the degree of entropic smoothing. As $\varepsilon \to 0$, STE approaches the corresponding unregularized OT cost, and as $\varepsilon \to \infty$, it approaches the MMD distance corresponding to the Gibbs kernel $e^{-c/\varepsilon}$ (Feydy et al., 2019).

We show that STE admits a pathwise differentiable representation, enabling construction of a doubly robust, asymptotically normal one-step estimator (Pfanzagl, 1982). While this first-order estimator is degenerate under the null of equal counterfactual outcome distributions, mirroring known degeneracy phenomena for debiased MMD-based tests (Muandet et al., 2017), we further develop a second-order influence function-based correction, yielding a valid and powerful test with an $n$-rate limiting distribution. With our asymptotic result, we construct a test with provable type I error control, for each fixed regularization parameter $\varepsilon > 0$. Since $\varepsilon$ is unknown in practice and an unfavorable choice may result in low power, we further propose STEAgg, a multiple testing procedure that combines evidence across a grid of $\varepsilon$ values. This approach is inspired by aggregated kernel tests such as Schrab et al. (2023), which combine MMD statistics across bandwidths while maintaining non-asymptotic type I error guaranties. We prove that STEAgg-based tests achieve nominal asymptotic type I error control.

**Our Contributions.**

1. We introduce Sinkhorn treatment effects. Unlike prior causal OT approaches, ours allows for multivariate outcomes.
2. We characterize the smoothness of the STE functional, showing it is first-order differentiable and second-order differentiable under the null of no DTE.
3. We use this differentiability to construct efficient first- and second-order bias-corrected estimators of STE.
4. We propose STE- and STEAgg-based hypothesis tests for the causal null, at a fixed $\varepsilon > 0$ and aggregated over a finite grid of $\varepsilon$ values, respectively.
5. We demonstrate the practical advantages of our approach on simulated and high-dimensional image data.

## 2. Background

**Notation.** For a Polish space $\mathcal{W}$, let $\mathcal{P}(\mathcal{W})$ be the space of Borel probability measures and $\mathcal{M}(\mathcal{W})$ be the Banach space of finite signed Radon measures, equipped with total variation norm. We further define the subspace of balanced measures $\mathcal{M}_0(\mathcal{W}) := \{\mu \in \mathcal{M}(\mathcal{W}) : \mu(\mathcal{W}) = 0\}$ and, for $\mathcal{W}$ compact, the Banach space $\mathcal{C}(\mathcal{W})$ of continuous functions on $\mathcal{W}$, endowed with the uniform norm. For any $\mu \in \mathcal{P}(\mathcal{W})$, let $\mathrm{supp}(\mu) \subset \mathcal{W}$ denote the support of $\mu$. For every multi-index $\alpha = (\alpha_1, \ldots, \alpha_d) \in \mathbb{N}_0^d$ with $|\alpha| = \sum_{i=1}^d \alpha_i$, define the differential operator $D^\alpha = \frac{\partial^{|\alpha|}}{\partial x_1^{\alpha_1} \ldots \partial x_d^{\alpha_d}}$ with $D^0 f = f$. For $s \in \mathbb{N}_0$, when $\mathcal{W}$ has non-empty interior $\Omega := \mathrm{int}(\mathcal{W})$ and $\overline{\Omega} = \mathcal{W}$, let $\mathcal{C}^s(\mathcal{W})$ denote the set of functions $f \in \mathcal{C}(\mathcal{W})$ that have continuous derivatives of all orders $\le s$ on $\Omega$ and the derivatives have continuous extensions to $\mathcal{W}$. When $\mathcal{W} \subset \mathbb{R}^d$ is a compact set with non-

empty interior $\Omega$ a bounded Lipschiz domain (for example when $\mathcal{W}$ is a closed ball), we define for $s \in \mathbb{N}_0$ the Hilbert case of Sobolev spaces on $\mathcal{W}$:

$$W^s(\mathcal{W}) := W^s(\Omega) := \left\{ f \in L^2(\Omega) : D^\alpha f \in L^2(\Omega) \right\}.$$

The space $W^s(\mathcal{W})$ is a Hilbert space with inner product $\langle f, g \rangle_{W^s(\mathcal{W})} := \sum_{|\alpha| \le s} \langle D^\alpha f, D^\alpha f \rangle_{L^2(\mathcal{W})}$. For any measurable $f : \mathcal{W} \to \mathbb{R}$ and $P \in \mathcal{P}(\mathcal{W})$, we use the shorthand notation $Pf := \int f \, dP$. For any bounded linear operator $\mathcal{A} : \mathcal{F} \to \mathcal{G}$, with $\mathcal{F}$ and $\mathcal{G}$ Banach spaces, we denote the operator norm by $\|\mathcal{A}\|_{\mathcal{F} \to \mathcal{G}}$. We use $\odot$ to denote the elementwise product of functions or tensors. The notation $\otimes$ denotes the tensor product (of functions, tensors, or linear operators) or the product measure.

**Nonparametric Statistical Model.** We consider a model $\mathcal{P}$ of distributions of $Z = (X, A, Y)$, where $X \in \mathcal{X} \subset \mathbb{R}^p$ are covariates, $A \in \{0, 1\}$ is a binary treatment indicator, and $Y \in \mathcal{Y} \subset \mathbb{R}^d$ is the observed outcome. Let $\mathcal{Z} := \mathcal{X} \times \{0, 1\} \times \mathcal{Y}$ be endowed with the product topology. The model $\mathcal{P} \subset \mathcal{P}(\mathcal{Z})$ is unrestricted, save for three conditions:

1. $\mathcal{X}$ is Polish and $\mathcal{Y}$ is a bounded closed ball.
2. The model is dominated: there exists $\sigma$-finite $\lambda$ such that $p := \frac{dP}{d\lambda}$ is well defined for all $P \in \mathcal{P}$.
3. Strong positivity holds: $e_P(a \mid x) := P(A = a \mid X = x)$ satisfies $\inf_{P \in \mathcal{P}} \operatorname{ess\,inf}_{(a,x)} e_P(a|x) > 0$.

The first condition ensures $\mathcal{Z}$ is Polish, making regular conditional probability distributions well-defined. The second simplifies the notion of pathwise differentiability discussed in the sequel. The third is common in causal inference settings, and will be used in the causal identifiability result to follow.

**Causal Setup.** Let $Y_1$ and $Y_0$ denote the potential outcomes under treatment and control, respectively. Our objective is to assess the Sinkhorn divergence between the marginal distributions of $Y_1$ and $Y_0$. Consistent with other DTE works, we call each such distribution a *counterfactual distribution* (Martinez Taboada et al., 2023; Kennedy et al., 2023), with the understanding that it would be called an interventional distribution in the structural causal model framework (Bareinboim et al., 2022).

Under standard causal conditions (Hernán & Robins, 2024), the distribution of $Y_a$, $a \in \{0, 1\}$, is identifiable through an observed data distribution $P \in \mathcal{P}$. Concretely, this identifiability result involves $P_a \in \mathcal{P}(\mathcal{Y})$ defined so that, for all measurable $\mathcal{Y}' \subset \mathcal{Y}$:

$$P_a(\mathcal{Y}') = \int P\{Y \in \mathcal{Y}' \mid A = a, X = x\} \, dP_X(x), \quad (3)$$

where $P_X$ is the marginal of $X$ under $P$. Formally, (3) holds if $Z$ has an associated potential outcome $Y_a$ and the

following hold: no unmeasured confounders ($Y_a \perp\!\!\!\perp A \mid X$), consistency ($A = a$ implies $Y = Y_a$), and positivity ($e_P(a \mid X) > 0$ $P_X$-a.s.).

**Distributional Kernel Mean Embeddings.** For a fixed choice of $s > d/2$, we work with the Sobolev space $W^s(\mathcal{Y})$, which is an RKHS due to the lower bound on $s$ (Adams & Fournier, 2003, Thm. 4.12). Let $k : \mathcal{Y} \times \mathcal{Y} \to \mathbb{R}$, $k_y := k(y, \cdot)$, be the kernel associated with $W^s(\mathcal{Y})$, and let $(\mathcal{H}, \langle \cdot, \cdot \rangle_{\mathcal{H}})$ denote the corresponding RKHS with norm $\|\cdot\|_{\mathcal{H}}$ and unit ball $\mathcal{H}_1 \subset \mathcal{H}$. By Sobolev embedding and compactness of $\mathcal{Y}$, the kernel $k$ is bounded. Consequently, for every $P \in \mathcal{P}(\mathcal{Y})$, the map $h \mapsto Ph$ defines a bounded linear functional on $\mathcal{H}$, and admits a kernel mean embedding $m(P) \in \mathcal{H}$ given by the Bochner integral $m(P) = \int k_y \, dP(\mathbf{y})$ (Smola et al., 2007; Sriperumbudur et al., 2010). Identifying $\mathcal{H}$ isometrically with a subspace of $\ell^\infty(\mathcal{H}_1)$ via the map $J(h) : f \in \mathcal{H}_1 \mapsto \langle h, f \rangle_{\mathcal{H}}$, the composition $J \circ m$ embeds $\mathcal{P}(\mathcal{Y})$ into $\ell^\infty(\mathcal{H}_1)$. Since $W^s(\mathcal{Y})$ contains all polynomial functions restricted to $\mathcal{Y}$, $W^s(\mathcal{Y})$ is dense in $\mathcal{C}(\mathcal{Y})$ by the Stone-Weierstrass theorem, implying that $k$ is a $c$-universal kernel (Steinwart, 2001, Thm. 9) and characteristic (Micchelli et al., 2006). Hence, the KME map $m$ is injective and induces the maximum mean discrepancy $\operatorname{MMD}(\mu, \nu) := \|m(\mu) - m(\nu)\|_{\mathcal{H}} = \|\mu - \nu\|_{\ell^\infty(\mathcal{H}_1)}$, which metrizes $\mathcal{P}(\mathcal{Y})$.

Following Muandet et al. (2017), we represent counterfactual distributions via their KMEs. The counterfactual mean embedding for $P_a$ is

$$\psi^a(P) = \int \mathbb{E}_P[k_y \mid A = a, X = x] \, dP_X(x). \quad (4)$$

Let $\Psi(P) := \left( \psi^1(P), \psi^0(P) \right)$. Muandet et al. (2021) define the *kernel treatment effect* (KTE) as the MMD between $\psi^1(P)$ and $\psi^0(P)$. A natural estimator of the KTE is obtained by estimating $\psi^a(P)$ via inverse propensity weighting (IPW) (Imbens, 2004). While the resulting estimator is unbiased when the propensity score is known (Muandet et al., 2017, Thm. 5), valid root-$n$ inference in double machine learning settings typically requires debiasing based on the efficient influence function (Luedtke & Chung, 2024).

**Sinkhorn Divergence.** The EOT problem defined in (2) between measures $\mu$ and $\nu$ admits a unique optimal coupling $\pi^{(\mu,\nu)}$. The dual EOT problem admits optimal potentials $(\varphi_1^{(\mu,\nu)}, \varphi_0^{(\mu,\nu)}) \in L^1(\mu) \times L^1(\nu)$ called entropic potentials. The entropic potentials are unique up to an additive constant; see Appx. B.2 for details. The primal and dual solutions satisfy

$$\frac{d\pi^{(\mu,\nu)}}{d(\mu \otimes \nu)} = \exp\left( \tfrac{1}{\varepsilon} \left( \varphi_1^{(\mu,\nu)} \oplus \varphi_0^{(\mu,\nu)} - c \right) \right), \quad (5)$$

where $(\varphi_1^{(\mu,\nu)} \oplus \varphi_0^{(\mu,\nu)})(y_1, y_2) = \varphi_1^{(\mu,\nu)}(y_1) + \varphi_0^{(\mu,\nu)}(y_2)$, and $c(y_1, y_2) = \|y_1 - y_2\|^2/2$. Since $\operatorname{OT}_\varepsilon(\mu, \mu) \ne 0$,

the EOT cost is not a divergence. The Sinkhorn divergence (Ramdas et al., 2017; Feydy et al., 2019) corrects this through the centering

$$S_\varepsilon(\mu, \nu) = \text{OT}_\varepsilon(\mu, \nu) - \tfrac{1}{2}[\text{OT}_\varepsilon(\mu, \mu) + \text{OT}_\varepsilon(\nu, \nu)]. \quad (6)$$

It admits dual representation $S_\varepsilon(\mu, \nu) = \mu v_1^{(\mu,\nu)} + \nu v_0^{(\mu,\nu)}$ where $(v_1^{(\mu,\nu)}, v_0^{(\mu,\nu)})$ are the *centered entropic potentials*

$$v_1^{(\mu,\nu)} := \varphi_1^{(\mu,\nu)} - \varphi_1^{(\mu,\mu)}, \quad v_0^{(\mu,\nu)} = \varphi_0^{(\mu,\nu)} - \varphi_0^{(\nu,\nu)}.$$

**Higher-order Efficient Influence Function.** Let $\mathcal{P}$ be a nonparametric statistical model and $\Phi : \mathcal{P} \to \mathbb{R}$ a parameter that is pathwise differentiable at $P \in \mathcal{P}$ (van der Vaart, 1991). The efficient influence function (EIF) at $P$, denoted by $\dot\Phi_P \in L_0^2(P)$, characterizes the semiparametric efficiency bound for estimating $\Phi(P)$ (Bickel et al., 1993). For any regular parametric submodel $(P_t, t \in [0, \delta)) \subset \mathcal{P}$ with $P_0 = P$ and (Fisher) score function $s$—which take the form $s = \frac{d}{dt} \log p_t|_{t=0}$ under regularity conditions—pathwise differentiability implies $\frac{d}{dt}\Phi(P_t)|_{t=0} = \langle \dot\Phi_P, s \rangle_{L^2(P)}$. This representation directly generates the usual notion of a gradient from calculus, with the EIF playing the role of a gradient and the score playing the role of a direction along which an input to a function is perturbed. The EIF can be used to construct a one-step estimator that corrects for first-order plug-in bias and is semiparametrically efficient under regularity conditions. However, for nonnegative functionals such as the STE, first-order linearization may be insufficient under boundary null hypotheses where the first-order term vanishes (Luedtke et al., 2019; Williamson et al., 2023).

Following Robins et al. (2008) and van der Vaart (2014), we therefore employ second-order influence functions, which arise from the quadratic term in a higher-order von Mises expansion. Formally, the second-order influence function is a symmetric, degenerate kernel $\ddot\Phi_P \in L^2(P^{\otimes 2})$, satisfying:

$$\frac{d^2}{dt^2}\Phi(P_t)\Big|_{t=0} = \mathbb{E}_{P^{\otimes 2}}[\ddot\Phi_P(Z_1, Z_2)\, r(Z_1, Z_2)],$$

where $r$ is the second-order score function (Small & McLeish, 2011). For paths of the form $P_t = (1 + ts)P$, this reduces to $s \otimes s$. In a nonparametric model, there is only one second-order influence function, and this is known as the second-order *efficient* influence function (Robins et al., 2008). The second-order EIF is analogous to the Hessian from multivariable calculus. Technical details on second-order tangent spaces, score functions, and influence functions are deferred to Appx. B.1.

An unfortunate fact uncovered in the literature is that most functionals of interest fail to be smooth enough to admit a second-order EIF (Robins et al., 2008; van der Vaart, 2014). In the DTE setting, this appears to be true for counterfactual $f$-divergences, complicating the study of estimators'

limiting distributions under the null (Kennedy et al., 2023, Sec. 5.2). A major finding of this work is that the Sinkhorn treatment effect is smooth enough to admit a second-order EIF under the null of equal counterfactual distributions. We leverage this property to construct a bias-corrected estimator of the STE with tractable asymptotic behavior.

## 3. Sinkhorn Treatment Effect

Let $P \in \mathcal{P}$ denote a data-generating distribution and fix entropic regularization $\varepsilon > 0$. We define the Sinkhorn treatment effect from (1) as the functional $\mathcal{S} : \mathcal{P} \to \mathbb{R}$ given by $\mathcal{S}(P) = S_\varepsilon \circ J \circ \Psi(P)$. In this chaining argument, first, $\Psi$ maps $P$ to the pair of counterfactual mean embeddings from (4). Second, $J$ is the canonical isometric embedding of $\mathcal{H}$ in $\ell^\infty(\mathcal{H}_1)$, as discussed in Sec. 2. For notational convenience, we use $J$ to denote the induced canonical embedding of the product space $\mathcal{H} \times \mathcal{H}$. Lastly, $S_\varepsilon$ computes the Sinkhorn divergence between the counterfactual distributions.

The goal of this section is to characterize the smoothness properties of $\mathcal{S}$ and leverage them for efficient semiparametric estimation. Our analysis builds on recent advances in the differentiability theory of EOT (Goldfeld et al., 2024; Kokot & Luedtke, 2025). In particular, Goldfeld et al. (2024) analyze the Hadamard differentiability of the map $(\mu, \nu) \mapsto S_\varepsilon(\mu, \nu)$ in the $\ell^\infty(B^s) \times \ell^\infty(B^s)$ topology, where $B^s$ is a unit ball in $\mathcal{C}^s(\mathcal{Y})$. Following this, we embed $\mathcal{P}(\mathcal{Y})$ into the normed linear space $\ell^\infty(\mathcal{H}_1)$ via the KME, and view the Sinkhorn divergence as a smooth functional $S_\varepsilon : \mathcal{P}(\mathcal{Y}) \times \mathcal{P}(\mathcal{Y}) \subset \ell^\infty(\mathcal{H}_1) \times \ell^\infty(\mathcal{H}_1) \to \mathbb{R}$. Such identifications have been employed in literature on other spaces to derive limit theorems for EOT objects—see Appx. C.

We show that this smoothness propagates through the causal composition defining $\mathcal{S}$, yielding first-order pathwise differentiability at all $P \in \mathcal{P}$ and second-order pathwise differentiability under the null that $P_1 = P_0$. We present the corresponding first- and second-order EIFs in what follows.

To express these influence functions compactly, we introduce linear expectation operators acting on $L^2(P)$. For any $P \in \mathcal{P}(\mathcal{Z})$, we define the following three linear operators $P_X, P_{A|X}, P_{Y|A,X} : L^2(P) \to L^2(P)$ such that for any $f \in L^2(P)$, the evaluations at $z = (x, a, y)$ satisfy

$$\left(P_{Y|A,X}f\right)(z) := \int_{\mathcal{Y}} f(x, a, y')\, P(dy' \mid A = a, X = x)$$

$$\left(P_{A|X}f\right)(z) := \sum_{a' \in \{1,0\}} f(x, a', y)\, e_P(a' \mid x)$$

$$(P_X f)(z) := \int_{\mathcal{X}} f(x', a, y)\, P(dx'). \quad (7)$$

These operators provide an alternative way to write standard augmented inverse probability weighted influence functions (Robins et al., 1994). For example, for $\theta(P) := \int \eta_P(x) P_X(dx) = \int h_P(x, a, y) dP(x, a, y)$

with $\eta_P(x) := E_P[Y|A = 1, X = x]$ and $h_P(x, a, y) := \frac{a}{e_P(1|x)}y$, that influence function is

$$(x, a, y) \mapsto \frac{a}{e_P(a|x)}\{y - \eta_P(x)\} + \eta_P(x) - \theta(P)$$
$$= \left(I - \left(I - (I - P_X)P_{A|X}\right) P_{Y|A,X}\right) h_P.$$

This representation allows for a concise expression of the influence functions of $\mathcal{S}$, particularly in the second-order case. We begin by presenting the first-order result.

**Lemma 3.1** (First-order differentiability). *The parameter* $\mathcal{S} : \mathcal{P} \to \mathbb{R}$ *is pathwise differentiable at all* $P \in \mathcal{P}$ *with first-order efficient influence function*

$$\dot{\mathcal{S}}_P = \left(I - \left(I - (I - P_X)P_{A|X}\right) P_{Y|A,X}\right) f_P, \quad (8)$$

*where* $f_P(x, a, y) = v_a^{(P_1, P_0)}(y)/e_P(a|x)$.

The proof of Lem. 3.1 relies on Hadamard differentiability of $S_\varepsilon$ (Kokot & Luedtke, 2025) and pathwise differentiability of the Hilbert-valued parameter $\Psi$ (Luedtke & Chung, 2024), and is provided in Appx. D.1.

Before establishing the second-order pathwise differentiability of $\mathcal{S}$, we set some kernel operator notation. We define the *self-transport kernel*, written as the density of the same-marginal Schrödinger bridge: $\xi_\mu = d\pi^{(\mu,\mu)}/d(\mu \otimes \mu)$ as in (5) with $\mu = \nu$. We define two kernel operators $T_\mu : \mathcal{C}(\mathcal{Y}) \to \mathcal{C}(\mathcal{Y})$ and $H_\mu : \mathcal{M}(\mathcal{Y}) \to \mathcal{C}(\mathcal{Y})$ as,

$$T_\mu f = \int \xi_\mu(\cdot, y) f(y) d\mu(y), \quad H_\mu \gamma = \int \xi_\mu(\cdot, y) d\gamma(y). \quad (9)$$

Let $\mathcal{C}(\mathcal{Y})/\mathbb{R}$ denote the quotient space of all continuous functions identified up to shifts by constant functions, and denote by $[f]$ the corresponding equivalence class of $f \in \mathcal{C}(\mathcal{Y})$. Using the operators introduced above, define

$$K_\mu : \mathcal{M}_0(\mathcal{Y}) \to \mathcal{C}(\mathcal{Y})/\mathbb{R}, \quad K_\mu = \varepsilon(I - T_\mu^2)^{-1}H_\mu.$$

The operator $K_\mu$ is termed as the Hadamard operator of Sinkhorn divergence in Kokot & Luedtke (2025) and Gonzalez-Sanz et al. (2022), and plays a central role in establishing second-order pathwise differentiability of STE. Its definition relies on lifting $T_\mu$ to quotient spaces: since $T_\mu \mathbb{1}_\mathcal{Y} = \mathbb{1}_\mathcal{Y}$, $T_\mu$ has a nontrivial null space spanned by constant functions, rendering it non-invertible on $\mathcal{C}(\mathcal{Y})$. As shown in Thm. 3.8 in Lavenant et al. (2024), $T_\mu$ is a well-defined linear operator on $\mathcal{C}(\mathcal{Y})/\mathbb{R}$, and $(I - T_\mu^2)^{-1}$ exists and is a bounded linear operator on $\mathcal{C}(\mathcal{Y})/\mathbb{R}$ (see Section 3 of Lavenant et al., 2024).

Since $\mathcal{C}(\mathcal{Y})/\mathbb{R}$ is canonically dual to $\mathcal{M}_0(\mathcal{Y})$, the bilinear form $\gamma_0' K_\mu \gamma_0$ is well-defined for all $\gamma_0, \gamma_0' \in \mathcal{M}_0(\mathcal{Y})$. As shown in Kokot & Luedtke (2025), $K_\mu$ is precisely the Hadamard operator of Sinkhorn divergence, governing the

second-order expansion of the Sinkhorn divergence. We denote by $k_\mu : \mathcal{Y} \times \mathcal{Y} \to \mathbb{R}$ the associated symmetric kernel, representing the bilinear form $\gamma_1(K_\mu\gamma_2)$ as $(\gamma_1 \otimes \gamma_2)k_\mu$ for $\gamma_1, \gamma_2 \in \mathcal{M}_0(\mathcal{Y})$. The explicit form of $k_\mu$ is derived in Lem. C.1. Letting $\mathcal{H}_0 := \{P \in \mathcal{P} : P_1 = P_0\}$ with $(P_1, P_0)$ as defined in (3), we now state our second-order pathwise differentiability result.

**Theorem 3.2** (Second-order differentiability under the null). *At any* $P \in \mathcal{H}_0$, $\mathcal{S} : \mathcal{P} \to \mathbb{R}$ *is second-order pathwise differentiable with the second-order efficient influence function*

$$\ddot{\mathcal{S}}_P = \left(I - \left(I - (I - P_X)P_{A|X}\right) P_{Y|A,X}\right)^{\otimes 2} g_P, \quad (10)$$

*where* $g_P(z_1, z_2) = \omega_P(a_1, x_1) \omega_P(a_2, x_2) k_{P_1}(y_1, y_2)$ *for* $z_j = (x_j, a_j, y_j)$ *and* $\omega_P(a, x) = (2a - 1)/e_P(a|x)$.

For shorthand, we will occasionally write $g_P = \omega_P^{\otimes 2} \odot k_{P_1}$, identifying $\omega_P^{\otimes 2}$ and $k_{P_1}$ with their canonical extensions to $\mathcal{Z}^2$ that ignore the $y$ and $(x, a)$ coordinates, respectively.

A second-order analysis of $\mathcal{S}$ is possible under the null because the first derivative of the Sinkhorn divergence vanishes in these cases. This eliminates, through the second-order chain rule, the term that would ordinarily involve the second-order pathwise derivative of the KME map $\Psi$, whose presence would typically preclude the existence of a second-order EIF (van der Vaart, 2014, pp. 681–682).[1] Hence the second-order pathwise derivative of STE depends only on the first-order perturbation of $\Psi$, equivalently on its EIF, together with the second derivative of $S_\varepsilon$. A similar phenomenon was observed in Luedtke et al. (2019) for a test based on a different discrepancy measure.

## 4. One-Step Estimation of STE

We now leverage the first- and second-order EIFs of STE to construct one-step bias-corrected estimators of $\mathcal{S}(P)$. Let $P^* \in \mathcal{P}$ be our data-generating distribution and $Z_1, \ldots, Z_n$ be independent draws from $P^*$, with corresponding empirical distribution denoted by $P_n$. Let $\widehat{P}$ be an initial estimator of $P^*$ constructed from an independent sample of size $n$. For notational ease, we present our one-step estimators under the sample splitting approach, assuming $P_n$ and $\widehat{P}$ are based on independent data. A cross-fitting approach (Schick, 1986; Klaassen, 1987; Chernozhukov et al., 2018) is recommended, and the resulting efficiency is discussed for both first- and second-order one-step estimators. Our test for the causal null is based on the second-order one-step estimator of $\mathcal{S}(P^*)$ for a fixed $\varepsilon > 0$. To reduce sensitivity to the choice of the unknown entropic regularization parameter $\varepsilon$, we also provide to a way to aggregate tests over a finite grid of $\varepsilon$ values.

---

[1]The second-order analysis is also possible for the squared counterfactual MMD. We derive its second-order EIF in Lem. F.2, which to the best of our knowledge, is a novel contribution.

### 4.1. One-Step Estimator

The one-step estimator of $\mathcal{S}^* := \mathcal{S}(P^*)$ is defined as $\widehat{\mathcal{S}} := \mathcal{S}(\widehat{P}) + P_n\dot{\mathcal{S}}_{\widehat{P}}$. We study conditions under which a cross-fitted version of $\widehat{\mathcal{S}}$ is asymptotically linear and semiparametrically efficient. As is standard in semiparametric theory, efficiency follows if $\widehat{\mathcal{S}} - \mathcal{S}(P^*) = P_n\dot{\mathcal{S}}_{P^*} + o_p(n^{-1/2})$, in which case the asymptotic variance attains the efficiency bound (Bickel et al., 1993; Chernozhukov et al., 2018). This expansion is obtained via the decomposition $\widehat{\mathcal{S}} - \mathcal{S}(P^*) - P_n\dot{\mathcal{S}}_{P^*} = \mathcal{R}_n + \mathcal{D}_n$, where $\mathcal{R}_n := \mathcal{S}(\widehat{P}) + P^*\dot{\mathcal{S}}_{\widehat{P}} - \mathcal{S}(P^*)$ is the remainder term from first order von Mises expansion and $\mathcal{D}_n := (P_n - P^*)(\dot{\mathcal{S}}_{\widehat{P}} - \dot{\mathcal{S}}_{P^*})$ is a stochastic drift term. Under appropriate convergence rates for the nuisance components of $\widehat{P}$, we show that both terms are $o_p(n^{-1/2})$ in Appx. E.1. Consequently, $\widehat{\mathcal{S}}$ is asymptotically linear, satisfying the weak convergence

$$\sqrt{n}\big(\widehat{\mathcal{S}} - \mathcal{S}^*\big) \xrightarrow{d} \mathcal{N}\big(0, \mathbb{E}_{P^*}\big[\dot{\mathcal{S}}_{P^*}^2(Z)\big]\big). \qquad (11)$$

This asymptotic normality enables the construction of asymptotically valid Wald-type confidence intervals for $\mathcal{S}^*$.

Under the null hypothesis ($P^* \in \mathcal{H}_0$), the first-order EIF is degenerate, i.e. $\dot{\mathcal{S}}_P = 0$, since $\upsilon_1^{(P_1,P_1)} = \upsilon_0^{(P_1,P_1)} = 0$. As a consequence, the first-order limit distribution in (11) collapses to a point mass, rendering the first-order inference invalid. To address this, we construct a second-order one-step estimator based on a Newton–Raphson bias correction, in the spirit of Robins et al. (2008), which attains $n$-rate convergence rate to $\mathcal{S}^*$ under the null. Inference based on this second-order expansion yields asymptotically valid hypothesis tests for detecting DTEs, with the second-order one-step estimator serving as the test statistic.

The second-order one-step estimator for $\mathcal{S}^*$ is defined as

$$\overline{\mathcal{S}} := \mathcal{S}(\widehat{P}) + P_n\dot{\mathcal{S}}_{\widehat{P}} + \frac{1}{2}\mathbb{U}_n\ddot{\mathcal{S}}_{\widehat{P}}, \qquad (12)$$

where $\mathbb{U}_n$ denotes the canonical U-statistic operator from the same samples as $P_n$. However, second-order pathwise differentiability of $\mathcal{S}$ is only established under the null, and therefore the second-order EIF $\ddot{\mathcal{S}}_{\widehat{P}}$ need not be well-defined at a general $\widehat{P} \notin \mathcal{H}_0$. To address this, we construct an extension of the second-order EIF to distributions $P \notin \mathcal{H}_0$. This extension is defined by the same analytic expression as in (10), where for notational simplicity we denote this extension by the same symbol, $\ddot{\mathcal{S}}_P$. Since $k_{P_1} \neq k_{P_0}$ when $P \notin \mathcal{H}_0$, the definition in non-null cases depends on the choice of counterfactual distribution used to construct the self-transport kernel. We fix this choice to $\widehat{P}_1$ in our theoretical arguments for concreteness, though any convex combination of $\widehat{P}_1$ and $\widehat{P}_0$ would enjoy similar guarantees. Note that $\ddot{\mathcal{S}}_P$ is already $P$-centered, and therefore, does not require further centering common in second-order one-step estimation (Luedtke et al., 2019).

### 4.2. Inference Under Null

We now study the asymptotic properties of (12). Define the centering operator $C_{P^*}$ such that for any measurable kernel $\ell : \mathcal{Z} \to \mathcal{Z} \to \mathbb{R}$, $C^*\ell(z, z')$ is equal to

$$\ell(z, z') - P^*\ell(\cdot, z') - P^*\ell(z, \cdot) + [(P^*)^{\otimes 2}\ell]\mathbb{1},$$

where $\mathbb{1}$ is the constant one function. The following breakdown is key to the inference of $\overline{\mathcal{S}}$:

$$\overline{\mathcal{S}} - \mathcal{S}^* - \frac{1}{2}\mathbb{U}_n\ddot{\mathcal{S}}_{P^*} = \mathcal{D}_n + \mathcal{U}_n + \mathcal{R}_n,$$

where $\mathcal{D}_n, \mathcal{U}_n, \mathcal{R}_n$ are drift, U-process consistency, and remainder terms, respectively, and are defined as

$$\mathcal{D}_n = (P_n - P^*)\left(\dot{\mathcal{S}}_{\widehat{P}} + \int \ddot{\mathcal{S}}_{\widehat{P}}(z, \cdot)\, dP^*(z)\right),$$
$$\mathcal{U}_n = \frac{1}{2}\mathbb{U}_n(C_{P^*}\ddot{\mathcal{S}}_{\widehat{P}} - C_{P^*}\ddot{\mathcal{S}}_{P^*}),$$
$$\mathcal{R}_n = \mathcal{S}(\widehat{P}) + (P^* - \widehat{P})\dot{\mathcal{S}}_{\widehat{P}} + \frac{1}{2}(P^* - \widehat{P})^2\ddot{\mathcal{S}}_{\widehat{P}} - \mathcal{S}(P^*).$$

In the next result, we show that all three terms above are $o_P(n^{-1})$ under suitable conditions. Consequently, the asymptotic distribution of $n(\overline{\mathcal{S}} - \mathcal{S}(P^*))$ is determined by the limiting distribution of $n\mathbb{U}_n\ddot{\mathcal{S}}_{P^*}/2$. The drift term $\mathcal{D}_n$ is $o_P(n^{-1})$ provided that the integrand $\dot{\mathcal{S}}_{\widehat{P}} + \int \ddot{\mathcal{S}}_{\widehat{P}}(z, \cdot)\, dP^*(z)$ is $o_P(n^{-1/2})$ in $L^2(P^*)$ norm (Lem. E.1). This result follows from a conditioning argument that combines Chebyshev's inequality with the dominated convergence theorem. We prove the desired rate on our integrand in Lem. E.5. The U-process term $\mathcal{U}_n$ is also $o_P(n^{-1})$ using similar arguments using Chebyshev's inequality, only requiring $L^2(P^* \otimes P^*)$ convergence of $C_{P^*}\ddot{\mathcal{S}}_{\widehat{P}}$ to $\ddot{\mathcal{S}}_{P^*}$—see Lem. E.6. Finally, the remainder term $\mathcal{R}_n$ is $o_p(n^{-1})$ under regularity conditions commonly satisfied by smooth statistical functionals (Luedtke et al., 2019; Robins et al., 2009). We verify these conditions for $\mathcal{R}_n$ in Lem. E.8 and discuss the feasibility of the required assumptions later in this section. Similar to the first-order one-step estimator, a cross-fitting approach for $\overline{\mathcal{S}}$ is recommended (Kim & Ramdas, 2024), as it can improve sample efficiency.

For brevity, we use the expectation operator shorthand $P_{Y|a,X}$ defined in (33), $\hat{e}$ for $e_{\widehat{P}}$, and $e$ for $e_{P^*}$. Recall $k_\mu$ is the kernel associated with operator $K_\mu$ and defined in Lem. C.1. We now state the main theorem on asymptotics of the second-order one-step estimator $\overline{\mathcal{S}}$.

**Theorem 4.1** (Null distribution). *Let $P^* \in \mathcal{H}_0$. Suppose:*

1. *the estimated nuisance error term*

$$\int \left(\frac{\hat{e}(a\,|\,x)}{e(a\,|\,x)} - 1\right)\big(\widehat{P}_{Y\,|\,a,x} - P^*_{Y\,|\,a,x}\big)k_{\widehat{P}_1}(\cdot, y)\, dP^*(x)$$

*is $o_p(n^{-1/2})$, uniformly in $y \in \mathcal{Y}$, $a \in \{1, 0\}$,*

2. $\|C_{P^*}\ddot{\mathbb{S}}_{\widehat{P}} - C_{P^*}\ddot{\mathbb{S}}_{P^*}\|_{L^2(P^*\otimes P^*)} \xrightarrow{p} 0$, and

3. the von Mises remainder satisfies $\mathcal{R}_n = o_p(n^{-1})$.

Then,

$$n\overline{\mathbb{S}} = \frac{n}{2}\mathbb{U}_n\ddot{\mathbb{S}}_{P^*} + o_p(1) \xrightarrow{d} \sum_{j=1}^{\infty} \frac{\lambda_j}{2}(N_j^2 - 1), \qquad (13)$$

where $\{\lambda_j\}_{j=1}^{\infty}$ are the eigenvalues of the integral operator $f \mapsto \int \ddot{\mathbb{S}}_{P^*}(\cdot, z)f(z)\,dP^*(z)$, counted with multiplicity, and $\{N_j\}_{j=1}^{\infty}$ are iid standard normal random variables.

The proof of Thm. 4.1 is presented in Appx. E.2. We now discuss the feasibility of the conditions. We let the initial estimator $\widehat{P}$ of the data-generating distribution $P^*$ satisfies $\|\widehat{P}_a - P_a^*\|_{\ell^\infty(\mathcal{H}_1)} = \mathcal{O}_p(n^{-r})$ for some $1/4 < r < 1/2$ and $a \in \{1, 0\}$. The first condition is standard in the doubly robust causal inference literature (Luedtke & Chung, 2024) and is implied by $o_p(n^{-1/4})$ convergence of the nuisance estimators in their appropriate norms. The second condition is a consistency requirement ensuring stability of the second-order term; it follows from convergence of $\omega_{\widehat{P}}^{\otimes 2} \odot k_{\widehat{P}_1}$ to $\omega_{P^*}^{\otimes 2} \odot k_{P_1^*}$ in $L^2(P^* \otimes P^*)$, together with a uniform bound on the Radon–Nikodym derivatives $d\widehat{P}/dP^*$ over $\widehat{P} \in \mathcal{P}$. The third condition controls the remainder of the second-order von Mises expansion and is $o_p(n^{-1})$ under $n^{-1/3}$-rate conditions on the nuisance estimators (Lem. E.8).

### 4.3. Formulation of Test

To test the null hypothesis that $P^* \in \mathcal{H}_0$, we use the test statistic $n\overline{\mathbb{S}}$. Under the null, Thm. 4.1 establishes that $n\overline{\mathbb{S}}$ converges weakly to a weighted sum of centered chi-squared distributions. A non-conservative test of nominal level $\alpha \in (0, 1)$ rejects the null hypotheses when $n\overline{\mathbb{S}}$ exceeds the $(1-\alpha)$th quantile of the limiting distribution, $q_{1-\alpha}$. The weak convergence in (13) implies the corresponding test asymptotically attains nominal type I error:

$$\limsup_{n\to\infty} P^*(n\overline{\mathbb{S}} \geq q_{1-\alpha}) = \alpha. \qquad (14)$$

Moreover, if $\hat{q}_{1-\alpha}$ is a consistent estimator of $q_{1-\alpha}$, then the same conclusion holds with $q_{1-\alpha}$ replaced by $\hat{q}_{1-\alpha}$ in (14). To obtain such an estimator, we use the general approximation strategy of Gretton et al. (2009), which was adapted to tests based on second-order influence functions in Luedtke et al. (2019). This approximates the spectrum of the operator $f \mapsto \int \ddot{\mathbb{S}}_{P^*}(\cdot, z)f(z)dP(z)$ through that of an empirical Gram matrix. In our case, this is the Gram matrix of the kernel $\ddot{\mathbb{S}}_{P^*}$, which depends on $P^*$ through three nuisance parameters: the propensity $e^*$, outcome regression model $P_{Y|A,X}^*$, and entropic potentials $(v_1^{(P_1^*, P_0^*)}, v_0^{(P_1^*, P_0^*)})$. Since these nuisances are unknown,

we estimate them to obtain $\ddot{\mathbb{S}}_{\widehat{P}}$, form its empirical Gram matrix, and compute its eigenvalues $\{\hat{\lambda}_j\}_{j=1}^n$. The limiting distribution is then approximated by $\sum_{j=1}^n \hat{\lambda}_j(N_j^2 - 1)$, and its $(1-\alpha)$-th quantile $\hat{q}_{1-\alpha}$ is obtained via Monte Carlo simulation. Under sufficient regularity—specifically, if the nuisance estimators converge at rates fast enough to guarantee operator-norm consistency of the plug-in kernel and continuity of the limiting spectral distribution—nominal type 1 error control follows by standard kernel spectral approximation and perturbation arguments (Shawe-Taylor et al., 2005; Zwald & Blanchard, 2005; Rosasco et al., 2010).

Regarding power, the proposed test will reject fixed alternatives $P^* \notin \mathcal{H}_0$ with probability tending to 1. This follows from the relation $\overline{\mathbb{S}} = \widehat{\mathbb{S}} + \mathbb{U}_n\ddot{\mathbb{S}}_{\widehat{P}}/2$ between the second- and first-order bias-corrected estimators, the approximate 1-degeneracy of $\mathbb{U}_n\ddot{\mathbb{S}}_{\widehat{P}}$ resulting from its $\widehat{P}$-centering, and the asymptotic normality in (11)—see Lem. E.9.

### 4.4. STEAgg: Max-Aggregated Test

In finite samples, power of the STE-based test can depend on the choice of entropic regularization parameter $\varepsilon$: small values cause dispersion in the asymptotic distribution of the U-statistic, while large values oversmooth the discrepancy and reduce power. To reduce sensitivity to this choice, we propose a multiple testing procedure, STEAgg, that aggregates evidence over a finite grid of stable regularization parameters $\Xi = \{\varepsilon_1, \ldots, \varepsilon_m\} \subset (0, \infty)$. We also establish asymptotic nominal type I error guarantee for STEAgg.

For each $\varepsilon \in \Xi$, let $\mathcal{T}_{n,\varepsilon} = n\overline{\mathbb{S}}$ denote the test statistic with explicit dependence on $n$ and $\varepsilon$. Under the null, Thm. 4.1 shows that $\mathcal{T}_{n,\varepsilon} \xrightarrow{d} W(\varepsilon)$, where $W(\varepsilon)$ is the Gaussian chaos limit in (13), induced by the spectral decomposition of the degenerate kernel $\ddot{\mathbb{S}}_{\varepsilon, P^*}$, with explicit dependence on $\varepsilon$. We define the aggregated test statistic

$$\mathcal{T}_n^{\text{agg}} = \max_{\varepsilon \in \Xi} \frac{\mathcal{T}_{n,\varepsilon}}{q_{n,\varepsilon}}, \qquad (15)$$

where $q_{n,\varepsilon}$ consistently estimates the $(1-\beta)$ quantile of $W(\varepsilon)$ for a pre-specified $\beta \in (0, 1)$. Normalization by $q_{n,\varepsilon}$ calibrates each statistic $\mathcal{T}_{n,\varepsilon}$ to the scale of its own marginal null distribution. This ensures that aggregation compares evidence for different $\varepsilon$ on a common scale, rather than allowing monotonic dependence on $\varepsilon$ to dominate the maximum.

**Theorem 4.2** (Asymptotic behavior of STEAgg). *If the null holds ($P^* \in \mathcal{H}_0$), the conditions of Thm. 4.1 hold, and $q_{n,\varepsilon} \xrightarrow{P} q_\varepsilon > 0$ for each $\varepsilon \in \Xi$, then*

$$\mathcal{T}_n^{\text{agg}} \xrightarrow{d} W^{\text{agg}} := \max_{\varepsilon \in \Xi} W(\varepsilon)/q_\varepsilon.$$

*If $P^* \notin \mathcal{H}_0$ is a fixed alternative, the nuisance estimators satisfy the conditions in Lems. E.2 and E.3,*

for each $\varepsilon \in \Xi$, $\ddot{\mathbb{S}}_{\varepsilon,P^*} \in L^2(P^* \otimes P^*)$, and $\|C_{P^*}\ddot{\mathbb{S}}_{\varepsilon,\widehat{P}} - C_{P^*}\ddot{\mathbb{S}}_{\varepsilon,P^*}\|_{L^2(P^* \otimes P^*)} = o_p(1)$, then the proposed test will reject the null with probability converging to 1 as $n \to \infty$.

The proof proceeds by first deriving the joint limit distribution of $(\mathcal{T}_{n,\varepsilon_1}, \ldots, \mathcal{T}_{n,\varepsilon_m})$, then applying Slutsky's theorem to obtain the limiting distribution—see Appx. H.

## 5. Finite-Sample Estimation

We derive explicit, computable formulas for the first- and second-order one-step estimators, $\widehat{\mathbb{S}}$ and $\overline{\mathbb{S}}$. Nuisance parameters are estimated on one data split, with all remaining computations performed on the empirical distribution of the other split. In finite samples, expectation operators in (8) and (10) are replaced by matrix operations, kernel evaluations by Gram matrices, and function evaluations by vectors, yielding implementable formulas in closed form.

**Sample splitting and nuisance estimation.** Let $\mathcal{D}_n^1 = \{z_i\}_{i=1}^n$ and $\mathcal{D}_n^2 = \{z_i'\}_{i=1}^n$ be independent samples from $P$, where $z_i = (x_i, a_i, y_i)$ and $z_i' = (x_i', a_i', y_i')$. Using $\mathcal{D}_n^2$, we estimate the nuisances: the propensity score, outcome regression, centered entropic potentials (consequently self-transport entropic potential). Modern nonparametric learning methods can be used for estimating $e_P$ and $P_{Y \mid A, X}$, while entropic potentials should be computed via the Sinkhorn algorithm (Cuturi, 2013). Let $\mathbf{1}_n$ and $\mathbf{1}_{n \times n}$ denotes the all-ones vector and matrix, respectively.

**Debiasing.** Using the precomputations described in Appx. G, we first construct discrete approximations $(\mathbf{P}_1, \mathbf{P}_0)$ of the counterfactual laws $(P_1, P_0)$ on the atoms of $\mathcal{D}_n^1$. Then, using Algs. 1 and 2, we evaluate the first- and second-order EIFs on $\mathcal{D}_n^1$, yielding $\mathbf{I}^1 \in \mathbb{R}^n$ and $\mathbf{I}^2 \in \mathbb{R}^{n \times n}$. In both cases, the intermediate steps $T_i$ correspond to the four additive components of the operator $(I - P_{Y \mid A, X} + P_{Y, A \mid X} - P)$ appearing in (8) and (10). The resulting one-step estimator is $\widehat{\mathbb{S}}_n = S_\varepsilon(\mathbf{P}^1, \mathbf{P}^0) + (\mathbf{1}_n^\top \mathbf{I}^1)/n$, and the test statistic (12) is $\overline{\mathbb{S}}_n = \widehat{\mathbb{S}}_n + (\mathbf{1}_n^\top \mathbf{I}^2 \mathbf{1}_n - \mathrm{Tr}(\mathbf{I}^2))/2n(n-1)$.

**Computational complexity.** Treating the generic estimators of propensity and outcome regression as black boxes, the procedure's remaining complexity is $O(n^2(n+t))$, with $t$ the number of Sinkhorn iterations. This comprises $O(n^2 t)$ for the Sinkhorn algorithm (Dvurechensky et al., 2018), $O(n^2)$ for the precomputations and Alg. 1, and $O(n^3)$ for Alg. 2. Several strategies can substantially reduce these costs; see Appx. I.3. In particular, low-rank Nyström approximations to the Gibbs kernel matrix used in the Sinkhorn algorithm can reduce the per-iteration Sinkhorn cost from $O(n^2)$ to $O(nr)$, where $r \ll n$ is the Nyström rank. Similar Nyström-based ideas can also be used to accelerate the computations in Algs. 1 and 2.

## 6. Experiments

We evaluate our methods on simulated Gaussian outcomes and microscopy medical image outcomes (Veeling et al., 2018). In both settings, the causal mechanism is simulated to control the treatment effect, so ground truth regarding the null hypothesis is known. We compare our proposed test to an MMD-based distributional treatment effect (shortened as MTE) test constructed using a similar second-order influence function–based debiasing strategy—see Appx. F.

The propensity score nuisance $e_{P^*}$ is estimated using XG-Boost (Chen, 2016) with hyperparameters selected via 5-fold cross-validation. The distribution $P_{Y \mid A, X}^*$ is modeled using a conditional normalizing flow with affine coupling layers, following the RealNVP framework (Dinh et al., 2017). Centered entropic potentials are computed via the Sinkhorn algorithm (Cuturi, 2013). See Appx. I.2 for computational details, memory usage, and wall-clock time for our experiments.

### 6.1. Simulation Study

We conduct a simulation study to assess the finite-sample performance of the proposed one-step STE estimator and test under known ground truth. We consider a low-dimensional model where the observed data $(X, A, Y) \sim P$ is simulated as follows: the covariate $X \in \mathbb{R}^3$ is sampled from a standard normal distribution, the treatment assignment assumes a logistic model on $X$, and the conditional outcome $Y \mid A, X$ follows a Gaussian distribution.

To probe the sensitivity of MTE and STE, we consider two regimes governing the separation between the counterfactual outcome distributions $P_0$ and $P_1$. **Exp (i)** (Mean shift): The counterfactual distributions differ only in their means: $P_0 = \mathcal{N}(0, \Sigma)$ and $P_1 = \mathcal{N}(\theta \mathbf{1}_2, \Sigma)$, where $\theta \geq 0$ controls the magnitude of the shift. **Exp (ii)** (Covariance shift): The counterfactual distributions share the same mean but differ in covariance: $P_0 = \mathcal{N}(0, \Sigma)$ and $P_1 = \mathcal{N}(\mathbf{0}_2, \Sigma + \theta \Delta)$, where $\Delta = uv^\top + vu^\top$ and $u, v$ are the eigenvectors of $\Sigma$. This is a symmetric rank-2 covariance perturbation that induces a shearing-type geometric deformation. The parameter $\theta$ controls the separation between counterfactual distributions, with $\theta = 0$ corresponding to the null and $\theta > 0$ alternatives. Results are averaged over $10^3$ Monte Carlo replications and use the median heuristic to select $\varepsilon$. The exact setup, sensitivity analysis over a finite $\varepsilon$-grid, and max-aggregated tests are reported in Appx. I.1.1.

Fig. 2 reports (I) empirical type I error of the MTE and STE tests at the nominal $5\%$ level, (II) empirical power across values of $\theta$ in Exp (i), (III) empirical power across values of $\theta$ in Exp (ii), (IV) mean squared error of the plug-in and one-

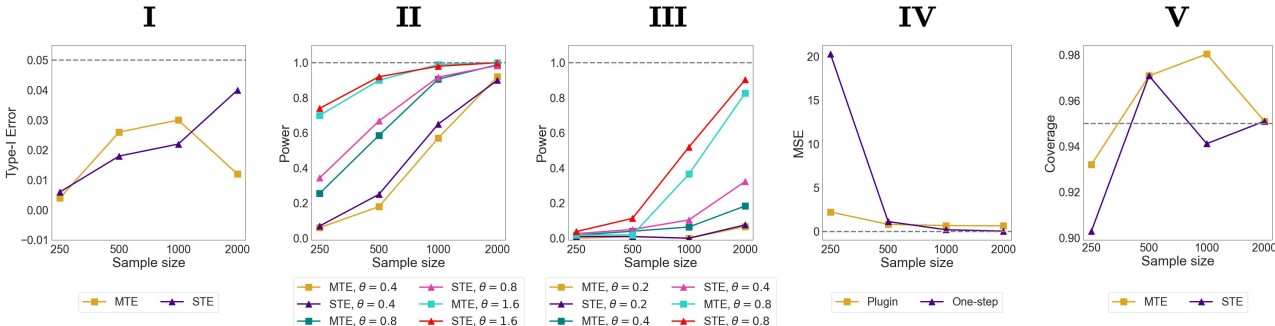

*Figure 2.* **I**: Type I error of the MTE and STE under null ($\theta = 0.0$); **II**: Power of MTE and STE with increasing gap between counterfactual distributions (increasing $\theta$) under Exp (i); **III**: Power of MTE and STE with increasing $\theta$ under Exp (ii); **IV**: Mean squared error of plugin vs one-step STE for $\theta = 1.6$ from Exp (i); **V**: Coverage of Wald-type $95\%$ confidence intervals for $\theta = 1.6$ from Exp (i).

step STE estimators, and (V) empirical coverage of Wald-type $95\%$ confidence intervals. As the sample size increases, both tests approach a nominal type I error and unit power in this setting. Panel (IV) shows that one-step estimation substantially reduces plug-in bias. Though the confidence interval coverages in (V) are similar, the estimand covered by the STE interval better captures the geometry of the problem than does the one covered by the MTE interval. Indeed for Exp (ii), when $\theta = 0.2$, the (MTE, STE) values are $(0.26, 4.12) \times 10^{-3}$, when $\theta = 0.4$, the (MTE, STE) values are $(1.04, 16.42) \times 10^{-3}$, and at $\theta = 0.8$, the (MTE, STE) values are $(4.60, 68.43) \times 10^{-3}$.

## 6.2. Image Dataset

We next consider a setting with high-dimensional, structured outcomes. We use the PatchCamelyon (PCam) dataset (Veeling et al., 2018), which consists of $96 \times 96$ microscopy medical images, labeled by the presence or absence of metastatic tissue. We partition the images into two sets: $\mathcal{D}_1$ (metastatic) and $\mathcal{D}_0$ (non-metastatic), each containing over $100K$ images. To reduce dimensionality, we use the first ten principal components of ResNet-18 (He et al., 2016) embeddings.

We generate synthetic causal data $(X, A, Y)$. Covariates $X \in \mathbb{R}^3$ are drawn from a standard Gaussian distribution and determine both latent disease status and treatment assignment via logistic models. Outcomes are generated by sampling PCam images conditional on disease status and treatment. Specifically, non-diseased units always generate outcomes from $\mathcal{D}_0$. Diseased untreated units generate outcomes from $\mathcal{D}_1$, while diseased treated units generate outcomes from $\mathcal{D}_0$ with covariate-dependent probability $q(x) = \text{expit}\big(\theta + 10^{-2} \mathbf{1}_3^\top x\big)$, and from $\mathcal{D}_1$ otherwise. This construction yields heterogeneous, covariate-dependent treatment effects on the distribution of image outcomes. Each simulation uses $10^4$ observations, split evenly to form

$\widehat{P}$ and $P_n$. Results are averaged over $10^3$ Monte Carlo replications for a range of values of $\theta$ between $[-20, 20]$. The case $\theta = -20$ serves as an approximate null, as treatment success is effectively unobservable at this sample size, while $\theta = 20$ corresponds to near-certain treatment success. Additional implementation details are provided in Appx. I.1.2.

Empirical type I error and power are reported in Fig. 5 against the probability of treatment success. For a moderate sample size of $n = 5000$ and $\alpha = 0.05$, STE reliably controls type I error at 0.04, while MTE has it inflated at roughly 0.11. Given that it fails to control type I error, it is not surprising that MTE has slightly higher power in this setting, though both have power that increases with the probability of success of treatment. As in the previous simulation, STE better captures the geometry of the problem than MTE. When $\theta = 20$, STE $\approx 2$ while MTE $\approx 5 \times 10^{-3}$.

## 7. Conclusion

We propose a novel optimal transport-based DTE that addresses limitations of existing approaches. Unlike kernel-based measures such as MMD, which suffer from flat geometry under limited overlap, and $f$-divergence-based DTEs, which require mutual absolute continuity, our formulation lifts Euclidean geometry to probability spaces and naturally accommodates discrete outcomes. Our contributions invite future work in several directions. One involves addressing the numerical instability from large eigenvalues of $(I - T_\mu^2)^{-1}$. Recent work proposes replacing the Sinkhorn kernel $k_\mu$ with $\xi_\mu$ for faster computations (Kokot & Luedtke, 2025, Lem. 12), though establishing the theoretical validity of this substitution for STE remains open. Another direction is to scale inference to larger datasets by extending our approach to batched Sinkhorn divergences (Peyré et al., 2019).

## Acknowledgements

The authors thank Alex Kokot for helpful discussions. This work was supported by the Patient Centered Outcomes Research Initiative (PCORI, ME-2024C2-39990). The content is solely the responsibility of the authors and does not necessarily represent the official views of the funding agency.

## Impact Statement

This paper develops theoretical and statistical tools for comparing counterfactual outcome distributions using entropic optimal transport. By providing principled estimators and asymptotic characterizations, the proposed methods improve distributional causal analysis in applications such as economics (Firpo, 2007; Chernozhukov et al., 2013), healthcare, and clinical research (Robins et al., 2000), where treatment effects often manifest beyond changes in means.

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

# Appendix

# A. Notations

*Table 1.* Notations.

| Symbol | Description |
|---|---|
| $\mathcal{C}^s(\mathcal{W})$ | Set of functions on $\mathcal{W}$ with continuous derivatives of order $\leq s$. |
| $W^s(\mathcal{W})$ | Sobolev space of order $s$ on $\mathcal{W}$. |
| $\mathcal{P}(\mathcal{W})$ | Borel probability measures on $\mathcal{W}$. |
| $\mathcal{M}(\mathcal{W})$ | Finite signed Radon measures on $\mathcal{W}$. |
| $\mathcal{M}_0(\mathcal{W})$ | Balanced measures: $\{\mu \in \mathcal{M}(\mathcal{W}) : \mu(\mathcal{W}) = 0\}$. |
| $C(\mathcal{W})$ | Continuous functions on $\mathcal{W}$, equipped with supnorm. |
| $Pf$ | Shorthand for $\int f \, dP$. |
| $\|\mathcal{A}\|_{\mathcal{F} \to \mathcal{G}}$ | Operator norm of a bounded linear operator $\mathcal{A} : \mathcal{F} \to \mathcal{G}$. |
| $\odot$ | Elementwise product of functions/tensors. |
| $\otimes$ | Tensor product of functions/tensors. Product measure. |
| $Z = (X, A, Y)$ | Observed data: covariates $X$, treatment $A \in \{0, 1\}$, outcome $Y$. |
| $\mathcal{X} \subset \mathbb{R}^p$ | Covariate space. |
| $\mathcal{Y} \subset \mathbb{R}^d$ | Outcome space. |
| $\mathcal{Z} = \mathcal{X} \times \{0, 1\} \times \mathcal{Y}$ | Sample space for $Z$. |
| $\mathcal{P}$ | Nonparametric statistical model in $\mathcal{P}(\mathcal{Z})$ satisfying conditions in Sec. 2. |
| $e_P(a \mid x)$ | Propensity score $P(A = a \mid X = x)$. |
| $Y_1, Y_0$ | Counterfactual outcomes under treatment/control. |
| $P_1, P_0$ | Counterfactual outcome distributions under treatment/control. |
| $c(x, y)$ | Cost function for optimal transport, assume quadratic $c(x, y) = \|x - y\|^2/2$. |
| $\varepsilon$ | Entropic regularization parameter. |
| $\mathrm{KL}(\cdot \mid \cdot)$ | Kullback-Leibler divergence. |
| $\mathrm{OT}_\varepsilon(\mu, \nu)$ | Entropic optimal transport cost between $\mu$ and $\nu$ with regularization parameter $\varepsilon$. |
| $\pi^{(\mu, \nu)}$ | Unique optimal coupling for $\mathrm{OT}_\varepsilon(\mu, \nu)$. |
| $\varphi_1^{(\mu,\nu)}, \varphi_0^{(\mu,\nu)}$ | Dual entropic potentials. |
| $v_1^{(\mu,\nu)}, v_0^{(\mu,\nu)}$ | Centered entropic potentials between $\mu$ and $\nu$.. |
| $S_\varepsilon(\mu, \nu)$ | Sinkhorn divergence between $\mu$ and $\nu$. |
| $g_\varepsilon : \mathcal{Y} \times \mathcal{Y} \to \mathbb{R}$ | Gaussian kernel $g_\varepsilon = \exp(-c/\varepsilon)$. |
| $\mathcal{G}$ | Gaussian RKHS associated with $g_\varepsilon$. |
| $\mathcal{H}$ | Sobolev RKHS $W^s(\mathcal{Y})$ for some fixed $s > d/2$; inner product $\langle \cdot, \cdot \rangle_\mathcal{H}$. |
| $k : \mathcal{Y} \times \mathcal{Y} \to \mathbb{R}$ | RKHS kernel associated with $\mathcal{H}$. |
| $\mathcal{H}_1$ | Unit ball in $\mathcal{H}$. |
| $k_y$ | Representer $k_y(\cdot) = k(y, \cdot)$. |
| $m(\mu)$ | Kernel mean embedding (KME): $m(\mu) = \int k_y \, d\mu(y)$. |
| $J$ | Canonical embedding $J : \mathcal{H} \hookrightarrow \ell_\infty(\mathcal{H}_1)$ (extended to product spaces). |
| $\mathrm{MMD}(\mu, \nu)$ | $\|m(\mu) - m(\nu)\|_\mathcal{H}$ = maximum mean discrepancy. |
| $\psi_a(P)$ | Counterfactual mean embedding: $\psi_a(P) = \int \mathbb{E}_P[k_Y \mid A = a, X = x] \, dP_X(x)$. |
| $\Psi(P)$ | Pair of counterfactual embeddings $(\psi_1(P), \psi_0(P))$. |
| $\mathcal{S}(P)$ | Sinkhorn treatment effect. |
| $\mathcal{H}_0$ | Null set $\{P : P_1 = P_0\}$. |
| $L^2(P)$ | Square-integrable functions under $P$. |
| $P_X$ | Marginalization operator: $(P_X f)(x, a, y) = \int f(x', a, y) \, P(dx')$. |
| $P_{A\mid X}$ | Treatment-marginalization: $(P_{A\mid X} f)(x, a, y) = \sum_{a' \in \{0,1\}} f(x, a', y) e_P(a' \mid x)$. |
| $P_{Y\mid A,X}$ | Outcome regression operator: $(P_{Y\mid A,X} f)(x, a, y) = \int f(x, a, y') \, P(dy' \mid A = a, X = x)$. |
| $\dot{\mathcal{S}}_P$ | First-order efficient influence function of $\mathcal{S} : Q \mapsto \mathcal{S}(Q)$ at $P$. |
| $\ddot{\mathcal{S}}_P$ | (Extended) Second-order efficient influence function of $\mathcal{S} : Q \mapsto \mathcal{S}(Q)$ at $P$. |
| $\mathcal{S}^*$ | Population STE, $\mathcal{S}^* = \mathcal{S}(P^*)$ for data-generating distribution $P^* \in \mathcal{P}$. |
| $\widehat{\mathcal{S}}$ | First-order one-step estimator of $\mathcal{S}^*$. |
| $\overline{\mathcal{S}}$ | Second-order one-step estimator of $\mathcal{S}^*$. |
| $\xi_\mu$ | Self-transport kernel (density of $\pi^{(\mu,\mu)}$ w.r.t. $\mu \otimes \mu$). |

*Continued on next page*

| Symbol | Description |
|--------|-------------|
| $T_\mu$ | $(T_\mu f)(\cdot) = \int \xi_\mu(\cdot, y) f(y) \, d\mu(y).$ |
| $H_\mu$ | $(H_\mu \gamma)(\cdot) = \int \xi_\mu(\cdot, y) \, d\gamma(y).$ |
| $C(\mathcal{Y})/\mathbb{R}$ | Continuous functions modulo additive constants; $[f]$ denotes the equivalence class. |
| $K_\mu$ | Hadamard operator: $K_\mu = \varepsilon (I - T_\mu^2)^{-1} H_\mu.$ |
| $k_\mu$ | Kernel representing the bilinear form induced by $K_\mu$ on $\mathcal{M}_0(\mathcal{Y}).$ |

## B. Background

### B.1. Pathwise differentiability and EIF

We refer the reader to Luedtke & Chung (2024) for a detailed exposition of pathwise differentiability. We briefly recall the key definitions needed for our analysis. As introduced in Sec. 2, let $\mathcal{P} \subset \mathcal{P}(\mathcal{Z})$ denote our statistical model, where $\mathcal{Z} := \mathcal{X} \times \{0, 1\} \times \mathcal{Y}$ is a Polish space equipped with its Borel $\sigma$-algebra $\mathcal{B}_{\mathcal{Z}}$. We assume that all distributions in $\mathcal{P}$ are dominated by a $\sigma$-finite measure $\lambda$.

A parametric submodel $(P_t : t \in [0, \delta)) \subset \mathcal{P}$ with $P_0 = P$ is said to be *quadratic mean differentiable* (QMD) at $P$ if there exists a (Fisher) score function $s \in L_0^2(P)$ such that

$$\left\| \sqrt{p_t} - \sqrt{p} - ts\sqrt{p} \right\|_{L^2(\lambda)} = o(t),$$

where $p_t = dP_t/d\lambda$ and $p = dP/d\lambda$. We denote by $\mathfrak{P}(P, \mathcal{P}, s)$ the collection of all QMD submodels at $P$ with score function $s$. The set $\{s \in L_0^2(P) : \mathfrak{P}(P, \mathcal{P}, s) \neq \emptyset\}$ is called the *tangent set* of $\mathcal{P}$ at $P$. Its closed linear span in $L_0^2(P)$ is the *tangent space*, denoted by $\dot{\mathcal{P}}_P$.

Let $\Phi : \mathcal{P} \to \mathcal{H}$ be a Hilbert-valued parameter. We say that $\Phi$ is *pathwise differentiable* at $P$, relative to the model $\mathcal{P}$, if there exists a continuous linear operator $D\Phi_P : \dot{\mathcal{P}}_P \to \mathcal{H}$ such that, for all $(P_t : t \in [0, \delta)) \in \mathfrak{P}(P, \mathcal{P}, s)$,

$$\left\| \Phi(P_t) - \Phi(P) - tD\Phi_P(s) \right\|_{\mathcal{H}} = o(t). \tag{16}$$

The operator $D\Phi_P$ is called the *local parameter* of $\Phi$ at $P$. Its image is a closed subspace of $\mathcal{H}$, denoted by $\dot{\mathcal{H}}_P$, and referred to as the local parameter space. The Hermitian adjoint of $D\Phi_P$, denoted by $D^*\Phi_P : \mathcal{H} \to \dot{\mathcal{P}}_P$, is called the *efficient influence operator*. The operators $D\Phi_P$ and $D^*\Phi_P$ satisfy the adjoint relationship $\langle h, D\Phi_P(s) \rangle_{\mathcal{H}} = \langle D^*\Phi_P(h), s \rangle_{L^2(P)}$ for all $h \in \mathcal{H}$, $s \in \dot{\mathcal{P}}_P$. We say that $\Phi$ admits an efficient influence function (EIF) at $P$ if, for $P$-almost every $z \in \mathcal{Z}$, the map $h \mapsto D^*\Phi_P(h)(z)$ defines a bounded linear functional on $\mathcal{H}$. In this case, by the Riesz representation theorem, there exists a function $\phi_P : \mathcal{Z} \to \mathcal{H}$ such that

$$D^*\Phi_P(h)(z) = \langle h, \phi_P(z) \rangle_{\mathcal{H}}, \quad \text{for all } h \in \mathcal{H} \text{ and } P\text{-a.e. } z. \tag{17}$$

Sufficient conditions for the existence of the EIF when $\mathcal{H}$ and $\dot{\mathcal{H}}_P$ are RKHS are given in (Luedtke & Chung, 2024, Theorem 1). In the special case of real-valued parameters, $\mathcal{H} = \mathbb{R}$, a function $\phi_P : \mathcal{Z} \to \mathbb{R}$ is an influence function of $\Phi$ at $P$ if the local parameter admits the representation

$$D\Phi_P(s) = \langle \phi_P, s \rangle_{L^2(P)}.$$

The *efficient* influence function is the unique influence function that lies in the tangent space $\dot{\mathcal{P}}_P$, and can therefore be obtained as the orthogonal projection of any influence function onto $\dot{\mathcal{P}}_P$. For fully nonparametric models, $\dot{\mathcal{P}}_P = L_0^2(P)$, so

the EIF reduces to $\phi_P - P\phi_P$ (Bickel et al., 1993).

**Second-order scores and influence functions** Again, we consider a smooth one-dimensional parametric submodel $(P_t : t \in [0, \delta)) \subset \mathcal{P}$ through $P_0 = P$, with corresponding densities $p_t$ with respect to a common dominating measure. For a general treatment of higher-order scores and influence functions in $r$-dimensional submodels ($r \geq 1$), see Robins et al. (2008, Sec. 2).

Let $s : \mathcal{Z} \to \mathbb{R}$ denote the first-order score function $s(z) = \frac{\partial t}{\partial t} p_t(z)\big|_{t=0}$. The second-order score function (Waterman & Lindsay, 1996; Small & McLeish, 2011; van der Vaart, 2014) $r : \mathcal{Z} \times \mathcal{Z} \to \mathbb{R}$ is defined through the perturbation of the product measure $P_t^{\otimes 2} = P_t \otimes P_t$. Specifically, it is the second derivative of the joint density relative to the baseline product measure $P^2$. A direct calculation is presented below

$$r(z_1, z_2) = \frac{1}{2p(z_1)p(z_2)} \left( \frac{d^2}{dt^2} (p_t(z_1)p_t(z_2)) \right) \bigg|_{t=0}$$
$$= \frac{1}{2p(z_1)} \frac{d^2 p_t(z_1)}{dt^2} \bigg|_{t=0} + \frac{1}{2p(z_2)} \frac{d^2 p_t(z_2)}{dt^2} \bigg|_{t=0} + s(z_1)s(z_2).$$

Thus, unlike the first-order score, the second-order score is not simply the second derivative of the univariate log-likelihood; rather, it arises from the second-order perturbation of the product measure. In particular, for a linear path $P_t = (1 + ts)P$, this reduces to the product of scores: $r(z_1, z_2) = s(z_1)s(z_2)$.

The second-order influence function of $\Phi$ at $P$ is a symmetric measurable map $\ddot{\Phi}_P \in L_0^2(P^{\otimes 2})$ such that $\mathbb{E}_P[\ddot{\Phi}_P] = 0$ and the following holds

$$\frac{d}{dt} \Phi(P_t)\bigg|_{t=0} = \iint \ddot{\Phi}_P(z_1, z_2)(s(z_1) + s(z_2)) \, dP(z_1) \, dP(z_2), \quad \text{and}$$
$$\frac{d^2}{dt^2} \Phi(P_t)\bigg|_{t=0} = \iint \ddot{\Phi}_P(z_1, z_2) \, r(z_1, z_2) \, dP(z_1) \, dP(z_2). \tag{18}$$

Our presentation of second-order pathwise differentiability differs slightly from that of Robins et al. (2008), in that we carry an extra factor of $1/2$ in our expression for $r$ and $2$ in our expression for the second-order part of $\ddot{\Phi}_P$. Concretely, in the notation of Robins et al. (2008), the second-order influence function of $\Phi$ at $P$ would be $\Gamma(\ddot{\Phi}_P)$, where $\Gamma : L_0^2(P^{\otimes 2}) \to L_0^2(P^{\otimes 2})$ is the bijective map defined by

$$\Gamma(f) : (z_1, z_2) \mapsto \int [f(z_1, z) + f(z, z_2)] dP(z) + \frac{1}{2} \left[ f(z, z_2) - \int [f(z_1, z) + f(z, z_2)] dP(z) \right].$$

In the case where the first-order EIF is 0 a.s., the first-order part of $\ddot{\Phi}_P$ is also 0, and so our second-order EIF is twice that from Robins et al. (2008, Def. 2.1), that is, $\ddot{\Phi}_P = 2\Gamma(\ddot{\Phi}_P)$.

### B.2. Entropic Optimal Transport

We briefly review entropic optimal transport. We refer to Léonard (2013) and Cuturi (2013) for comprehensive treatments. Throughout, we work on the space of Borel probability measures $\mathcal{P}(\mathcal{Y})$ defined on the Polish space $(\mathcal{Y}, d_{\mathcal{Y}})$.

Given $\mu, \nu \in \mathcal{P}(\mathcal{Y})$, the Kantorovich formulation of the optimal transport problem (Kantorovich, 2006; Villani, 2009) with quadratic cost is

$$\text{OT}(\mu, \nu) = \inf_{\pi \in \Pi(\mu,\nu)} \int \frac{\|y_1 - y_2\|^2}{2} \, d\pi(y_1, y_2), \tag{19}$$

where $\Pi(\mu, \nu)$ denotes the set of couplings of $\mu$ and $\nu$, that is joint distributions on $\mathcal{Y} \times \mathcal{Y}$ with marginals $\mu$ and $\nu$. Despite its appealing geometric properties, classical OT poses significant computational and statistical challenges. In particular, the

empirical plug-in estimator converges at the slow rate $n^{-1/d}$ in dimension $d$, reflecting the curse of dimensionality (Fournier & Guillin, 2015; Manole & Niles-Weed, 2024). Entropic optimal transport addresses these difficulties by introducing a Kullback-Leibler (KL) divergence-based regularization term. For $\varepsilon > 0$ regularization, the EOT problem is defined as

$$\mathrm{OT}_\varepsilon (\mu, \nu) = \inf_{\pi \in \Pi(\mu, \nu)} \int \frac{\|y_1 - y_2\|^2}{2} d\pi(y_1, y_2) + \varepsilon \mathrm{KL}(\pi \mid \mu \otimes \nu), \tag{20}$$

where $\mathrm{KL}(\cdot \mid \cdot)$ denotes the KL divergence. The regularized problem admits efficient solutions via the Sinkhorn algorithm, whose computational complexity scales quadratically in the sample size (Cuturi, 2013). The EOT problem enjoys strong duality, with dual formulation

$$\mathrm{OT}_\varepsilon (\mu, \nu) = \sup_{(\varphi_1, \varphi_0) \in L^1(\mu) \times L^1(\nu)} \left[ \int \varphi_1 \, d\mu + \int \varphi_0 \, d\nu - \varepsilon \int \exp\left( \frac{\varphi_1 \oplus \varphi_0 - \| \cdot - \cdot \|^2/2}{\varepsilon} \right) d(\mu \otimes \nu) + \varepsilon \right], \quad \tag{21}$$

where $(\varphi_1 \oplus \varphi_0)(y, y') = \varphi_1(y) + \varphi_0(y')$. The dual maximizers $(\varphi_1^{(\mu,\nu)}, \varphi_0^{(\mu,\nu)})$, known as entropic potentials, satisfy the Schrödinger system

$$\varphi_1^{(\mu,\nu)} = -\varepsilon \log \left( \int \exp\left( \frac{1}{\varepsilon} \left( \varphi_0^{(\mu,\nu)}(y') - \| \cdot - y' \|^2/2 \right) d\nu(y') \right) \right),$$

$$\varphi_0^{(\mu,\nu)} = -\varepsilon \log \left( \int \exp\left( \frac{1}{\varepsilon} \left( \varphi_1^{(\mu,\nu)}(y) - \| y - \cdot \|^2/2 \right) d\mu(y) \right) \right). \tag{22}$$

Note that $(\varphi_1^{(\mu,\nu)}, \varphi_0^{(\mu,\nu)})$ are unique up to an additive constant, meaning that if $(\tilde{\varphi}_1^{(\mu,\nu)}, \tilde{\varphi}_2^{(\mu,\nu)})$ is another pair of entropic potentials satisfying (22), there exists a constant $c > 0$ such that $(\tilde{\varphi}_1^{(\mu,\nu)}, \tilde{\varphi}_2^{(\mu,\nu)}) \equiv (\varphi_1^{(\mu,\nu)} + c, \varphi_0^{(\mu,\nu)} - c)$. Following Luise et al. (2019), we define unique potentials $(\varphi_1^{(\mu,\nu)}, \varphi_0^{(\mu,\nu)})$ such that—at an arbitrarily chosen reference point $y_0 \in \mathcal{Y}$— $\varphi_1^{(\mu,\nu)}(y_0) = 0$.

A limitation of EOT is that $\mathrm{OT}_\varepsilon (\mu, \mu) \neq 0$, which precludes its direct use as a discrepancy measure. To remove this bias, Ramdas et al. (2017) and Feydy et al. (2019) introduced the Sinkhorn divergence

$$S_\varepsilon (\mu, \nu) = \mathrm{OT}_\varepsilon (\mu, \nu) - \frac{1}{2} \mathrm{OT}_\varepsilon (\mu, \mu) - \frac{\mathbb{1}}{2} \mathrm{OT}_\varepsilon (\nu, \nu).$$

## C. Hadamard Differentiability of EOT

### C.1. Setup

In this section, we present the results on the Hadamard differentiability of EOT objects, specifically, the entropic potentials and the EOT cost, when viewed as mappings on the Banach space $\ell^\infty(\mathcal{H}_1) \times \ell^\infty(\mathcal{H}_1)$. As discussed in Sec. 3, the map $J$ allows us to identify each probability measure in $\mathcal{P}(\mathcal{Y}) \times \mathcal{P}(\mathcal{Y})$ with a unique element of $\ell^\infty(\mathcal{H}_1) \times \ell^\infty(\mathcal{H}_1)$. Working with this representation, we equip the space of measures with the product norm

$$\|(\gamma^1, \gamma^2)\|_\infty := \|\gamma^1\|_{\ell^\infty(\mathcal{H}_1)} \vee \|\gamma^2\|_{\ell^\infty(\mathcal{H}_1)}, \quad (\gamma^1, \gamma^2) \in \ell^\infty(\mathcal{H}_1) \times \ell^\infty(\mathcal{H}_1),$$

Related differentiability results have been established in closely related normed spaces. In particular, on $\ell^\infty(B^s) \times \ell^\infty(B^s)$, where $B^s = \{f \in \mathcal{C}^s(\mathcal{Y}) : \|f\|_{\mathcal{C}^s(\mathcal{Y})} \leq 1\}$ denotes the unit ball of the $s$-Hölder space $\mathcal{C}^s(\mathcal{Y})$, Goldfeld et al. (2024) proved first- and second-order Hadamard differentiability for the entropic potentials and the Sinkhorn divergence. More recently, Kokot & Luedtke (2025) derived an explicit expression for the first-order Hadamard derivative of the entropic potentials and for the second-order derivative of the Sinkhorn divergence under the null in the Gaussian RKHS topology. In this section, we extend these Hadamard differentiability results for the relevant EOT objects to the Sobolev RKHS topology.

Let $\mu \in \mathcal{P}(\mathcal{Y})$ be a reference measure, and define $\mathcal{P}_\mu$ at $\mu$ by

$$\mathcal{P}_\mu = \{\nu \in \mathcal{P}(\mathcal{Y}) : \operatorname{supp}(\nu) \subset \operatorname{supp}(\mu)\}.$$

We define the tangent cone to $\mathcal{P}_\mu$ at $\mu$ by

$$\mathcal{M}_{0,\mu} = \overline{\{t(\nu - \mu) : \nu \in \mathcal{P}_\mu, t > 0\}}^{\ell^\infty(\mathcal{H}_1)}.$$

Tangent cone represents the collection of all admissible first-order perturbation directions at $\mu$ induced by paths remaining in $\mathcal{P}_\mu$, with the closure taken in $\ell^\infty(\mathcal{H}_1)$ topology. Hadamard differentiability at $\mu$ is therefore understood with respect to perturbations along paths $(\mu_t : t \in \mathbb{R}) \subset \mathcal{P}_\mu$, whose first-order increments lie in $\mathcal{M}_{0,\mu}$.

### C.2. Sinkhorn Hadamard operator

Fix $\mu \in \mathcal{P}(\mathcal{Y})$ and recall the operators $H_\mu$ and $T_\mu$ defined in (9). Since $T_\mu \mathbb{1}_\mathcal{Y} = \mathbb{1}_\mathcal{Y}$, the operator $T_\mu$ induces a well-defined operator on the quotient space $\mathcal{C}(\mathcal{Y})/\mathbb{R}$, still denoted by $T_\mu : \mathcal{C}(\mathcal{Y})/\mathbb{R} \to \mathcal{C}(\mathcal{Y})/\mathbb{R}$ for convenience. Because $\mathcal{M}_0(\mathcal{Y})$ is canonically dual to $\mathcal{C}(\mathcal{Y})/\mathbb{R}$, the Sinkhorn operator is defined by

$$K_\mu : \mathcal{M}_0(\mathcal{Y}) \to \mathcal{C}(\mathcal{Y})/\mathbb{R}, \quad K_\mu \gamma = \varepsilon (I - T_\mu^2)^{-1}[H_\mu \gamma].$$

This expression is well-defined because, by Lavenant et al. (2024, Thm. 3.8), the inverse $(I - T_\mu^2)^{-1}$ exists as a bounded operator on $\mathcal{C}(\mathcal{Y})/\mathbb{R}$. Consequently, for any $\gamma_1, \gamma_2 \in \mathcal{M}_0(\mathcal{Y})$, the evaluation of the Sinkhorn Hadamard operator $B_\mu(\gamma_1, \gamma_2) := \langle \gamma_1, K_\mu \gamma_2 \rangle$ is well-defined by duality between $\mathcal{M}_0(\mathcal{Y})$ and $\mathcal{C}(\mathcal{Y})/\mathbb{R}$.

Moreover, the quotient space $\mathcal{C}(\mathcal{Y})/\mathbb{R}$ is isometrically isomorphic to the closed subspace of $\mathcal{C}(\mathcal{Y})$ consisting of functions orthogonal to constants (equivalently, functions with zero constant component). For any $\gamma \in \mathcal{M}_0(\mathcal{Y})$, the function $H_\mu \gamma$ has zero constant component, hence belongs to this subspace and canonically represents its equivalence class $[H_\mu \gamma] \in \mathcal{C}(\mathcal{Y})/\mathbb{R}$. Accordingly, when it will not cause confusion, we write $(I - T_\mu^2)^{-1} H_\mu \gamma$ instead of $(I - T_\mu^2)^{-1}[H_\mu \gamma]$.

Following Lavenant et al. (2024), we now refine this construction in the RKHS setting. Let $\mathcal{H}_\mu$ denote the RKHS associated with the self-transport kernel $\xi_\mu$. The RKHS is well-defined because $\xi_\mu$ is continuous, symmetric, and positive definite. By construction, both $H_\mu$ and $T_\mu$ map into $\mathcal{H}_\mu$. Since constant functions lie in $\mathcal{H}_\mu$ and satisfy $T_\mu \mathbb{1}_\mathcal{Y} = \mathbb{1}_\mathcal{Y}$, the operator $T_\mu$ preserves constants and therefore induces a well-defined operator on the quotient space $T_\mu : \mathcal{H}_\mu/\mathbb{R} \to \mathcal{H}_\mu/\mathbb{R}$. Similarly, because $H_\mu \gamma : \mathcal{M}(\mathcal{Y}) \to \mathcal{H}_\mu$ is injective and has no constant component when $\gamma \in \mathcal{M}_0(\mathcal{Y})$, the map $[H_\mu(\cdot)] : \mathcal{M}_0(\mathcal{Y}) \to \mathcal{H}_\mu/\mathbb{R}$ is also injective; for brevity, we will simply denote this map as $H_\mu : \mathcal{M}_0(\mathcal{Y}) \to \mathcal{H}_\mu/\mathbb{R}$ when clear from context. The quotient space $\mathcal{H}_\mu/\mathbb{R}$, equipped with the norm $\|[h]\|_{\mathcal{H}_\mu/\mathbb{R}} = \inf_{\lambda \in \mathbb{R}} \|h - \lambda \mathbb{1}_\mathcal{Y}\|_{\mathcal{H}_\mu}$, is isometrically isomorphic to the closed subspace $\mathbb{1}_\mathcal{Y}^\perp := \{h \in \mathcal{H}_\mu : \langle h, \mathbb{1}_\mathcal{Y} \rangle_{\mathcal{H}_\mu} = 0\}$. Hence $\mathcal{H}_\mu/\mathbb{R}$ is itself a Hilbert space. The spectrum of $T_\mu : \mathcal{H}_\mu/\mathbb{R} \to \mathcal{H}_\mu/\mathbb{R}$ is contained in $[0, q]$ for some $q < 1$, and so the spectrum of $(I - T_\mu^2)^{-1}$ is contained in $[\varepsilon, \varepsilon/(1 - q^2)]$ (Lavenant et al., 2024, Thm. 4.3).

We now derive a kernel representation of $B_\mu(\gamma_1, \gamma_2)$ in terms of the inner product in the Hilbert space $\mathcal{H}_\mu/\mathbb{R}$.

**Lemma C.1.** *The kernel $k_\mu : \mathcal{Y} \times \mathcal{Y} \to \mathbb{R}$, defined so*

$$k_\mu(y_1, y_2) = \left\langle (I - T_\mu^2)^{-1/2}[\xi_\mu(y_1, \cdot)], (I - T_\mu^2)^{-1/2}[\xi_\mu(y_2, \cdot)] \right\rangle_{\mathcal{H}_\mu/\mathbb{R}},$$

*is symmetric positive semi-definite and, for any $\gamma_1, \gamma_2 \in \mathcal{M}_0(\mathcal{Y})$, satisfies*

$$B_\mu(\gamma_1, \gamma_2) = \int_\mathcal{Y} \int_\mathcal{Y} k_\mu(y, y') \, d\gamma_1(y) \, d\gamma_2(dy').$$

A kernel representation satisfying the above display is not unique; in particular, $B_\mu(\gamma_1, \gamma_2) = \int \int [k_\mu(y, y') + a(y) + b(y')] \, d\gamma_1(y) \, d\gamma_2(dy')$ for any measurable functions $a, b : \mathcal{Y} \to \mathbb{R}$.

*Proof.* With a slight abuse of notation, we will use $K_\mu \gamma_2$ to denote its representative element in $\mathbf{1}_\mathcal{Y}^\perp \subset \mathcal{H}_\mu$. Then, $B_\mu(\gamma_1, \gamma_2) = \int K_\mu \gamma_2 \, d\gamma_1$. Since $H_\mu \gamma_1$ is a kernel mean embedding, $\int K_\mu \gamma_2 \, d\gamma_1 = \langle H_\mu \gamma_1, K_\mu \gamma_2 \rangle_{\mathcal{H}_\mu}$. Because $H_\mu \gamma_1, K_\mu \gamma_2 \in \mathbf{1}_\mathcal{Y}^\perp$, we have that

$$B(\gamma_1, \gamma_2) = \langle H_\mu \gamma_1, K_\mu \gamma_2 \rangle_{\mathcal{H}_\mu} = \langle [H_\mu \gamma_1], [K_\mu \gamma_2] \rangle_{\mathcal{H}_\mu/\mathbb{R}}.$$

Since $(I - T_\mu^2)^{-1}$ is strictly positive, $(I - T_\mu^2)^{-1/2}$ exists on $\mathcal{H}_\mu/\mathbb{R}$. This and its self-adjointness (Lavenant et al., 2024, Prop. 4.2) give

$$B_\mu(\gamma_1, \gamma_2) = \left\langle [H_\mu \gamma_1], (I - T_\mu^2)^{-1}[H_\mu \gamma_2] \right\rangle_{\mathcal{H}_\mu/\mathbb{R}} = \left\langle (I - T_\mu^2)^{-1/2}[H_\mu \gamma_1], (I - T_\mu^2)^{-1/2}[H_\mu \gamma_2] \right\rangle_{\mathcal{H}_\mu/\mathbb{R}}.$$

Let $Q : \mathcal{H}_\mu \to \mathcal{H}_\mu/\mathbb{R}$ be the standard quotient map such that $Q : h \mapsto [h]$. Since $Q$ is a bounded linear map, it commutes with Bochner integration. That is for any Bochner integrable $\mathcal{H}_\mu$-valued function $h$, $Q(\int h \, d\gamma) = \int Q(h) \, d\gamma$. The boundedness of $\xi_\mu$ on compact $\mathcal{Y}$ implies Bochner integrability of $h = \xi_\mu$, and then we have that $[H_\mu \gamma_i] = [\int \xi_\mu(\cdot, y) \, d\gamma_i(y)] = \int [\xi_\mu(\cdot, y)] \, d\gamma_i(y)$ for $i \in \{1, 2\}$. Since $(I - T_\mu^2)^{-1}$ is a bounded linear operator on $\mathcal{H}_\mu/\mathbb{R}$ (Lavenant et al., 2024, Thm. 4.3), we can exchange the operator $(I - T_\mu^2)^{-1/2}$ and this gives

$$(I - T_\mu^2)^{-1/2}[H_\mu \gamma] = (I - T_\mu^2)^{-1/2} \int_\mathcal{Y} [\xi_\mu(\cdot, y)] \, d\gamma(y) = \int_\mathcal{Y} (I - T_\mu^2)^{-1/2}[\xi_\mu(\cdot, y)] \, d\gamma(y).$$

Using the above, and the bilinearity and continuity of inner products, we have

$$
\begin{aligned}
B_\mu(\gamma_1, \gamma_2) &= \left\langle \int_\mathcal{Y} (I - T_\mu^2)^{-1/2}[\xi_\mu(y_1, \cdot)] d\gamma_1(y_1), \int_\mathcal{Y} (I - T_\mu^2)^{-1/2}[\xi_\mu(y_2, \cdot)] d\gamma_2(y_2) \right\rangle_{\mathcal{H}_\mu/\mathbb{R}} \\
&= \int_\mathcal{Y} \int_\mathcal{Y} \left\langle (I - T_\mu^2)^{-1/2}[\xi_\mu(y_1, \cdot)], (I - T_\mu^2)^{-1/2}[\xi_\mu(y_2, \cdot)] \right\rangle_{\mathcal{H}_\mu/\mathbb{R}} d\gamma_1(y_1) \, d\gamma_2(y_2) \\
&=: \int_\mathcal{Y} \int_\mathcal{Y} k_\mu(y_1, y_2) \, d\gamma_1(y_1) \, d\gamma_2(y_2).
\end{aligned}
$$

Symmetry of $k_\mu$ follows by the symmetry of inner products. To see that the kernel $k_\mu$ is positive semi-definite, note that for every choice of points $y_1, \ldots, y_m \in \mathcal{Y}$ and vector $(c_1, \ldots, c_m) \in \mathbb{R}^m$,

$$
\begin{aligned}
\sum_{i,j} c_i c_j k_\mu(y_i, y_j) &= \sum_{i,j} c_i c_j \left\langle (I - T_\mu^2)^{-1/2}[\xi_\mu(y_i, \cdot)], (I - T_\mu^2)^{-1/2}[\xi_\mu(y_j, \cdot)] \right\rangle_{\mathcal{H}_\mu/\mathbb{R}} \\
&= \left\| \sum_i c_i (I - T_\mu^2)^{-1/2}[\xi_\mu(y_i, \cdot)] \right\|_{\mathcal{H}_\mu/\mathbb{R}}^2 \geq 0.
\end{aligned}
$$

$\square$

### C.3. Hadamard differentiability of EOT potentials and divergence

Now we are ready to present the Hadamard differentiability results of entropic potentials and Sinkhorn divergence in the $\ell^\infty(\mathcal{H}_1)$ topology. First, we characterize some properties of the entropic potentials without proof.

**Lemma C.2** (Properties of entropic potentials). *Recalling that the cost function $c$ is $C^\infty$ and $\mathcal{Y}$ is compact subset of $\mathbb{R}^d$, for every $(\mu, \nu) \in \mathcal{P}(\mathcal{Y}) \times \mathcal{P}(\mathcal{Y})$,*

1. *Genevay et al. (2019, Thm. 2): The entropic potentials $(\varphi_1^{(\mu,\nu)}, \varphi_0^{(\mu,\nu)})$ are uniformly bounded in $\mathcal{H}$, and their norms*

*satisfy* $\left\|\varphi_i^{(\mu,\nu)}\right\|_{\mathcal{H}} \leq \mathcal{O}\left(1+\varepsilon^{-s+1}\right)$ *for* $i \in \{1,0\}$*, and the constant depends on* $d$ *and* $|\mathcal{Y}|$*.*

2. *Goldfeld et al. (2024, Lem. 1): The map* $(\mu,\nu) \mapsto \left(\varphi_1^{(\mu,\nu)}, \varphi_0^{(\mu,\nu)}\right)$ *is continuous relative to the topology of weak convergence, that is if* $(\mu_n,\nu_n)$ *converges to* $(\mu,\nu)$ *weakly as* $n \to \infty$*, then* $\left(\varphi_1^{(\mu_n,\nu_n)}, \varphi_0^{(\mu_n,\nu_n)}\right) \to \left(\varphi_1^{(\mu,\nu)}, \varphi_0^{(\mu,\nu)}\right)$ *in* $\mathcal{H} \times \mathcal{H}$*.*

The second claim of Lem. C.2 follows directly from a similar continuity of entropic potentials in the $C^k(\mathcal{Y})$ norm for any arbitrary $k \in \mathbb{N}$ (Goldfeld et al., 2024, Lem. 1) because the Sobolev norm can be upperbounded by the $C^s(\mathcal{Y})$ norm for the compact domain.

**Lemma C.3** (Sobolev RKHS properties). *The following properties hold for Sobolev RKHS* $\mathcal{H} = H^s(\mathcal{Y})$ *for* $s > d/2$:

1. *For every smooth function* $F : \mathbb{R} \to \mathbb{R}$ *such that* $F(0) = 0$*, there exists a non-decreasing function* $C_F : \mathbb{R}^+ \to \mathbb{R}^+$ *such that*
$$\|F(h)\|_{\mathcal{H}} \leq C_F\left(\|h\|_{L^\infty(\mathcal{Y})}\right)\|h\|_{\mathcal{H}}.$$

2. *For all* $h_1, h_2 \in \mathcal{H}$*, the pointwise product* $h_1 h_2$ *belongs to* $\mathcal{H}$ *(defined a.e. in* $\mathcal{Y}$*). Further, there exists a constant* $K_2 \equiv K_2(s, d, |\mathcal{Y}|)$ *such that*
$$\|h_1 h_2\|_{\mathcal{H}} \leq K_2\|h_1\|_{\mathcal{H}}\|h_2\|_{\mathcal{H}}.$$

3. *For* $h \in \mathcal{H}$ *and* $\gamma \in \mathcal{M}(\mathcal{Y})$*, define* $T_\gamma h : y \mapsto \int_{\mathcal{Y}} e^{-\|y-y'\|^2/2\varepsilon} h(y')\, d\gamma(y')$*. Then for all* $h, \gamma$*, it holds that* $T_\gamma h \in \mathcal{H}$ *and moreover, there exists a constant* $K_3 \equiv K_3(s, d, |\mathcal{Y}|)$ *such that*

$$\|T_\gamma h\|_{\mathcal{H}} \leq K_3\|h\|_{\mathcal{H}}\|\gamma\|_{\ell^\infty(\mathcal{H})}.$$

*Proof.* The first result follows from Sobolev inequalities (Gagliardo–Nirenberg–Moser estimates) (Gagliardo, 1959; Nirenberg, 1959; Moser, 1966) for composition. Since $s > d/2$, $H^s(\mathcal{Y}) \subset \mathcal{C}(\mathcal{Y})$ (Taylor, 1996, Chap. 4, Prop. 1.3). Therefore, all $h \in \mathcal{H}$ are continuous and hence bounded on $\mathcal{Y}$, i.e. $h \in L^\infty(\mathcal{Y})$. For every smooth function $F : \mathbb{R} \to \mathbb{R}$ such that $F(0) = 0$, there exists a non-decreasing function $C_F : \mathbb{R}^+ \to \mathbb{R}^+$ such that $\|F(h)\|_{\mathcal{H}} \leq C_F(\|h\|_{L^\infty(\mathcal{Y})})\|h\|_{\mathcal{H}}$ (Taylor et al., 1996, Chap. 13, Prop. 3.9).

Because we chose $s > d/2$, $\mathcal{H}$ is a Banach algebra (Adams & Fournier, 2003, Thm. 4.39), which implies that for any $h_1, h_2 \in \mathcal{H}$, the pointwise product $h_1 h_2$ (defined a.e. in $\mathcal{Y}$) belongs to $\mathcal{H}$. Further, there exists a constant $K_2 \equiv K_2(s, d, \mathcal{Y})$, such that $\|h_1 h_2\|_{\mathcal{H}} \leq K_2\|h_1\|_{\mathcal{H}}\|h_2\|_{\mathcal{H}}$.

For the third result, the function $T_\gamma h$ is well-defined because $\mathcal{H} \hookrightarrow \mathcal{C}(\mathcal{Y})$, so $h$ has a continuous representative in $\mathcal{C}(\mathcal{Y})$. Let $k(y, y') = e^{-\frac{1}{2\varepsilon}\|y-y'\|^2}$. We have that

$$|T_\gamma h(y)| = \left|\int k(y, y')\, h(y')\, d\gamma(y')\right| \leq \|h\|_{L^\infty(\mathcal{Y})}\|\gamma\|_{\mathrm{TV}} < \infty.$$

Now we show that $T_\gamma h \in \mathcal{H}$. Let $k(y, y') = e^{-\frac{1}{2\varepsilon}\|y-y'\|^2}$. For every $y' \in \mathcal{Y}$, the function $y' \mapsto k(y, y')$ is $C^\infty$. For each $y \in \mathcal{Y}$, $T_\gamma h(y) = \gamma(k(y, \cdot)\, h)$. Since $y' \mapsto k(y, y') \in C^\infty(\mathcal{Y})$ and $\mathcal{H}$ is an algebra $k(y, \cdot)h$ belongs to $\mathcal{H}$. As a consequence,

$$|T_\gamma h(y)| \leq \|k(y, \cdot)h\|_{\mathcal{H}}\|\gamma\|_{\ell^\infty(\mathcal{H}_1)}$$
$$\leq K_2\|k(y, \cdot)\|_{\mathcal{H}}\|h\|_{\mathcal{H}}\|\gamma\|_{\ell^\infty(\mathcal{H}_1)}$$
$$\implies \|T_\gamma h\|_{L^\infty(\mathcal{Y})} \leq K_2 M_0\|h\|_{\mathcal{H}}\|\gamma\|_{\ell^\infty(\mathcal{H}_1)},$$

where $M_0 = \sup_y \|k(y, \cdot)\|_{\mathcal{H}} < \infty$ because $\mathcal{Y}$ is compact. Similarly, for any multi-index $\alpha$, define $M_\alpha = \sup_y \|\partial_y^\alpha k(y, \cdot)\|_{\mathcal{H}} < \infty$. Using the same argument and knowing that $\partial^\alpha T_\gamma h(y) = \gamma(\partial^\alpha k(y, \cdot)h)$, we have that

$\|\partial^\alpha T_\gamma h\|_{L^\infty(\mathcal{Y})} < \infty$. Then it holds that

$$
\begin{aligned}
\|T_\gamma h\|_{\mathcal{H}} &= \left( \sum_{|\alpha| \leq s} \|\partial^\alpha T_\gamma h\|_{L^2(\mathcal{Y})}^2 \right)^{1/2} \\
&\leq |\mathcal{Y}|^{1/2} \sum_{|\alpha| \leq s} \|\partial^\alpha T_\gamma h\|_{L^\infty(\mathcal{Y})} \\
&\leq K_2 |\mathcal{Y}|^{1/2} \|h\|_{\mathcal{H}} \|\gamma\|_{\ell^\infty(\mathcal{H}_1)} \sum_{|\alpha| \leq s} M_\alpha =: K_3 \|h\|_{\mathcal{H}} \|\gamma\|_{\ell^\infty(\mathcal{H}_1)}.
\end{aligned}
$$

The first inequality follows from elementary inequalities between $L^2$ and $L^\infty$ norms. The last inequality uses that $\mathcal{H}$ is an algebra from previous result. $\qquad\square$

Next, we prove the Hadamard differentiability of Sinkhorn divergence in the dual Sobolev RKHS topology. The Hadamard derivative is of the same form as presented in Goldfeld et al. (2024, Lem. 3), but in an ambient RKHS space ($\mathcal{H}$) rather than a dual Hölder space. We present a proof here for completeness.

**Lemma C.4** (Hadamard derivative of $S_\varepsilon$). *For a fixed $\varepsilon > 0$, the functional $S_\varepsilon : (\mu, \nu) \mapsto S_\varepsilon(\mu, \nu), \mathcal{P}(\mathcal{Y}) \times \mathcal{P}(\mathcal{Y}) \subset \ell^\infty(\mathcal{H}_1) \times \ell^\infty(\mathcal{H}_1) \to \mathbb{R}$ is Hadamard differentiable at $(\mu, \nu)$, tangentially to $\mathcal{M}_{0,\mu} \times \mathcal{M}_{0,\nu}$, with Hadamard derivative $S'_\varepsilon[\mu, \nu] : \mathcal{M}_{0,\mu} \times \mathcal{M}_{0,\nu} \to \mathbb{R}$ given by*

$$
S'_\varepsilon[\mu, \nu](\gamma^1, \gamma^2) = \int v_1^{(\mu,\nu)} \, d\gamma^1 + \int v_0^{(\mu,\nu)} \, d\gamma^2.
$$

*Proof.* By linearity of derivatives, it suffices to compute the derivative of $(\alpha, \beta) \mapsto \mathrm{OT}_\varepsilon(\alpha, \beta)$. Consider paths $(\mu_t)_{t>0}$ and $(\nu_t)_{t>0}$ in $\mathcal{P}_\mu$ and $\mathcal{P}_\nu$ respectively, such that $\mu_t = \mu + t\gamma_t^1$ and $\nu_t = \nu + t\gamma_t^2$, with $\gamma_t^i \to \gamma^i$ as $t \downarrow 0$ in $\ell^\infty(\mathcal{H}_1)$, for $i \in \{1, 2\}$ and some $(\gamma^1, \gamma^2) \in \mathcal{M}_{0,\mu} \times \mathcal{M}_{0,\nu}$. Then,

$$
\frac{1}{t} \left( \mathrm{OT}_\varepsilon(\mu_t, \nu_t) - \mathrm{OT}_\varepsilon(\mu, \nu) \right) = \frac{1}{t} \left[ \left( \mu_t \varphi_1^{(\mu_t,\nu_t)} + \nu_t \varphi_0^{(\mu_t,\nu_t)} \right) - \left( \mu \varphi_1^{(\mu,\nu)} + \nu \varphi_0^{(\mu,\nu)} \right) \right].
$$

For any $(\mu, \nu) \in \mathcal{P}(\mathcal{Y}) \times \mathcal{P}(\mathcal{Y})$, recall the density of the EOT plan

$$
\xi^{(\mu,\nu)}(y_1, y_2) = \exp\left( \frac{1}{\varepsilon} \left( \varphi_1^{(\mu,\nu)}(y_1) + \varphi_0^{(\mu,\nu)}(y_2) - \frac{\|y_1 - y_2\|^2}{2} \right) \right).
$$

From the optimality of $\left( \varphi_1^{(\mu_t,\nu_t)}, \varphi_0^{(\mu_t,\nu_t)} \right)$, we have that

$$
\begin{aligned}
\frac{1}{t} \left( \mathrm{OT}_\varepsilon(\mu_t, \nu_t) - \mathrm{OT}_\varepsilon(\mu, \nu) \right) &\geq \frac{1}{t} \left[ \left( \mu_t \varphi_1^{(\mu,\nu)} + \nu_t \varphi_0^{(\mu,\nu)} \right) - \left( \mu \varphi_1^{(\mu,\nu)} + \nu \varphi_0^{(\mu,\nu)} \right) \right] \\
&\quad - \frac{\varepsilon}{t} \left[ \int \int \xi^{(\mu,\nu)}(y_1, y_2) \, d\mu_t(y_1) \, d\nu_t(y_2) - 1 \right] \\
&= \gamma_t^1 \varphi_1^{(\mu,\nu)} + \gamma_t^2 \varphi_0^{(\mu,\nu)} - t\varepsilon \int \int \xi^{(\mu,\nu)}(y_1, y_2) \, d\gamma_t^1(y_1) \, d\gamma_t^2(y_2),
\end{aligned}
$$

where the last equality follows by expanding $\mu_t$ (and $\nu_t$) as $\mu + t\gamma_t^1$ (and $\nu + t\gamma_t^2$), and noting that $\mu\left( \xi^{(\mu,\nu)}(\cdot, y_2) \right) = 1$ and $\nu\left( \xi^{(\mu,\nu)}(y_1, \cdot) \right) = 1$ for all $y_1, y_2 \in \mathcal{Y}$. Consequently,

$$
\liminf_{t \downarrow 0} \frac{1}{t} \left( \mathrm{OT}_\varepsilon(\mu_t, \nu_t) - \mathrm{OT}_\varepsilon(\mu, \nu) \right) \geq \lim_{t \downarrow 0} \left( \gamma_t^1 \varphi_1^{(\mu,\nu)} + \gamma_t^2 \varphi_0^{(\mu,\nu)} - t\varepsilon \int \int \xi^{(\mu,\nu)}(y_1, y_2) \, d\gamma_t^1(y_1) \, d\gamma_t^2(y_2) \right).
$$

We will show that the limit on the RHS is equal to $\gamma^1 \varphi_1^{(\mu,\nu)} + \gamma^2 \varphi_0^{(\mu,\nu)}$. We already have that

$$\left| (\gamma_t^1 - \gamma^1)\,\varphi_1^{(\mu,\nu)} + (\gamma_t^2 - \gamma^2)\varphi_0^{(\mu,\nu)} \right| \leq \left\| \gamma_t^1 - \gamma^1 \right\|_{\ell^\infty(\mathcal{H}_1)} \left\| \varphi_1^{(\mu,\nu)} \right\|_{\mathcal{H}} + \left\| \gamma_t^2 - \gamma^2 \right\|_{\ell^\infty(\mathcal{H}_1)} \left\| \varphi_0^{(\mu,\nu)} \right\|_{\mathcal{H}} \xrightarrow{t \downarrow 0} 0$$

because $\gamma_t^i \to \gamma^i$ as $t \to 0$ in $\ell^\infty(\mathcal{H}_1)$ for $i \in \{1, 2\}$ and entropic potentials are uniformly bounded by the first part of Lem. C.2. Therefore, it suffices to show that

$$\int \int \xi^{(\mu,\nu)}(y_1, y_2)\, d\gamma_t^1(y_1)\, d\gamma_t^2(y_2) = \mathcal{O}(1) \quad \text{as } t \downarrow 0. \tag{23}$$

We show this by using three properties for the Sobolev RKHS $\mathcal{H}$ detailed in Lem. C.3. Choosing $F(h) = e^h - 1$ in the first property gives us that there exists a non-decreasing function $C_F : \mathbb{R}^+ \to \mathbb{R}^+$ such that for all $h \in \mathcal{H}$,

$$\|e^h - 1\|_{\mathcal{H}} \leq C_F(\|h\|_{L^\infty(\mathcal{Y})})\|h\|_{\mathcal{H}}.$$

Because the constant function $\mathbf{1} \in \mathcal{H}$, we have that

$$\|e^h\|_{\mathcal{H}} \leq \|e^h - 1\|_{\mathcal{H}} + \|\mathbf{1}\|_{\mathcal{H}} \leq C_F(\|h\|_{L^\infty(\mathcal{Y})})(\|h\|_{\mathcal{H}} + 1). \tag{24}$$

The Sobolev embedding theorem (Adams & Fournier, 2003, Thm. 4.12) gives that $\mathcal{H}$ embeds continuously in $L^\infty(\mathcal{Y})$. Further, Lem. C.2 gives that the functions of the form $h = \varphi_i^{(\mu,\nu)}$ are bounded in $\mathcal{H}$, uniformly over $i \in \{1, 0\}$ and $(\mu, \nu) \in \mathcal{P}(\mathcal{Y}) \times \mathcal{P}(\mathcal{Y})$; denote the bound by $M_1 \equiv M_1(\varepsilon, s, d, |\mathcal{Y}|)$. As a consequence, $h = \varphi_i^{(\mu,\nu)}$ are uniformly bounded in $L^\infty(\mathcal{Y})$ norm, denote the bound by $M_2 \equiv M_2(\varepsilon, s, d, |\mathcal{Y}|)$. Therefore, using (24), for all $i \in \{1, 0\}$ and $(\mu, \nu) \in \mathcal{P}(\mathcal{Y}) \times \mathcal{P}(\mathcal{Y})$,

$$\left\| e^{\frac{1}{\varepsilon} \varphi_i^{(\mu,\nu)}} \right\|_{\mathcal{H}} \leq C_F\left( \frac{M_2}{\varepsilon} \right)\left( \frac{M_1}{\varepsilon} + 1 \right) =: K_1(\varepsilon, s, d, |\mathcal{Y}|) \equiv K_1. \tag{25}$$

The second property from Lem. C.3 shows that $\mathcal{H}$ is a Banach algebra. Therefore, for every $h_1, h_2$, the pointwise product $h_1 h_2 \in \mathcal{H}$ (defined a.e. $\mathcal{Y}$), belongs to $\mathcal{H}$ and there exists a constant $K_2 \equiv K_2(s, d, |\mathcal{Y}|)$ such that

$$\|h_1 h_2\|_{\mathcal{H}} \leq K_2 \|h_1\|_{\mathcal{H}} \|h_2\|_{\mathcal{H}}. \tag{26}$$

Let $k_\varepsilon(y, y') = e^{-\frac{1}{2\varepsilon}\|y - y'\|^2}$, and define $G_\gamma h : y \mapsto \int_{\mathcal{Y}} k_\varepsilon(y, y')\, h(y')\, d\gamma(y')$. The third property from Lem. C.3 shows that for all $h \in \mathcal{H}$ and $\gamma \in \mathcal{M}(\mathcal{Y})$, $G_\gamma h \in \mathcal{H}$, and there exists a constant $K_3 \equiv K_3(s, d, |\mathcal{Y}|)$ such that

$$\|G_\gamma h\|_{\mathcal{H}} \leq K_3 \|h\|_{\mathcal{H}} \|\gamma\|_{\ell^\infty(\mathcal{H}_1)}. \tag{27}$$

The three bounds (24), (26), and (27) allow us to establish the following string of inequalities

$$\left| \int \int \xi^{(\mu,\nu)}(y_1, y_2)\, d\gamma_t^1(y_1)\, d\gamma_t^2(y_2) \right| = \left| \left\langle \gamma_t^2,\, e^{\frac{1}{\varepsilon}\varphi_0^{(\mu,\nu)}} T_{\gamma_t^1} e^{\frac{1}{\varepsilon}\varphi_1^{(\mu,\nu)}} \right\rangle \right|$$

$$\leq \left\| \gamma_t^2 \right\|_{\ell^\infty(\mathcal{H}_1)} \left\| e^{\frac{1}{\varepsilon}\varphi_0^{(\mu,\nu)}} T_{\gamma_t^1} e^{\frac{1}{\varepsilon}\varphi_1^{(\mu,\nu)}} \right\|_{\mathcal{H}}$$

$$\leq K_2 \left\| \gamma_t^2 \right\|_{\ell^\infty(\mathcal{H}_1)} \left\| e^{\frac{1}{\varepsilon}\varphi_0^{(\mu,\nu)}} \right\|_{\mathcal{H}} \left\| T_{\gamma_t^1} e^{\varphi_1^{(\mu,\nu)}/\varepsilon} \right\|_{\mathcal{H}}$$

$$\leq K_2 K_3 \left\| e^{\frac{1}{\varepsilon}\varphi_0^{(\mu,\nu)}} \right\|_{\mathcal{H}} \left\| e^{\frac{1}{\varepsilon}\varphi_1^{(\mu,\nu)}} \right\|_{\mathcal{H}} \left\| \gamma_t^1 \right\|_{\ell^\infty(\mathcal{H}_1)} \left\| \gamma_t^2 \right\|_{\ell^\infty(\mathcal{H}_1)}$$

$$\leq K_1^2 K_2 K_3 \left\| \gamma_t^1 \right\|_{\ell^\infty(\mathcal{H}_1)} \left\| \gamma_t^2 \right\|_{\ell^\infty(\mathcal{H}_1)}. \tag{28}$$

The first equality follows by the fact that $e^{\frac{1}{\varepsilon}\varphi_0^{(\mu,\nu)}}, T_{\gamma_t^1} e^{\frac{1}{\varepsilon}\varphi_1^{(\mu,\nu)}} \in \mathcal{H}$, and $\mathcal{H}$ is a Banach algebra and therefore, their product

also belongs to $\mathcal{H}$. The second, third, and last inequalities follow from (26), (27), and (25) respectively. Because $\gamma_t^i \to \gamma^i$ in $\ell^\infty(\mathcal{H}_1)$, there exists a $t_0 > 0$ such that $\sup_{t \leq t_0} \|\gamma_t^i\|_{\ell^\infty(\mathcal{H}_1)} < \infty$ for $i \in \{1, 2\}$. And therefore, Equation (23) holds. Following this, we have that

$$\gamma^1 \varphi_1^{(\mu,\nu)} + \gamma^2 \varphi_0^{(\mu,\nu)} \leq \liminf_{t \downarrow 0} \frac{1}{t} \left( \mathrm{OT}_\varepsilon \left( \mu_t, \nu_t \right) - \mathrm{OT}_\varepsilon \left( \mu, \nu \right) \right). \tag{29}$$

Now by optimality of $(\varphi_1^{(\mu,\nu)}, \varphi_0^{(\mu,\nu)})$, we have that

$$
\begin{aligned}
\frac{1}{t} \left( \mathrm{OT}_\varepsilon \left( \mu_t, \nu_t \right) - \mathrm{OT}_\varepsilon \left( \mu, \nu \right) \right) &\leq \frac{1}{t} \left[ \left( \mu_t \varphi_1^{(\mu_t,\nu_t)} + \nu_t \varphi_0^{(\mu_t,\nu_t)} \right) - \left( \mu \varphi_1^{(\mu_t,\nu_t)} + \nu \varphi_0^{(\mu_t,\nu_t)} \right) \right] \\
&\quad + \frac{\varepsilon}{t} \left[ \int \int \xi^{(\mu_t,\nu_t)}(y_1, y_2) \, d\mu(y_1) \, d\nu(y_2) - 1 \right] \\
&= \gamma_t^1 \, \varphi_1^{(\mu_t,\nu_t)} + \gamma_t^2 \varphi_0^{(\mu_t,\nu_t)} + t\varepsilon \int \int \xi^{(\mu_t,\nu_t)}(y_1, y_2) \, d\gamma_t^1(y_1) \, d\gamma_t^2(y_2),
\end{aligned}
\tag{30}
$$

where the equality follows because $\mu_t \left( \xi^{(\mu_t,\nu_t)}(\cdot, y_2) \right) = 1$ and $\nu_t \left( \xi^{(\mu_t,\nu_t)}(y_1, \cdot) \right) = 1$ for all $y_1, y_2 \in \mathcal{Y}$. Consequently,

$$
\begin{aligned}
\limsup_{t \downarrow 0} \frac{1}{t} \left( \mathrm{OT}_\varepsilon \left( \mu_t, \nu_t \right) - \mathrm{OT}_\varepsilon \left( \mu, \nu \right) \right) &\leq \lim_{t \downarrow 0} \left( \gamma_t^1 \, \varphi_1^{(\mu_t,\nu_t)} + \gamma_t^2 \varphi_0^{(\mu_t,\nu_t)} + t\varepsilon \int \int \xi^{(\mu_t,\nu_t)}(y_1, y_2) \, d\gamma_t^1(y_1) \, d\gamma_t^2(y_2) \right) \\
&= \lim_{t \downarrow 0} \left( \gamma_t^1 \, \varphi_1^{(\mu_t,\nu_t)} + \gamma_t^2 \varphi_0^{(\mu_t,\nu_t)} \right).
\end{aligned}
$$

The equality follows again from Equation (28) because the upper bound established there is uniform over all $(\mu, \nu)$. For any sequence $(f_t)_{t>0} \subset \mathcal{H}$ and $(\eta_t, t > 0) \subset \mathcal{M}_0(\mathcal{Y})$, such that $f_t \to f$ in $\mathcal{H}$ and $\eta_t \to \eta$ in $\ell^\infty(\mathcal{H}_1)$, we have that $\eta_t f_t \to \eta f$ as $t \downarrow 0$. This hold because

$$|\eta_t f_t - \eta f| \leq |(\eta_t - \eta) f_t| + |\eta (f_t - f)| \leq \|\eta_t - \eta\|_{\ell^\infty(\mathcal{H}_1)} \|f_t\|_{\mathcal{H}} + \|\eta\|_{\ell^\infty(\mathcal{H}_1)} \|f_t - f\|_{\mathcal{H}}.$$

Since $f_t \to f$ in $\mathcal{H}$, for every $\epsilon > 0$, there exists a $t_0 > 0$ such that $\|f_t\|_{\mathcal{H}} < \epsilon$ for all $t < t_0$. Further, $\lim_{t \downarrow 0} \|f_t - f\|_{\mathcal{H}} = 0$ and $\lim_{t \downarrow 0} \|\eta_t - \eta\|_{\ell^\infty(\mathcal{H}_1)} = 0$, which proves the claim. Therefore,

$$\limsup_{t \downarrow 0} \frac{1}{t} \left( \mathrm{OT}_\varepsilon \left( \mu_t, \nu_t \right) - \mathrm{OT}_\varepsilon \left( \mu, \nu \right) \right) \leq \gamma^1 \varphi_1^{(\mu,\nu)} + \gamma^2 \varphi_0^{(\mu,\nu)}. \tag{31}$$

Combining (29) and (31), we prove that

$$\lim_{t \downarrow 0} \frac{1}{t} \left( \mathrm{OT}_\varepsilon \left( \mu_t, \nu_t \right) - \mathrm{OT}_\varepsilon \left( \mu, \nu \right) \right) = \gamma^1 \varphi_1^{(\mu,\nu)} + \gamma^2 \varphi_0^{(\mu,\nu)}.$$

$\square$

Now we focus on the first-order Hadamard differentiability of the function $(\mu, \nu) \mapsto (\varphi_1^{(\mu,\nu)}, \varphi_0^{(\mu,\nu)}), \mathcal{P}(\mathcal{Y}) \times \mathcal{P}(\mathcal{Y}) \subset \ell^\infty(\mathcal{H}_1) \times \ell^\infty(\mathcal{H}_1) \to \mathcal{H} \times \mathcal{H}$.

**Lemma C.5** (Hadamard differentiability of entropic potentials in $\mathcal{C}(\mathcal{Y})$). *The $\mathcal{P}(\mathcal{Y}) \times \mathcal{P}(\mathcal{Y}) \to \mathcal{C}(\mathcal{Y}) \times \mathcal{C}(\mathcal{Y})$ map* $(\mu, \nu) \mapsto (\varphi_1^{(\mu,\nu)}, \varphi_0^{(\mu,\nu)})$ *is Hadamard differentiable at* $(\mu, \nu)$*, tangentially to* $\mathcal{M}_{0,\mu} \times \mathcal{M}_{0,\nu}$*.*

*Proof.* The Hadamard differentiable of the map $\Phi : (\mu, \nu) \mapsto (\varphi_1^{(\mu,\nu)}, \varphi_0^{(\mu,\nu)})$ in $\mathcal{C}(\mathcal{Y}) \times \mathcal{C}(\mathcal{Y})$ has been established in the dual Hölder space topology by Goldfeld et al. (2024, Thm. 3), and in the Gaussian RKHS topology by Kokot & Luedtke (2025, Cor. 43), with the latter deriving explicit derivatives. Extending the differentiability in the Sobolev RKHS topology for measures follows by continuously embedding the Gaussian RKHS in the Sobolev RKHS. Let $\mathcal{G}$ be the Gaussian RKHS

with kernel $(y_1, y_2) \mapsto \exp(-\|y_1 - y_2\|^2 / 2\varepsilon)$, with norm $\|\cdot\|_{\mathcal{G}}$ and unit ball denoted by $\mathcal{G}_1$. The Hadamard differentiability of $\Phi$ as a map $\mathcal{P}(\mathcal{Y}) \times \mathcal{P}(\mathcal{Y}) \subset \ell^\infty(\mathcal{G}_1) \times \ell^\infty(\mathcal{G}_1) \to \mathcal{C}(\mathcal{Y}) \times \mathcal{C}(\mathcal{Y})$, is established in Kokot & Luedtke (2025, Cor. 43). We show that the same map is Hadamard differentiable when the domain is instead equipped with the stronger topology induced by $\ell^\infty(\mathcal{H}_1) \times \ell^\infty(\mathcal{H}_1)$.

By the continuous embedding of the Gaussian RKHS into the Sobolev RKHS, there exists a constant $C_s > 0$ such that for any $g \in \mathcal{G}$, $\|g\|_{\mathcal{H}} \le C_s \|g\|_{\mathcal{G}}$. Now we dualize this inclusion. If $g \in \mathcal{G}_1$, then $C_s^{-1} g \in \mathcal{H}_1$. Therefore, for every signed measure $\gamma$,

$$\|\gamma\|_{\ell^\infty(\mathcal{G}_1)} = \sup_{g \in \mathcal{G}_1} \left| \int g \, d\gamma \right| \le C_s \sup_{h \in \mathcal{H}_1} \left| \int h \, d\gamma \right| = C_s \, \|\gamma\|_{\ell^\infty(\mathcal{H}_1)}.$$

Therefore, the identity map from $\ell^\infty(\mathcal{H}_1) \to \ell^\infty(\mathcal{G}_1)$ is continuous. Since $\Phi$ is Hadamard differentiable at $(\mu, \nu)$ as a map defined on $\mathcal{P}(\mathcal{Y}) \times \mathcal{P}(\mathcal{Y}) \subset \ell^\infty(\mathcal{G}_1) \times \ell^\infty(\mathcal{G}_1)$, the defining limit remains valid for any perturbation sequences converging in the stronger topology $\ell^\infty(\mathcal{H}_1) \times \ell^\infty(\mathcal{H}_1)$. So the Hadamard expansion proved in Kokot & Luedtke (2025, Cor. 43) applies without change.

Therefore, $\Phi$ as a map $\mathcal{P}(\mathcal{Y}) \times \mathcal{P}(\mathcal{Y}) \subset \ell^\infty(\mathcal{H}_1) \times \ell^\infty(\mathcal{H}_1) \to \mathcal{C}(\mathcal{Y}) \times \mathcal{C}(\mathcal{Y})$, is Hadamard differentiable at $(\mu, \nu)$ tangentially to $\mathcal{M}_{0,\mu} \times \mathcal{M}_{0,\nu}$. Moreover, the Hadamard derivative is exactly the same as the derivative obtained when the domain is viewed as a subset of $\ell^\infty(\mathcal{G}_1) \times \ell^\infty(\mathcal{G}_1)$. $\qquad\square$

Now, we lift the Hadamard differentiability of entropic potentials from $\mathcal{C}(\mathcal{Y}) \times \mathcal{C}(\mathcal{Y})$ to $\mathcal{H} \times \mathcal{H}$.

**Proposition C.6** (Hadamard differentiability of entropic potentials in $\mathcal{H}$). *The $\mathcal{P}(\mathcal{Y}) \times \mathcal{P}(\mathcal{Y}) \to \mathcal{H} \times \mathcal{H}$ map $(\mu, \nu) \mapsto (\varphi_1^{(\mu,\nu)}, \varphi_0^{(\mu,\nu)})$ is Hadamard differentiable at $(\mu, \nu)$, tangentially to $\mathcal{M}_{0,\mu} \times \mathcal{M}_{0,\nu}$.*

*Proof.* The argument follows the same strategy as Goldfeld et al. (2024, Thm. 3), so we only sketch the main steps and refer to that result for full details. For each $i \in \{0, 1\}$, Goldfeld et al. (2024, Lem. 4) establish that the maps $(\mu, \nu) \mapsto \varphi_i^{(\mu,\nu)}$, $i \in \{1, 0\}$, are Hadamard differentiable in $\mathcal{C}(\mathcal{Y})$. Since $\mathcal{Y}$ is compact, the canonical embedding $\mathcal{C}(\mathcal{Y}) \hookrightarrow L^2(\mathcal{Y})$ is continuous. It follows that, for every multi-index $\alpha = (\alpha_1, \dots, \alpha_d) \in \mathbb{N}_0^d$ ($\mathbb{N}_0 = \mathbb{N} \cup \{0\}$) with $0 < |\alpha| \le s$, the functional $(\mu, \nu) \mapsto D^\alpha \varphi_i^{(\mu,\nu)}, \mathcal{P}_\mu \times \mathcal{P}_\nu \subset \ell^\infty(\mathcal{H}_1) \times \ell^\infty(\mathcal{H}_1) \to L^2(\mathcal{Y})$ is Hadamard differentiable. We denote the derivative by $[D^\alpha \varphi_i]'[\mu, \nu] : \mathcal{M}_{0,\mu} \times \mathcal{M}_{0,\nu} \to L^2(P^*)$.

Now consider paths $(\mu_t)_{t \ge 0} = (\mu + t\gamma_t^1)_{t \ge 0} \subset \mathcal{P}_\mu$ and $(\nu_t)_{t \ge 0} = (\nu + t\gamma_t^2)_{t \ge 0} \subset \mathcal{P}_\nu$ such that $\gamma_t^i \to \gamma^i$ as $t \downarrow 0$ in $\ell^\infty(\mathcal{H}_1)$. Pick any sequence $t_n \downarrow 0$ as $n \to \infty$, then $q_{t_n} := t_n^{-1} \left( \varphi_i^{(\mu_{t_n}, \nu_{t_n})} - \varphi_i^{(\mu,\nu)} \right)$ is Cauchy in $\mathcal{H}$. This is because for any $n, m \in \mathbb{N}$,

$$\|q_{t_n} - q_{t_m}\|_{\mathcal{H}}^2 = \sum_{|\alpha| \le s} \left\| \frac{D^\alpha \varphi_i^{(\mu_{t_n}, \nu_{t_n})} - D^\alpha \varphi_i^{(\mu,\nu)}}{t_n} - \frac{D^\alpha \varphi_i^{(\mu_{t_m}, \nu_{t_m})} - D^\alpha \varphi_i^{(\mu,\nu)}}{t_m} \right\|_{L^2(\mathcal{Y})}^2,$$

which converges to $0$ because each component term in the finite sum of the RKHS above is Cauchy in $L^2$. Since there are finitely many terms for $|\alpha| \le s$, the RHS converges to $0$ as $n, m \to \infty$. By completeness of $\mathcal{H}$, the limit exists in $\mathcal{H}$. So let $t_n^{-1} \left( \varphi_i^{(\mu_{t_n}, \nu_{t_n})} - \varphi_i^{(\mu,\nu)} \right) \to \overline{\varphi_i}$ in $\mathcal{H}$. Then, by definition of the limit in the Sobolev space $\mathcal{H}$, $t_n^{-1} \left( D^\alpha \varphi_i^{(\mu_{t_n}, \nu_{t_n})} - D^\alpha \varphi_i^{(\mu,\nu)} \right) \to D^\alpha \overline{\varphi_i}$ in $L^2(\mathcal{Y})$. On the other hand, by the Hadamard differentiability of $(\mu, \nu) \mapsto D^\alpha \varphi_i^{(\mu,\nu)}$, the same limit must equal $[D^\alpha \varphi_i]'[\mu, \nu](\gamma^1, \gamma^2)$. Therefore,

$$D^\alpha \overline{\varphi_i} = [D^\alpha \varphi_i]'[\mu, \nu](\gamma^1, \gamma^2) \qquad \text{for all } \alpha \in \mathbb{N}_0^d \text{ with } |\alpha| \le s.$$

This shows that $\overline{\varphi_i}$ is exactly the Sobolev function in $\mathcal{H}$ whose weak derivatives coincide with the coordinatewise directional derivatives, that is $D^\alpha \overline{\varphi_i} = [D^\alpha \varphi_i]'[\mu, \nu])(\gamma^1, \gamma^2)$. Therefore, if the Hadamard derivative $\varphi_i'[\mu, \nu]$ exists, then $\varphi_i'[\mu, \nu](\gamma^1, \gamma^2) = \overline{\varphi_i}$. Using this pointwise characterization, indeed the map $(\gamma^1, \gamma^2) \mapsto \varphi_i'[\mu, \nu](\gamma^1, \gamma^2)$ is linear and

continuous from $\mathcal{M}_{0,\mu} \times \mathcal{M}_{0,\nu}$ into $\mathcal{H}$ because $D^\alpha \varphi'_i[\mu,\nu] = [D^\alpha \varphi_i]'[\mu,\nu]$ is a linear and continuous function on $\mathcal{M}_{0,\mu} \times \mathcal{M}_{0,\nu}$ and summing over finitely such $\alpha$ gives the linearity and continuity of $\varphi'_i[\mu,\nu]$ on $\mathcal{M}_{0,\mu} \times \mathcal{M}_{0,\nu}$. The explicit form of $\varphi'_i[\mu,\nu]$ is presented in Kokot & Luedtke (2025, Cor. 43). $\square$

The following corollary provides the explicit form of the Hadamard derivatives of the centered entropic potentials maps $v_1 : (\mu,\nu) \mapsto v_1^{(\mu,\nu)}$ and $v_0 : (\mu,\nu) \mapsto v_0^{(\mu,\nu)}$ under the null.

**Corollary C.7.** *The Hadamard derivatives of $v_i : (\mu,\nu) \mapsto v_i^{(\mu,\nu)}, \mathcal{P}(\mathcal{Y}) \times \mathcal{P}(\mathcal{Y}) \subset \ell^\infty(\mathcal{H}_1) \times \ell^\infty(\mathcal{H}_1) \to \mathcal{H}$ for $i \in \{1,0\}$ are given by*

$$v'_1[\mu,\mu](\gamma^1,\gamma^2) = K_\mu(\gamma^1 - \gamma^2) + \varepsilon\rho\mathbf{1} \quad and \quad v'_0[\mu,\mu)](\gamma^1,\gamma^2) = -K_\mu(\gamma^1 - \gamma^2) - \varepsilon\rho\mathbf{1}$$

*for a constant $\rho \in \mathbb{R}$. This constant is fully expressed in Kokot & Luedtke (2025, Thm. 40), and omitted here because it is inconsequential in our case and ends up getting canceled out.*

The corollary is stated without proof and follows directly from Kokot & Luedtke (2025, Cor. 43) by plugging $v'_i[\mu,\mu](\gamma^1,\gamma^2) = \varphi'_i[\mu,\mu](\gamma^1,\gamma^2) - \varphi'_i[\mu,\mu](\gamma^{2-i},\gamma^{2-i})$ for $i \in \{1,0\}$.

**Lemma C.8** (Thm. 48 in Kokot & Luedtke, 2025). *The map $S_\varepsilon : (\mu,\nu) \mapsto S_\varepsilon(\mu,\nu), \mathcal{P}(\mathcal{Y}) \times \mathcal{P}(\mathcal{Y}) \subset \ell^\infty(\mathcal{H}_1) \times \ell^\infty(\mathcal{H}_1) \to \mathbb{R}$ is Hadamard differentiable at $(\mu,\mu)$ with Hadamard derivative $S'_\varepsilon[\mu,\mu] : \mathcal{M}_{0,\mu} \times \mathcal{M}_{0,\mu} \to \mathbb{R}$ given by*

$$S''_\varepsilon[\mu,\mu](\gamma^1,\gamma^2) = (\gamma^1 - \gamma^2)K_\mu(\gamma^1 - \gamma^2).$$

We do not reproduce the full proof here, since the argument follows from Kokot & Luedtke (2025, Thm. 48) after replacing the required topologies. Kokot & Luedtke (2025, Thm. 48) derive the second-order Hadamard derivative (under the null) of a localized quadratic approximation of the Sinkhorn divergence from Kokot & Luedtke (2025, Lem. 45). In that argument, second-order Hadamard differentiability of the localized Sinkhorn divergence reduces to first-order Hadamard differentiability of the entropic potentials. This required first-order differentiability in the $\ell^\infty(\mathcal{H}_1) \times \ell^\infty(\mathcal{H}_1)$ topology is established in Prop. C.6, together with an explicit characterization of the corresponding derivative. Substituting this derivative formula into the proof of Kokot & Luedtke (2025, Thm. 48) yields the stated Hadamard derivative $S''_\varepsilon[\mu,\mu](\gamma^1,\gamma^2) = (\gamma^1 - \gamma^2) K_\mu (\gamma^1 - \gamma^2)$.

## D. Pathwise Differentiability of STE

In this appendix, we prove the main results stated in Sec. 3. We begin by introducing notation that will be used throughout the proofs. For any score function $s \in \dot{\mathcal{P}}_P$, define

$$s_X(x) := \mathbb{E}_P\left[s(X,A,Y) \mid X = x\right] \quad and \quad s_{Y \mid A,X}(y \mid a,x) := s(x,a,y) - \mathbb{E}_P\left[s(X,A,Y) \mid A = a, X = x\right]. \quad (32)$$

For notational convenience, we also introduce evaluation operators: for functions $f : \mathcal{Z} \to \mathbb{R}$, $g : \{0,1\} \times \mathcal{X} \to \mathbb{R}$, and $h : \mathcal{X} \to \mathbb{R}$, we use the operator notation $\mathcal{E}_z f = f(z)$, $\mathcal{E}_{a,x} g = g(a,x)$, and $\mathcal{E}_x h = h(x)$.

We recall the expectation operators $P_{Y \mid A,X}, P_{A \mid X}$, and $P_X$ introduced in (7). For $f : \mathcal{Z} \to \mathbb{R}$, $a \in \{0,1\}$, $x \in \mathcal{X}$, and $z' \in \mathcal{Z}'$, we define the shorthand notation

$$(P_{Y \mid a,X} f)(z') := (P_{Y \mid A,X} f)(x',a,y')$$
$$(P_{Y \mid a,x} f)(z') := (P_{Y \mid A,X} f)(x,a,y') \quad (33)$$

Since the right-hand side of the first equality is invariant in $(a',y')$ and the second is invariant in $z'$, we sometimes write $P_{Y \mid a,X} f)(x') := (P_{Y \mid a,X} f)(z')$ for a generic $(a',y')$ or $P_{Y \mid a,x} f := (P_{Y \mid a,x} f)(z')$ for a generic $z'$. We also define the

composed operators

$$P_{A,X} := P_X P_{A \mid X} \quad \text{and} \quad P_{Y,A \mid X} := P_{A \mid X} P_{Y \mid A,X}. \tag{34}$$

Similarly, because $(P_{Y \mid A,X} f)(z)$ is invariant in $y$ and $(P_{Y,A \mid X} f)(z)$ is invariant in $(a, y)$, we sometimes write $P_{Y \mid A,X} f(a, x) := P_{Y \mid A,X} f(z)$ and $P_{Y,A \mid X} f(x) := P_{Y,A \mid X} f(z)$. Finally, when clear from context, we extend $P_{A \mid X}$ (and $P_{A,X}$) and $P_X$ to functions of $(A, X)$ and $X$, respectively. Specifically, for $f : \{0, 1\} \times \mathcal{X} \to \mathbb{R}$ and $g : \mathcal{X} \to \mathbb{R}$, we set $P_{A \mid X} f(x) = \sum_{a \in \{1,0\}} f(a, x) e_P(a \mid x)$, $P_{A,X} f(x) = \int_{\mathcal{X}} \sum_{a \in \{1,0\}} f(a, x) e_P(a \mid x) dP_X(x)$, and $P_X g = \int_{\mathcal{X}} g(x) dP_X(x)$.

### D.1. First-order pathwise differentiability of STE

The first-order pathwise differentiability of $\mathcal{S} = S_\varepsilon \circ J \circ \Psi$ is established in three steps: (1) Hadamard differentiability of $S_\varepsilon$ (Lem. C.4), (2) Hadamard differentiability of $J$ (Lem. D.1), and (3) pathwise differentiability of $\Psi$ (Lem. D.2).

**Lemma D.1** (Hadamard derivative of $J$). *The functional $J : \mathcal{H} \times \mathcal{H} \to \ell^\infty(\mathcal{H}_1) \times \ell^\infty(\mathcal{H}_1)$ is Hadamard differentiable at every $(f, g) \in \mathcal{H} \times \mathcal{H}$ and the Hadamard derivative $J'[f, g] : \mathcal{H} \times \mathcal{H} \to \ell^\infty(\mathcal{H}_1) \times \ell^\infty(\mathcal{H}_1)$ is given by*

$$J'[f, g](h^1, h^2) = J(h^1, h^2).$$

*Proof.* Consider paths $(f_t)_{t>0}$ and $(g_t)_{t>0}$ in $\mathcal{H}$ such that $f_t = f + t h_t^1$ and $g_t = g + t h_t^2$ such that $h_t^1 \to h^1$ and $h_t^2 \to h^2$ as $t \to 0$ in $\mathcal{H}$ for some $(h^1, h^2) \in \mathcal{H} \times \mathcal{H}$. Then,

$$\frac{1}{t} \left[ J(f_t, g_t) - J(f, g) \right] = J(h_t^1, h_t^2)$$

$$\implies \lim_{t \to 0} \frac{1}{t} \left[ J(f_t, g_t) - J(f, g) \right] = \lim_{t \to 0} J(h_t^1, h_t^2).$$

We have that $J(h_t^1, h_t^2)$ converges to $J(h^1, h^2)$ in the supnorm as

$$\left\| J(h_t^1, h_t^2) - J(h^1, h^2) \right\|_{\ell^\infty(\mathcal{H}_1)} = \sup_{h \in \mathcal{H}_1} \left| \langle h, h_t^1 - h^1 \rangle_{\mathcal{H}} \right| + \sup_{h \in \mathcal{H}_1} \left| \langle h, h_t^2 - h^2 \rangle_{\mathcal{H}} \right|$$

$$\leq \left\| h_t^1 - h^1 \right\|_{\mathcal{H}} + \left\| h_t^2 - h^2 \right\|_{\mathcal{H}} \to 0.$$

This completes the proof. $\qquad\square$

**Lemma D.2** (Pathwise derivative of $\Psi$ Luedtke & Chung, 2024, Ex. 3). *Recall that (1) the Sobolev kernel $k : \mathcal{Y} \times \mathcal{Y} \to \mathbb{R}$ is bounded, symmetric, and positive definite and (2) the model $\mathcal{P}$ satisfies strong positivity. Then, the parameter $\Psi : \mathcal{P} \to \mathcal{H} \times \mathcal{H}$ is pathwise differentiable at all $P \in \mathcal{P}$ with local parameter $D\Psi_P : \dot{\mathcal{P}}_P \to \mathcal{H} \times \mathcal{H}$ given by*

$$D\Psi_P(s) = \begin{bmatrix} D\psi_P^1(s) \\ D\psi_P^0(s) \end{bmatrix} = \begin{bmatrix} \int_{\mathcal{X}} \int_{\mathcal{Y}} k(y, \cdot) \left[ s_{Y \mid A,X}(y \mid 1, x) + s_X(x) \right] P_{Y \mid A,X}(dy \mid 1, x) P_X(dx) \\ \int_{\mathcal{X}} \int_{\mathcal{Y}} k(y, \cdot) \left[ s_{Y \mid A,X}(y \mid 0, x) + s_X(x) \right] P_{Y \mid A,X}(dy \mid 0, x) P_X(dx) \end{bmatrix}. \tag{35}$$

Finally, we combine three ingredients through a chain rule: the Hadamard differentiability of the Sinkhorn divergence $S_\varepsilon$, the Hadamard differentiability of the canonical embedding $J$, and the pathwise differentiability of KME $\Psi$. Together, these results yield the local parameter of the STE.

*Proof of Lem. 3.1.* Consider the submodel $(P_t, t \in [0, \delta)) \subset \mathfrak{P}(P, \mathcal{P}, s)$ starting at $P_0 = P$. In order to derive the local

parameter of $\mathcal{S}$, we consider the following residual

$$\frac{1}{t}\left[\mathcal{S}(P_t) - \mathcal{S}(P)\right] = \frac{1}{t}\left[S_\varepsilon \circ J(\Psi(P_t)) - S_\varepsilon \circ J(\Psi(P))\right]$$

$$= \frac{1}{t}\left[S_\varepsilon \circ J\left(\Psi(P) + t\frac{[\Psi(P_t) - \Psi(P)]}{t}\right) - S_\varepsilon \circ J(\Psi(P))\right].$$

From Lem. D.2, we know that $\left\|\frac{[\Psi(P_t)-\Psi(P)]}{t} - D\Psi_P(s)\right\|_{\mathcal{H}\times\mathcal{H}} = o(1)$. Since $S_\varepsilon$ and $J$ are both Hadamard differentiable functions from Lem. C.4 and Lem. D.1, respectively, $S_\varepsilon \circ J$ is also Hadamard differentiable via the chain rule. Consequently,

$$\lim_{t\to 0}\frac{1}{t}\left[S_\varepsilon \circ J\left(\Psi(P) + t\frac{[\Psi(P_t) - \Psi(P)]}{t}\right) - S_\varepsilon \circ J(\Psi(P))\right] = (S_\varepsilon \circ J)'\left[\Psi(P)\right]\left(D\Psi_P(s)\right).$$

The derivative $(S_\varepsilon \circ J)'$ can be obtained via the chain rule: for every $(f,g) \in \mathcal{H}\times\mathcal{H}$,

$$(S_\varepsilon \circ J)'[f,g](h_1,h_2) = S_\varepsilon'\left[J(f), J(g)\right]\left(J(h_1), J(h_2)\right).$$

Plugging $(f,g) = \Psi(P)$ and $(h_1,h_2) = D\Psi_P(s)$ in the above expression, and using Lem. C.4, we get that

$$\lim_{t\to 0}\frac{1}{t}\left(\mathcal{S}(P_t) - \mathcal{S}(P)\right) = \int v_1^{(P_1,P_0)}\, d(J \circ D\psi_P^1(s)) + \int v_0^{(P_1,P_0)}\, d(J \circ D\psi_P^2(s)).$$

Using that $v_1^{(P_1,P_0)}, v_0^{(P_1,P_0)} \in \mathcal{H}$ and the form of $D\psi_P^1$ and $D\psi_P^2$ from (35), we get that

$$D\mathcal{S}_P(s) = \sum_{a'\in\{1,0\}} \int_{\mathcal{X}}\int_{\mathcal{Y}} v_{a'}^{(P_1,P_0)}(y)\left[s_{Y\,|\,A,X}(y\,|\,a',x) + s_X(x)\right] P_{Y\,|\,A,X}(dy\,|\,a',x)P_X(dx).$$

By definition of $s_X$ and $s_{Y\,|\,A,X}$ from (32), we have that

$$D\mathcal{S}_P(s) = \sum_{a'\in\{1,0\}} \int \frac{\mathbb{1}(a=a')}{e_P(a'\,|\,x)}\left(v_{a'}^{(P_1,P_0)}(y) - \mathbb{E}_P\left[v_{a'}^{(P_1,P_0)}(Y)\,|\,A=a, X=x\right]\right) s(z)\,P(dz)$$

$$+ \sum_{a'\in\{1,0\}} \int \left(\mathbb{E}_P\left[v_{a'}^{(P_1,P_0)}(Y)\,|\,A=a', X=x\right] - \mathbb{E}_P\mathbb{E}_P\left[v_{a'}^{(P_1,P_0)}(Y)\,|\,A=a', X=x\right]\right) s(z)\,P(dz).$$

Indeed, $D\mathcal{S}_P$ is a linear function of $s$ and is therefore a valid local parameter for $\mathcal{S}$ at $P$. The expectations in the above expression can be simplified using operator notation of $P_{Y\,|\,A,X}, P_{Y,A\,|\,X}, P_X$.

$$D\mathcal{S}_P(s) = \sum_{a'\in\{1,0\}} \int \left[\frac{\mathbb{1}(a=a')}{e_P(a'\,|\,x)}(I_z - P_{Y\,|\,a,x})\, v_{a'}^{(P_1,P_0)}(Y)\right] s(z)\,P(dz)$$

$$+ \sum_{a'\in\{1,0\}} \int \left[(P_{Y\,|\,a',x} - P_X P_{Y\,|\,a',X})\, v_{a'}^{(P_1,P_0)}(Y)\right] s(z)\,P(dz)$$

$$= \int \left[(I_z - P_{Y\,|\,a,x})\left(\frac{A}{e_P(1\,|\,X)}v_1^{(P_1,P_0)}(Y) + \frac{1-A}{e_P(0\,|\,X)}v_0^{(P_1,P_0)}(Y)\right)\right] s(z)\,P(dz)$$

$$+ \int \left[(I_x - P_X)P_{Y\,|\,A,X}\left(\frac{A}{e_P(1\,|\,X)}v_1^{(P_1,P_0)}(Y) + \frac{1-A}{e_P(0\,|\,X)}v_0^{(P_1,P_0)}(Y)\right)\right] s(z)\,P(dz).$$

The equality above follows from the fact that for any measurable function $h : \mathcal{Y} \to \mathbb{R}$,

$$P_{Y\,|\,1,X}h(Y) = P_{Y,A\,|\,X}\left(\frac{A}{e_P(1\,|\,X)}h(Y)\right) \quad \text{and} \quad P_{Y\,|\,0,X}h(Y) = P_{Y,A\,|\,X}\left(\frac{1-A}{e_P(0\,|\,X)}h(Y)\right).$$

The four terms in $D\mathcal{S}_P(s)$ can be combined into

$$= \int \left(I_z - P_{Y\,|\,a,x} + (I_x - P_X)P_{Y,A\,|\,X}\right) \left(\frac{A}{e_P(1\,|\,X)}v_1^{(P_1,P_0)}(Y) + \frac{1-A}{e_P(0\,|\,X)}v_0^{(P_1,P_0)}(Y)\right) s(z)\,dP(z)$$

$$= \int \left(I_z - \left(I_{a,x} - (I_x - P_X)\,P_{A\,|\,X}\right)P_{Y\,|\,A,X}\right) \left(\frac{A}{e_P(1\,|\,X)}v_1^{(P_1,P_0)}(Y) + \frac{1-A}{e_P(0\,|\,X)}v_0^{(P_1,P_0)}(Y)\right) s(z)\,dP(z).$$

The (locally nonparametric) EIF $\dot{\mathcal{S}}_P : \mathcal{Z} \to \mathbb{R}$ is the unique function satisfying $D\mathcal{S}_P(s) = \langle \dot{\mathcal{S}}_P, s \rangle_{L^2(P)}$ for all $s$, and hence, this completes the proof. $\qquad\square$

### D.2. Second-order pathwise differentiability of STE under the null

*Proof of Thm. 3.2.* Let $P \in \mathcal{H}_0$, so that $P_1 = P_0$. Consider the submodel $(P_t, t \in [0, \delta)) \subset \mathfrak{P}(P, \mathcal{P}, s)$ starting at $P_0 = P$. By Lem. C.8, the Sinkhorn divergence $S_\varepsilon$ is second-order Hadamard differentiable under the null. Since $J$ is linear, it is also second-order Hadamard differentiable (with derivative zero). For notational convenience, define $T_\varepsilon = S_\varepsilon \circ J$. Our goal is to derive the second-order local parameter of $\mathcal{S}$, which amounts to computing the second derivative of $t \mapsto \mathcal{S}(P_t)$ at $t = 0$. By definition,

$$\begin{aligned}
D^2\mathcal{S}_P(s) = \frac{d^2}{dt^2}\mathcal{S}(P_t)\Big|_{t=0} &= \lim_{t \to 0} \frac{\mathcal{S}(P_t) - \mathcal{S}(P) - t\frac{d}{dt}\mathcal{S}(P_t)|_{t=0}}{t^2/2} \\
&= \lim_{t \to 0} \frac{\mathcal{S}(P_t) - \mathcal{S}(P) - tD\mathcal{S}_P(s)}{t^2/2} \\
&= \lim_{t \to 0} \frac{2}{t^2} \left\{ T_\varepsilon(\Psi(P_t)) - T_\varepsilon(\Psi(P)) - tT_\varepsilon'[\Psi(P)](D\Psi_P(s)) \right\}.
\end{aligned}$$

The last equality follows from the proof of Lem. 3.1 where we showed that $D\mathcal{S}_P(s) = T_\varepsilon'[\Psi(P)](D\Psi_P(s))$. Adding and subtracting the term $T_\varepsilon'[\Psi(P)](\Psi(P_t) - \Psi(P))$ to the RHS inside the limit yields

$$\begin{aligned}
&\frac{2}{t^2}\left[T_\varepsilon(\Psi(P_t)) - T_\varepsilon(\Psi(P)) - tT_\varepsilon'[\Psi(P)](D\Psi_P(s))\right] \\
&= \frac{[T_\varepsilon(\Psi(P_t)) - T_\varepsilon(\Psi(P)) - T_\varepsilon'[\Psi(P)]\,(\Psi(P_t) - \Psi(P))]}{t^2/2} + T_\varepsilon'[\Psi(P)]\left(\frac{\Psi(P_t) - \Psi(P) - tD\Psi_P(s)}{t^2/2}\right).
\end{aligned}$$

The first additive term in the above equality can be written as

$$\begin{aligned}
&\frac{2}{t^2}[T_\varepsilon(\Psi(P_t)) - T_\varepsilon(\Psi(P)) - T_\varepsilon'[\Psi(P)]\,(\Psi(P_t) - \Psi(P))] \\
&= \frac{2}{t^2}[T_\varepsilon\left(\Psi(P) + (\Psi(P_t) - \Psi(P))\right) - T_\varepsilon(\Psi(P)) - T_\varepsilon'[\Psi(P)]\,(\Psi(P_t) - \Psi(P))] \\
&= T_\varepsilon''[\Psi(P)](D\Psi_P(s)) + o(1).
\end{aligned}$$

Because $T_\varepsilon'[\Psi(P)]$ is a linear map, the limit of the second additive term can be written as

$$\lim_{t \to 0} T_\varepsilon'[\Psi(P)]\left(\frac{\Psi(P_t) - \Psi(P) - tD\Psi_P(s)}{t^2/2}\right) = T_\varepsilon'[\Psi(P)]\left(D^2\Psi_P(s)\right).$$

However, under the null hypothesis, we have $T_\varepsilon'[\Psi(P)] \equiv 0$, so the second term vanishes. Consequently,

$$D^2\mathcal{S}_P(s) = T_\varepsilon''[\Psi(P)](D\Psi_P(s)).$$

By the second-order chain rule for Hadamard derivatives (Goldfeld et al., 2024, Lemma 12),

$$T_\varepsilon''[f,g](\gamma^1,\gamma^2) = S_\varepsilon''[J(f,g)] \circ J'[f,g](\gamma^1,\gamma^2) + S_\varepsilon'[J(f,g)] \circ J''[f,g](\gamma^1,\gamma^2) = S_\varepsilon''[J(f,g)] \circ J(\gamma^1,\gamma^2).$$

The last equality follows because $J$ is linear and hence $J'' \equiv 0$. Setting $(f,g) = \Psi(P) = (m(P_1), m(P_1))$, we obtain

$$T_\varepsilon''[\Psi(P)](D\Psi_P(s)) = S_\varepsilon''[P_1, P_1](J(D\Psi_P(s)))$$

From Lem. C.8,

$$S_\varepsilon''[P_1, P_0](J(D\Psi_P(s))) = (\gamma^1 - \gamma^2)K_{P_1}(\gamma^1 - \gamma^2),$$

where $(\gamma^1, \gamma^2) \in \ell^\infty(\mathcal{H}_1) \times \ell^\infty(\mathcal{H}_1)$ are given by

$$\gamma^1 : h \in \mathcal{H}_1 \mapsto \int_\mathcal{X} \int_\mathcal{Y} \{s_{Y|A,X}(y\,|\,1,x) + s_X(x)\}\, h(y)\, P_{Y|A,X}(dy\,|\,1,x)\, P_X(dx),$$

$$\gamma^2 : h \in \mathcal{H}_1 \mapsto \int_\mathcal{X} \int_\mathcal{Y} \{s_{Y|A,X}(y\,|\,0,x) + s_X(x)\}\, h(y) P_{Y|A,X}(dy\,|\,0,x)\, P_X(dx).$$

Although $k_{P_1}(y, \cdot)$ may not belong to $\mathcal{H}$, universality of the RKHS $\mathcal{H}$ guarantees approximation by a convergent sequence of elements in $\mathcal{H}$. Writing the second-order local parameter of $\mathcal{S}$ in kernel $k_{P_1}$ notation,

$$D^2\mathcal{S}_P(s) = \int_\mathcal{Y} \int_\mathcal{Y} k_{P_1}(y_1, y_2)\, d(\gamma^1 - \gamma^2)(y_1)\, d(\gamma^1 - \gamma^2)(y_2).$$

Expanding the above using the definitions of $s_{Y|A,X}$ and $s_X$ from (32), we obtain that $D^2\mathcal{S}_P(s)$ is equal to

$$\int_\mathcal{Z} \int_\mathcal{Z} (Q_{P,z_1} \otimes Q_{P,z_2}) \left( \frac{a_1}{e_P(1\,|\,x_1)} - \frac{1-a_1}{e_P(0\,|\,x_1)} \right) \left( \frac{a_2}{e_P(1\,|\,x_2)} - \frac{1-a_2}{e_P(0\,|\,x_2)} \right) k_{P_1}(y_1, y_2)$$
$$\times s(z_1)\, s(z_2)\, P(dz_1)\, P(dz_2),$$

where $Q_{P,z_j} = (\mathcal{E}_{z_j} - (\mathcal{E}_{a_j,x_j} - (\mathcal{E}_{x_j} - P_X)P_{A|X})P_{Y|A,X})$. By definition, the (locally nonparametric) second-order EIF $\ddot{\mathcal{S}}_P : \mathcal{Z} \times \mathcal{Z} \to \mathbb{R}$ is the unique function that satisfies

$$D^2\mathcal{S}_P(s) = \int \int \ddot{\mathcal{S}}_P(z_1, z_2)\, s(z_1)\, s(z_2)\, P(dz_1)\, P(dz_2),$$

which yields the claimed form of $\ddot{\mathcal{S}}_P$ and completes the proof. □

## E. One-Step Estimators of STE

Let $\widehat{P}$ be an initial estimator of $P^*$ constructed from $n$ independent samples drawn from $P^*$, and let $P_n$ denote the empirical measure based on another set of $n$ independent samples from $P^*$, taken independently of $\widehat{P}$. In this section, we study the asymptotic distributions of the one-step estimator for STE under the alternative hypothesis $(P_1^* \neq P_0^*)$, and of the second-order one-step estimator for STE under the null hypothesis $(P_1^* = P_0^*)$. Throughout this section, for notational convenience, we write $\widehat{v}_a$ for $v_a^{(\widehat{P}_1, \widehat{P}_0)}$, $v_a^*$ for $v_a^{(P_1^*, P_0^*)}$, $\hat{e}(a\,|\,x)$ for $e_{\widehat{P}}(a\,|\,x)$, and $e^*(a\,|\,x)$ for $e_{P^*}(a\,|\,x)$, for $a \in \{1, 0\}$.

### E.1. Limit theorem for first-order one-step estimator

As discussed in Sec. 4, the one-step estimator $\widehat{\mathcal{S}} = \mathcal{S}(\widehat{P}) + P_n \dot{\mathcal{S}}_{\widehat{P}}$ can be expanded as

$$\widehat{\mathcal{S}} - \mathcal{S}^* = \underbrace{(P_n - P^*)(\dot{\mathcal{S}}_{\widehat{P}} - \dot{\mathcal{S}}_{P^*})}_{\mathcal{D}_n} + \underbrace{\left\{\mathcal{S}(\widehat{P}) + P^*\dot{\mathcal{S}}_{\widehat{P}} - \mathcal{S}(P^*)\right\}}_{\mathcal{R}_n} + P_n \dot{\mathcal{S}}_{P^*}.$$

Here, the goal is to show that the drift and remainder terms, $\mathcal{D}_n$ and $\mathcal{R}_n$ respectively, are both $o_p(n^{-1/2})$. This is proven in Lem. E.2 and Lem. E.3 respectively.

We begin with a helper lemma.

**Lemma E.1** (Empirical process drift bound). *Let $P \in \mathcal{P}$ and $f_n \in L^2(P)$ be a sequence of random measurable functions such that $\|f_n\|_{L^2(P)} = o_p(n^{-1/2})$. Let $P_n$ be an empirical distribution based on $n$ iid samples from $P$, independent of $f_n$, then $(P_n - P)f_n = o_P(n^{-1})$.*

*Proof.* We will show that for every $\delta > 0$,

$$\lim_{n \to \infty} \mathbb{P}\left(n(P_n - P)f_n > \delta\right) = 0,$$

where $\mathbb{P}$ is the probability taken over the measure $(P^*)^n$. Let $\{Z_1, \ldots, Z_n\}$ be the iid random variables used to construct $f_n$ and let $E_n$ denote the event that $n\|f_n\|_{L^2(P^*)}^2 \leq \delta^2$; $E_n^c$ be its complement. Then,

$$\begin{aligned}
\mathbb{P}\left(n(P_n - P)f_n > \delta\right) &\leq \mathbb{P}\left(\{n(P_n - P)f_n > \delta\} \cap E_n\right) + \mathbb{P}(E_n^c) \\
&= \mathbb{E}_P\left[\mathbb{P}\left(\{n(P_n - P)f_n > \delta\} \cap E_n \mid Z_1, \ldots, Z_n\right)\right] + \mathbb{P}(E_n^c) \qquad \text{(law of total probability)} \\
&= \mathbb{E}_P\left[\mathbb{1}_{E_n}\mathbb{P}\left(\{n(P_n - P)f_n > \delta\} \mid Z_1, \ldots, Z_n\right)\right] + \mathbb{P}(E_n^c).
\end{aligned}$$

Since we are given that $\sqrt{n}\|f_n\|_{L^2(P^*)} \xrightarrow{P} 0$, we have that $\lim_{n \to \infty} \mathbb{P}(E_n^c) = 0$. Therefore, we just need to show that the first term on the RHS above converges to 0 as $n \to \infty$. Consider the integrand in the first term. Using Chebyshev's inequality,

$$\begin{aligned}
\mathbb{P}\left(\{n(P_n - P)f_n > \delta\} \mid Z_1, \ldots, Z_n\right) &\leq \frac{n^2}{\delta^2}\mathbb{E}\left[\{(P_n - P)f_n\}^2 \mid Z_1, \ldots, Z_n\right] \\
&= \frac{n^2 \mathrm{Var}\left((P_n - P)f_n \mid Z_1, \ldots, Z_n\right)}{\delta^2} \\
&= \frac{n \mathrm{Var}\left(f_n \mid Z_1, \ldots, Z_n\right)}{\delta^2} \\
&\leq \frac{n\|f_n\|_{L^2(P)}^2}{\delta^2}.
\end{aligned}$$

Because under $E_n$, $n\|f_n\|_{L^2(P^*)}^2 \leq \delta$,

$$U_n := \mathbb{1}_{E_n}\mathbb{P}\left(\{n(P_n - P)f_n > \delta\} \mid Z_1, \ldots, Z_n\right) \leq \min\left\{1, \frac{n\|f_n\|_{L^2(P^*)}^2}{\delta^2}\right\}.$$

Since $n\|f_n\|_{L^2(P^*)}^2 \xrightarrow{P} 0$, we have that $U_n \xrightarrow{P} 0$ and $U_n$ is uniformly integrable. Using the dominated convergence theorem, we have that $\mathbb{E}[U_n] \xrightarrow{P} 0$. This completes the proof. $\qquad \square$

Recall the definition of expectations operators (7) and (33). We denote $\widehat{p} = \frac{d\widehat{P}}{d\lambda}$ and $p^* = \frac{dP^*}{d\lambda}$. Further, let $\widehat{p}_{Y \mid A, X}$ and $p_{Y \mid A, X}^*$ be the associated conditional densities.

**Lemma E.2** (Rate for first-order drift term). *Let $P^* \notin \mathcal{H}_0$ and assume that the nuisance estimators satisfy*

1. $\|\hat{e}(1 \mid \cdot) - e^*(1 \mid \cdot)\|_{L^2(P_X^*)} = o_p(1)$;

2. $\|(x,a,y) \mapsto \widehat{v}_a(y) - v_a^*(y)\|_{L^2(P^*)} = o_p(1)$;

3. $\int \left( \hat{p}_{Y|A,X}(y \mid a,x) - p^*_{Y|A,X}(y \mid a,x) \right)^2 d\lambda(y)\, dP_X^*(x) = o_p(1)$ *for* $a \in \{1,0\}$;

4. $\left\| \frac{\widehat{p}_{Y|A,X}}{p^*_{Y|A,X}} \right\|_{L^r(P^*)} = \mathcal{O}_p(1)$ *for some* $r > 1$.

*Then,*
$$|\mathcal{D}_n| = \left| (P_n - P^*)(\dot{\mathcal{S}}_{\widehat{P}} - \dot{\mathcal{S}}_{P^*}) \right| = o_p(n^{-1/2}).$$

The first three assumptions of this lemma impose standard consistency requirements on the three nuisance parameters—the propensity score, the entropic potentials, and the outcome regression—each in their respective norms (Luedtke & Chung, 2024, App. B.2.5). Consistency of the entropic potentials (the second assumption) in the $L^2$ norm follows directly from consistency in the Hölder norm established by del Barrio et al. (2023). The fourth assumption concerns the empirical likelihood ratio; this condition is mild given that $\|\widehat{p}_{Y|A,X}/p^*_{Y|A,X}\|_{L^1(P^*)} = 1$.

*Proof.* By Lem. E.1, a sufficient condition for showing that $|\mathcal{D}_n| = o_p(n^{-1/2})$ is that $\|\dot{\mathcal{S}}_{\widehat{P}} - \dot{\mathcal{S}}_{P^*}\|_{L^2(P^*)} = o_p(1)$, which we establish in what follows. Recall from (8), the EIF $\dot{\mathcal{S}}_P$ can be written as

$$\dot{\mathcal{S}}_P : (x,a,y) \mapsto f_P - P_{Y|A,X} f_P + P_{A,Y|X} f_P - P f_P, \quad f_P(x,a,y) = \frac{v_a^{(P_1,P_0)}(y)}{e_P(a \mid x)}.$$

Let $E > 0$ be the strong positivity constant, that is $\inf_{P \in \mathcal{P}} \operatorname{ess\,inf}_{(a,x)} e_P(a \mid x) > E$.

**Step 1: Control** $\left\| f_{\widehat{P}} - f_{P^*} \right\|_{L^2(P^*)}$. For each $a \in \{0,1\}$,

$$f_{\widehat{P}}(x,a,y) - f_{P^*}(x,a,y) = \frac{\widehat{v}_a}{\hat{e}(a \mid x)} - \frac{v_a^*}{e^*(a \mid x)} = \frac{\widehat{v}_a - v_a^*}{\hat{e}(a \mid x)} + v_a^* \frac{e^*(a \mid x) - \hat{e}(a \mid x)}{\hat{e}(a \mid x) e^*(a \mid x)}.$$

Using strong positivity, we have

$$\|f_{\widehat{P}} - f_{P^*}\|_{L^2(P^*)} \leq \frac{1}{E} \|(x,a,y) \mapsto \widehat{v}_a(y) - v_a^*(y)\|_{L^2(P^*)}$$
$$+ \frac{1}{E^2} \sum_{a \in \{1,0\}} \|v_a^*\|_{\mathcal{C}(\mathcal{Y})} \|\hat{e}(a \mid \cdot) - e^*(a \mid \cdot)\|_{L^2(P_X^*)} = o_p(1).$$

by assumptions 1 and 2 of the lemma, and uniform boundedness of $\|v_a^{(\mu,\nu)}\|_{\mathcal{C}(\mathcal{Y})}$. The centered entropic potentials are uniformly bounded because $\|\varphi_a^{(\mu,\nu)}\|_{\mathcal{C}(\mathcal{Y})}$ is uniformly bounded by a constant $C$, depending on $\|c\|_\infty$ (equivalently $|\mathcal{Y}|$ for quadratic cost), uniformly over $a \in \{1,0\}$ and $\mu, \nu \in \mathcal{P}(\mathcal{Y})$ (Goldfeld et al., 2024, Lem. 1).

**Step 2: Control** $\left\| \widehat{P}_{Y|A,X} f_{\widehat{P}} - P^*_{Y|A,X} f_{P^*} \right\|_{L^2(P^*)}$. Add and subtract $\widehat{P}_{Y|A,X} f_{P^*}$ to get

$$\left\| \widehat{P}_{Y|A,X} f_{\widehat{P}} - P^*_{Y|A,X} f_{P^*} \right\|_{L^2(P^*)} \leq \left\| \widehat{P}_{Y|A,X} \left( f_{\widehat{P}} - f_{P^*} \right) \right\|_{L^2(P^*)} + \left\| \left( \widehat{P}_{Y|A,X} - P^*_{Y|A,X} \right) f_{P^*} \right\|_{L^2(P^*)}.$$

For the first additive term, we use conditional Jensen's inequality, followed by the Hölder inequality.

$$\left\|\widehat{P}_{Y|A,X}\left(f_{\widehat{P}} - f_{P^*}\right)\right\|_{L^2(P^*)} \leq \left(P^*_{A,X}\widehat{P}_{Y|A,X}(f_{\widehat{P}} - f_{P^*})^2\right)^{1/2}$$

$$= \left(P^*\left(\frac{\widehat{p}_{Y|A,X}}{p^*_{Y|A,X}}(f_{\widehat{P}} - f_{P^*})^2\right)\right)^{1/2}$$

$$\leq \left\|\frac{\widehat{p}_{Y|A,X}}{p^*_{Y|A,X}}\right\|_{L^r(P^*)}^{1/2}\left\|f_{\widehat{P}} - f_{P^*}\right\|_{L^{2r/r-1}(P^*)}.$$

By the fourth assumption $\left\|\frac{\widehat{p}_{Y|A,X}}{p^*_{Y|A,X}}\right\|_{L^r(P^*)} = \mathcal{O}_p(1)$. Moreover, since $e_P$ and $v_a^{(P_1,P_0)}$ are uniformly bounded, the $L^2(P^*)$-consistency implies $L^s(P^*)$ consistency for all $s > 2$. As a consequence, following the same argument as in step 1, $\|f_{\widehat{P}} - f_{P^*}\|_{L^{2r/r-1}} = o_p(1)$. For the second additive term:

$$\left\|\left(\widehat{P}_{Y|A,X} - P^*_{Y|A,X}\right)f_{P^*}\right\|_{L^2(P^*)}^2 = P^*_{A,X}\left((a,x) \mapsto \frac{1}{e^*(a\mid x)^2}\left(\left(\widehat{P}_{Y|a,x} - P^*_{Y|a,x}\right)v_a^*\right)^2\right)$$

$$= P^*_X\left(\sum_{a\in\{1,0\}}\frac{1}{e^*(a\mid\cdot)}\left((\widehat{P}_{Y\mid a,\cdot} - P^*_{Y\mid a,\cdot})v_a^*\right)^2\right)$$

$$\leq \frac{1}{E}\sum_{a\in\{1,0\}}\int_{\mathcal{X}}\left(\int_{\mathcal{Y}}v_a^*(y)\left(\widehat{p}_{Y|A,X}(y\mid a,x) - p^*_{Y|A,X}(y\mid a,x)\right)d\lambda(y)\right)^2 dP^*_X(x)$$

$$\leq \frac{1}{E}\sum_{a\in\{1,0\}}\int_{\mathcal{X}}\left(\int_{\mathcal{Y}}|v_a^*(y)|\left|\widehat{p}_{Y|A,X}(y\mid a,\cdot) - p^*_{Y|A,X}(y\mid a,\cdot)\right|d\lambda(y)\right)^2 dP^*_X(x)$$

$$\leq \frac{C^2}{E}\sum_{a\in\{1,0\}}\int_{\mathcal{X}}\int_{\mathcal{Y}}\left(\widehat{p}_{Y|A,X}(y\mid a,x) - p^*_{Y|A,X}(y\mid a,x)\right)^2 d\lambda(y)\,dP^*_X(x)$$

$$= o_p(1).$$

The last inequality uses the Cauchy-Schwartz inequality and is $o_p(1)$ from the third assumption.

**Step 3: Control** $\left\|\widehat{P}_{A,Y|X}f_{\widehat{P}} - P^*_{A,Y|X}f_{P^*}\right\|_{L^2(P^*_X)}$. Note that $\widehat{P}_{A,Y|X} = \widehat{P}_{A|X}\widehat{P}_{Y|,A,X}$ and $P^*_{A,Y|X} = P^*_{A|X}P^*_{Y|,A,X}$. Add and subtract $P^*_{A|X}\widehat{P}_{Y|A,X}f_{\widehat{P}}$ to obtain

$$\left\|\widehat{P}_{A,Y|X}f_{\widehat{P}} - P^*_{A,Y|X}f_{P^*}\right\|_{L^2(P^*_X)} \leq \left\|\widehat{P}_{A|X}\left(\widehat{P}_{Y|A,X}f_{\widehat{P}} - P^*_{Y|A,X}f_{P^*}\right)\right\|_{L^2(P^*_X)}$$

$$+ \left\|\left(\widehat{P}_{A|X} - P^*_{A|X}\right)P^*_{Y|A,X}f_{P^*}\right\|_{L^2(P^*_X)}.$$

For the first additive term

$$
\begin{aligned}
\left\| \widehat{P}_{A|X} \left( \widehat{P}_{Y|A,X} f_{\widehat{P}} - P_{Y|A,X}^* f_{P^*} \right) \right\|_{L^2(P_X^*)}^2 &= P_X^* \left( x \mapsto \sum_{a \in \{1,0\}} \hat{e}(a \mid x) \left( \widehat{P}_{Y|a,x} f_{\widehat{P}} - P_{Y|a,x}^* f_{P^*} \right) \right)^2 \\
&\leq 2 P_X^* \left( x \mapsto \sum_{a \in \{1,0\}} \hat{e}(a \mid x)^2 \left( \widehat{P}_{Y|a,x} f_{\widehat{P}} - P_{Y|a,x}^* f_{P^*} \right)^2 \right) \\
&\leq 2 P_X^* \left( x \mapsto \sum_{a \in \{1,0\}} \hat{e}(a \mid x) \left( \widehat{P}_{Y|a,x} f_{\widehat{P}} - P_{Y|a,x}^* f_{P^*} \right)^2 \right) \\
&\leq 2 \frac{1-E}{E} P_X^* \left( x \mapsto \sum_{a \in \{1,0\}} e^*(a \mid x) \left( \widehat{P}_{Y|a,x} f_{\widehat{P}} - P_{Y|a,x}^* f_{P^*} \right)^2 \right) \\
&= 2 \frac{1-E}{E} \left\| \widehat{P}_{Y|A,X} f_{\widehat{P}} - P_{Y|A,X}^* f_{P^*} \right\|_{L^2(P^*)}^2 = o_p(1).
\end{aligned}
$$

The rate in the last equality follows from previous step. For the second additive term, because $\hat{e}(0 \mid X) - e^*(0 \mid X) = -(\hat{e}(1 \mid X) - e^*(1 \mid X))$, we have that

$$
\begin{aligned}
\left\| (\widehat{P}_{A|X} - P_{A|X}^*) P_{Y|A,X}^* f_{P^*} \right\|_{L^2(P_X^*)} &= \left\| x \mapsto \sum_{a \in \{1,0\}} (\hat{e}(a \mid x) - e^*(a \mid x)) P_{Y|a,x}^* f_{P^*} \right\|_{L^2(P_X^*)} \\
&= \left\| x \mapsto (\hat{e}(1 \mid x) - e^*(1 \mid x)) \left( P_{Y|1,x}^* - P_{Y|0,x}^* \right) f_{P^*} \right\|_{L^2(P_X^*)}
\end{aligned}
$$

Using Hölder's inequality,

$$
\begin{aligned}
\left\| (\widehat{P}_{A|X} - P_{A|X}^*) P_{Y|A,X}^* f_{P^*} \right\|_{L^2(P^*)} &\leq \left( \sup_{x \in \mathcal{X}} \left| \left( P_{Y|1,x}^* - P_{Y|0,x}^* \right) f_{P^*} \right| \right) \| \hat{e}(1 \mid \cdot) - e^*(1 \mid \cdot) \|_{L^2(P_X^*)} \\
&\leq 2 \sup_{a \in \{1,0\}, x \in \mathcal{X}} \left| \frac{P_{Y|a,x}^* v_a^*}{e^*(a \mid x)} \right| \| \hat{e}(1 \mid \cdot) - e^*(1 \mid \cdot) \|_{L^2(P_X^*)} \\
&\leq 2 \frac{C}{E} \| \hat{e}(1 \mid \cdot) - e^*(1 \mid \cdot) \|_{L^2(P_X^*)} = o_p(1).
\end{aligned}
$$

The last inequality follows from the uniform upper bound in $\widehat{v}_a$, the uniform strong positivity lower bound, and the first assumption.

**Step 4: Control** $\left| \widehat{P} f_{\widehat{P}} - P^* f_{P^*} \right|$. Decompose

$$
\left| \widehat{P} f_{\widehat{P}} - P^* f_{P^*} \right| \leq \left| \widehat{P} (f_{\widehat{P}} - f_{P^*}) \right| + \left| (\widehat{P} - P^*) f_{P^*} \right|.
$$

The first additive term is $o_p(1)$ from the $L^2(P^*)$ convergence of $f_{\widehat{P}} - f_{P^*}$ to zero established in Step 1 above. The second additive term is $o_p(1)$ by the weak law of large numbers, $\qquad \square$

**Lemma E.3** (Rate for first-order remainder term). *Let $P^* \notin \mathcal{H}_0$ and $\| n^r (\widehat{P}_a - P_a^*) \|_{\ell^\infty(\mathcal{H}_1)} = \mathcal{O}_p(1)$ for $a \in \{1,0\}$ and some $r > 1/4$. Further, for $a \in \{1,0\}$, assume that the estimated nuisance parameters satisfy*

$$
P_X^* \left[ \left( \frac{e^*(a \mid X)}{\hat{e}(a \mid X)} - 1 \right) \left( P_{Y \mid a,X}^* - \widehat{P}_{Y \mid a,X} \right) \widehat{v}_a \right] = o_p(n^{-1/2}).
$$

*Then, the remainder term satisfies*

$$\mathcal{R}_n = \mathcal{S}(\widehat{P}) + P^* \dot{\mathcal{S}}_{\widehat{P}} - \mathcal{S}(P^*) = o_p(n^{-1/2}).$$

*Proof.* Observe that

$$P^* \dot{\mathcal{S}}_{\widehat{P}} = \left( P^* - P^*_{A,X} \widehat{P}_{Y \mid A,X} + P^*_X \widehat{P}_{Y,A \mid X} - \widehat{P} \right) \left( (y,a,x) \mapsto \frac{a\,\widehat{v}_1}{\hat{e}(1 \mid x)} + \frac{(1-a)\,\widehat{v}_0}{\hat{e}(0 \mid x)} \right)$$

$$= P^*_X \left( \frac{e^*(1 \mid \cdot)}{\hat{e}(1 \mid \cdot)} P^*_{Y \mid 1,\cdot} \widehat{v}_1 + \frac{e^*(0 \mid \cdot)}{\hat{e}(0 \mid \cdot)} P^*_{Y \mid 0,\cdot} \widehat{v}_0 \right) - P^*_X \left( \frac{e^*(1 \mid \cdot)}{\hat{e}(1 \mid \cdot)} \widehat{P}_{Y \mid 1,\cdot} \widehat{v}_1 + \frac{e^*(0 \mid \cdot)}{\hat{e}(0 \mid \cdot)} \widehat{P}_{Y \mid 0,\cdot} \widehat{v}_0 \right)$$

$$+ P^*_X \left( \widehat{P}_{Y \mid 1,\cdot} \widehat{v}_1 + \widehat{P}_{Y \mid 0,\cdot} \widehat{v}_0 \right) - P_1 \widehat{v}_1 - P_0 \widehat{v}_0.$$

Add and subtract $P^*_1 \widehat{v}_1 + P^*_0 \widehat{v}_0$ on the RHS to obtain

$$P^* \dot{\mathcal{S}}_{\widehat{P}} = P^*_X \left( \frac{e^*(1 \mid \cdot)}{\hat{e}(1 \mid \cdot)} (P^*_{Y \mid 1,\cdot} - \widehat{P}_{Y \mid 1,\cdot}) \widehat{v}_1 + \frac{e^*(0 \mid \cdot)}{\hat{e}(0 \mid \cdot)} (P^*_{Y \mid 0,\cdot} - \widehat{P}_{Y \mid 0,\cdot}) \widehat{v}_0 \right)$$

$$- P^*_X \left( (P^*_{Y \mid 1,\cdot} - \widehat{P}_{Y \mid 1,\cdot}) \widehat{v}_1 + (P^*_{Y \mid 0,\cdot} - \widehat{P}_{Y \mid 0,\cdot}) \widehat{v}_0 \right) + (P^*_1 - P_1) \widehat{v}_1 + (P^*_0 - P_0) v_0$$

$$= (P^*_1 - P_1) \widehat{v}_1 + (P^*_0 - P_0) \widehat{v}_0$$

$$+ P^*_X \left( \left( \frac{e^*(1 \mid \cdot)}{\hat{e}(1 \mid \cdot)} - 1 \right) (P^*_{Y \mid 1,\cdot} - \widehat{P}_{Y \mid 1,\cdot}) \widehat{v}_1 + \left( \frac{e^*(0 \mid \cdot)}{\hat{e}(0 \mid \cdot)} - 1 \right) (P^*_{Y \mid 0,\cdot} - \widehat{P}_{Y \mid 0,\cdot}) \widehat{v}_0 \right)$$

From assumption, the second and third additive terms are $o_p(n^{-1/2})$. The assumption is valid because the term is a product of estimation error of two nuisance parameters - propensity score $e_{P^*}$ and outcome regression $P^*_{Y \mid A,X}$. The requirement of $o_p(n^{-1/2})$ convergence rate for this double product is standard. Therefore, now ignoring the established $o_p(n^{-1/2})$ term, it remains to show that

$$S_\varepsilon(\widehat{P}_1, \widehat{P}_0) - S_\varepsilon(P^*_1, P^*_0) - \left[ (P^*_1 - \widehat{P}_1) v_1 + (P^*_0 - \widehat{P}_0) v_0 \right] = o_p(n^{-1/2}). \tag{36}$$

The above is true via the Hadamard differentiability of $(\alpha, \beta) \mapsto S_\varepsilon(\alpha, \beta)$. For any $(\mu, \nu) \in \mathcal{P}(\mathcal{Y}) \times \mathcal{P}(\mathcal{Y})$ and $(\gamma^1, \gamma^2) \in \mathcal{M}_{0,\mu} \times \mathcal{M}_{0,\mu}$, consider paths of the form $\mu_t = \mu + t\gamma_t^1$ and $\nu_t = \nu + t\gamma_t^2$ such that $\gamma_t^1 \to \gamma_1$ and $\gamma_t^2 \to \gamma_2$ as $t \downarrow 0$ in $\ell^\infty(\mathcal{H}_1)$. Because $(\alpha, \beta) \mapsto S_\varepsilon(\alpha, \beta)$ is second-order Hadamard differentiable with a bounded second-order Hadamard derivative, the following first-order expansion is immediate from the definition

$$S_\varepsilon(\mu_t, \nu_t) - S_\varepsilon(\mu, \nu) - S'_\varepsilon[\mu, \nu](\mu_t - \mu, \nu_t - \nu) = O(t^2).$$

Plugging in the form of $S'_\varepsilon[\mu, \nu]$ from Lem. C.4, we have

$$S_\varepsilon(\mu_t, \nu_t) - S_\varepsilon(\mu, \nu) - \left[ (\mu_t - \mu) v_1^{(\mu,\nu)} + (\nu_t - \nu) v_0^{(\mu,\nu)} \right] = O(t^2).$$

Subtracting $t\gamma_t^1 v_1^{(\mu_t, \nu_t)} + t\gamma_t^2 v_0^{(\mu_t, \nu_t)}$ from both LHS and RHS of the above display, we have that

$$S_\varepsilon(\mu_t, \nu_t) - S_\varepsilon(\mu, \nu) - t\gamma_t^1 v_1^{(\mu_t, \nu_t)} - t\gamma_t^2 v_0^{(\mu_t, \nu_t)} = t\gamma_t^1 (v_1^{(\mu,\nu)} - v_1^{(\mu_t,\nu_t)}) + t\gamma_t^2 (v_0^{(\mu,\nu)} - v_0^{(\mu_t,\nu_t)}) + O(t^2).$$

From the Hadamard differentiability of the map $(\alpha, \beta) \mapsto (v_1^{(\alpha,\beta)}, v_0^{(\alpha,\beta)})$ at $(\mu, \nu)$ and boundedness of the derivative, we have that the RHS in the above display is $O(t^2)$. If $\|n^r(\widehat{P}_a - P^*_a)\|_{\ell^\infty(\mathcal{H}_1)} = \mathcal{O}_p(1)$ for $a \in \{1, 0\}$, then by functional delta method, we get that $\mathcal{R}_n$ is $\mathcal{O}_p(n^{-2r})$. This completes the proof for any $r > 1/4$. $\square$

### E.2. Limit theorem for second-order one-step estimator

This section proves Thm. 4.1 by proving that all three terms $\mathcal{D}_n$, $\mathcal{U}_n$, and $\mathcal{R}_n$ are $o_p(n^{-1})$. Throughout this section, for the sake of notational simplicity, we suppress superscripts and write $\hat{v}_1$ for $v_1^{(\widehat{P}_1, \widehat{P}_0)}$ and $\hat{v}_0$ for $v_0^{(\widehat{P}_1, \widehat{P}_0)}$. Similarly, we suppress the subscripts and write $\hat{e}$ for $e_{\widehat{P}}$ and $e^*$ for $e_{P^*}$. Recall the expectation operators defined in (7). Define the operators $Q_{\widehat{P}}$ and $Q_{P^*, \widehat{P}}$ as

$$Q_{\widehat{P}} := I - \widehat{P}_{Y \mid A, X} + \widehat{P}_{Y, A \mid X} - \widehat{P} \quad \text{and} \quad Q_{P^*, \widehat{P}} := P^* \circ Q_{\widehat{P}} = P^* - P^*_{A', X'} \widehat{P}_{Y' \mid A', X'} + P^*_{X'} \widehat{P}_{Y', A' \mid X'} - \widehat{P}. \quad (37)$$

First, we study the stability of the Sinkhorn operator.

**Lemma E.4** (Stability of Sinkhorn Operator). *Let $\mu \in \ell^\infty(\mathcal{H}_1)$ and $\hat{\mu}$ be an initial estimator of $\mu$ constructed using $n$ independent samples from $\mu$, and $\|n^r(\hat{\mu} - \mu)\|_{\ell^\infty(\mathcal{H}_1)} = \mathcal{O}_p(1)$ for some $r > 0$. Then,*

$$\|K_{\hat{\mu}} - K_\mu\|_{\mathcal{M}_0(\mathcal{Y}) \to \mathcal{C}(\mathcal{Y})/\mathbb{R}} = \mathcal{O}_p(n^{-r}).$$

*Proof.* Consider the following breakdown:

$$\begin{aligned}
\|K_{\hat{\mu}} - K_\mu\|_{\mathcal{M}_0(\mathcal{Y}) \to \mathcal{C}(\mathcal{Y})/\mathbb{R}} &= \left\|(I - T_{\hat{\mu}}^2)^{-1} H_{\hat{\mu}} - (I - T_\mu^2)^{-1} H_\mu\right\|_{\mathcal{M}_0(\mathcal{Y}) \to \mathcal{C}(\mathcal{Y})/\mathbb{R}} \\
&\leq \left\|(I - T_{\hat{\mu}}^2)^{-1}\right\|_{\mathcal{C}(\mathcal{Y})/\mathbb{R} \to \mathcal{C}(\mathcal{Y})/\mathbb{R}} \left\|(H_{\hat{\mu}} - H_\mu)\right\|_{\mathcal{M}_0(\mathcal{Y}) \to \mathcal{C}(\mathcal{Y})/\mathbb{R}} \\
&\quad + \left\|(I - T_{\hat{\mu}}^2)^{-1} - (I - T_\mu^2)^{-1}\right\|_{\mathcal{C}(\mathcal{Y})/\mathbb{R} \to \mathcal{C}(\mathcal{Y})/\mathbb{R}} \|H_\mu\|_{\mathcal{M}_0(\mathcal{Y}) \to \mathcal{C}(\mathcal{Y})/\mathbb{R}}.
\end{aligned}$$

Since centering (to find the representor element in $\mathcal{C}(\mathcal{Y})/\mathbb{R}$) is a contraction, we have that

$$\begin{aligned}
\|K_{\hat{\mu}} - K_\mu\|_{\mathcal{M}_0(\mathcal{Y}) \to \mathcal{C}(\mathcal{Y})/\mathbb{R}} &\leq \underbrace{\left\|(I - T_{\hat{\mu}}^2)^{-1}\right\|_{\mathcal{C}(\mathcal{Y})/\mathbb{R} \to \mathcal{C}(\mathcal{Y})/\mathbb{R}} \left\|(H_{\hat{\mu}} - H_\mu)\right\|_{\mathcal{M}_0(\mathcal{Y}) \to \mathcal{C}(\mathcal{Y})}}_{\textbf{(i)}} \\
&\quad + \underbrace{\left\|(I - T_{\hat{\mu}}^2)^{-1} - (I - T_\mu^2)^{-1}\right\|_{\mathcal{C}(\mathcal{Y})/\mathbb{R} \to \mathcal{C}(\mathcal{Y})/\mathbb{R}} \|H_\mu\|_{\mathcal{M}_0(\mathcal{Y}) \to \mathcal{C}(\mathcal{Y})}}_{\textbf{(ii)}}.
\end{aligned}$$

Let $\mathcal{G}$ be the Gaussian RKHS with kernel $g_\varepsilon : (y_1, y_2) \mapsto \exp(-\frac{1}{2\varepsilon}\|y_1 - y_2\|^2)$ and $\mathfrak{g}$ be the Gaussian KME operator: for any $\mu \in \mathcal{P}(\mathcal{Y})$, $\mathfrak{g}(\mu) := \int g_\varepsilon(y, \cdot)\, d\mu(y)$. Define the multiplicative operator $M_u : \ell^\infty(\mathcal{H}_1) \to \ell^\infty(\mathcal{H}_1)$ given by $M_u \mu = u\,\mu$. Let $b_\mu := \exp(\varphi_i^{(\mu, \mu)}/\varepsilon)$ for any $\mu \in \mathcal{P}(\mathcal{Y})$, then $\xi_\mu = (b_\mu \otimes b_\mu)\, g_\varepsilon$.

**Step 1: Control (i).** We can uniformly upperbound the operator norm $\|(I - T_{\hat{\mu}}^2)^{-1}\|_{\mathcal{C}(\mathcal{Y})/\mathbb{R} \to \mathcal{C}(\mathcal{Y})/\mathbb{R}}$ using uniform bounds on the spectrum of $T_{\hat{\mu}} : \mathcal{C}(\mathcal{Y})/\mathbb{R} \to \mathcal{C}(\mathcal{Y})/\mathbb{R}$ in Lavenant et al. (2024, Prop. 3.7) and the Neumann series expansion of $(I - T_{\hat{\mu}}^2)^{-1}$. This shows that there exists $q \in (0, 1)$ such that $\|(I - T_\nu^2)^{-1}\| \leq 1/(1 - q^2)$ on $\mathcal{C}(\mathcal{Y})/\mathbb{R}$ for all $\nu$. Now it remains to control the term $\|H_{\hat{\mu}} - H_\mu\|_{\mathcal{M}_0(\mathcal{Y})}$. First, trivially, $\|H_{\hat{\mu}} - H_\mu\|_{\mathcal{M}_0(\mathcal{Y}) \to \mathcal{C}(\mathcal{Y})} \leq \|H_{\hat{\mu}} - H_\mu\|_{\mathcal{M}(\mathcal{Y}) \to \mathcal{C}(\mathcal{Y})}$. Then, for any $\gamma \in \mathcal{M}(\mathcal{Y})$,

$$\begin{aligned}
\|H_{\hat{\mu}}\gamma - H_\mu\gamma\|_{\mathcal{C}(\mathcal{Y})} &= \left\|b_{\hat{\mu}}\mathfrak{g}(M_{b_{\hat{\mu}}}\gamma) - b_\mu\mathfrak{g}(M_{b_\mu}\gamma)\right\|_{\mathcal{C}(\mathcal{Y})} \\
&\leq \|b_{\hat{\mu}} - b_\mu\|_{\mathcal{C}(\mathcal{Y})} \left\|\mathfrak{g}(M_{b_{\hat{\mu}}}\gamma)\right\|_{\mathcal{C}(\mathcal{Y})} + \|b_\mu\|_{\mathcal{C}(\mathcal{Y})} \left\|\mathfrak{g}(M_{b_{\hat{\mu}}}\gamma) - \mathfrak{g}(M_{b_\mu}\gamma)\right\|_{\mathcal{C}(\mathcal{Y})} \\
&\leq \|b_{\hat{\mu}} - b_\mu\|_{\mathcal{C}(\mathcal{Y})} \left\|\mathfrak{g}(M_{b_{\hat{\mu}}}\gamma)\right\|_{\mathcal{G}} + \|b_\mu\|_{\mathcal{C}(\mathcal{Y})} \left\|\mathfrak{g}(M_{b_{\hat{\mu}}}\gamma) - \mathfrak{g}(M_{b_\mu}\gamma)\right\|_{\mathcal{G}}.
\end{aligned}$$

The last inequality follows from the fact that, for any $f \in \mathcal{G}$, $\|f\|_{\mathcal{C}(\mathcal{Y})} = \sup_y |f(y)| \leq \|f\|_{\mathcal{G}} \sup_y \sqrt{g_\varepsilon(y, y)} = \|f\|_{\mathcal{G}}$. We tackle the terms above separately.

- **Controlling** $\left\|\mathfrak{g}(M_{b_{\hat{\mu}}}\gamma)\right\|_{\mathcal{G}}$. We can bound the $\ell^\infty(\mathcal{G}_1)$ norm of measures $\{M_{b_\nu}\gamma, \nu \in \mathcal{P}(\mathcal{Y})\}$ by a uniform constant

times $\|\gamma\|_{\ell^\infty(\mathcal{G}_1)}$. Consider the set of functions in $\mathcal{C}(\mathcal{Y})$, $U := \{b_\nu : \nu \in \mathcal{P}(\mathcal{Y})\}$. We know that $\{b_\nu : \nu \in \mathcal{P}(\mathcal{Y})\}$ is a uniformly bounded subset of $\mathcal{C}(\mathcal{Y})$ in supnorm (Goldfeld et al., 2024, Lem. 1). Let $C = \sup_\nu \|b_\nu\|_{\mathcal{C}(\mathcal{Y})}$. We now show that $U$ is a compact subset of $\mathcal{C}(\mathcal{Y})$. Indeed, $\mathcal{P}(\mathcal{Y})$ is tight, and hence Prokhorov's theorem (Billingsley, 2013, Thm. 5.1) gives relative compactness. Since weak limits of probability measures on $\mathcal{Y}$ are again probability measures, $\mathcal{P}(\mathcal{Y})$ is closed, and therefore compact. The map $\nu \mapsto b_\nu$ is continuous from $\mathcal{P}(\mathcal{Y})$ (equipped with the topology of weak convergence) into $\mathcal{C}(\mathcal{Y})$ (Goldfeld et al., 2024, Lem. 1). Therefore, it follows that $U$ is the continuous image of a compact set. Hence $U$ is compact in $\mathcal{C}(\mathcal{Y})$. At this point, we can use Kokot & Luedtke (2025, Lem. 22) to show that there exists a constant $C_1 > 0$ such that for all $\gamma \in \mathcal{M}(\mathcal{Y}) \subset \ell^\infty(\mathcal{G}_1)$ and $\nu \in \mathcal{P}(\mathcal{Y})$,

$$\|M_{b_\nu}\gamma\|_{\ell^\infty(\mathcal{G}_1)} \leq C_1\|\gamma\|_{\ell^\infty(\mathcal{G}_1)}.$$

Therefore, $\left\|\mathfrak{g}(M_{b_{\hat{\mu}}}\gamma)\right\|_{\mathcal{G}} \leq C_1\|\gamma\|_{\ell^\infty(\mathcal{G}_1)}$.

- **Controlling $\|b_{\hat{\mu}} - b_\mu\|_{\mathcal{C}(\mathcal{Y})}$.** From Goldfeld et al. (2024, Thm. 3), we know that the map $\Phi : \mathcal{P}(\mathcal{Y}) \times \mathcal{P}(\mathcal{Y}) \to \mathcal{C}(\mathcal{Y}) \times \mathcal{C}(\mathcal{Y})$ given by $(\mu, \nu) \mapsto (\varphi_1^{(\mu,\nu)}, \varphi_0^{(\mu,\nu)})$ is Hadamard differentiable. Further the function $b : \mathcal{C}(\mathcal{Y}) \to \mathcal{C}(\mathcal{Y})$ given by $u \mapsto \exp(u/\varepsilon)$ is Fréchet differentiable on $\mathcal{C}(\mathcal{Y})$ with derivative given by the linear operator $b'[u](h) = \frac{1}{\varepsilon}\exp(u/\varepsilon)h$. Hence $b$ is Hadamard differentiable and by chain rule, the mapping $(\mu, \nu) \mapsto (\exp(\varphi_1^{(\mu,\nu)}/\varepsilon), \exp(\varphi_0^{(\mu,\nu)}/\varepsilon))$ is Hadamard differentiable.

This implies that the function $\mu \mapsto b_\mu$ is Hadamard differentiable and there exists a continuous linear operator $b'[\mu] : \mathcal{C}(\mathcal{Y}) \to \mathcal{C}(\mathcal{Y})$ such that along the path $\mu_t = \mu + t\gamma_t$ with $\gamma_t \to \gamma$ in $\ell^\infty(\mathcal{H}_1)$ for some $\gamma \in \mathcal{M}_{0,\mu}$,

$$\lim_{t \to 0} \frac{\|b_{\mu_t} - b_\mu - tb'[\mu](\gamma_t)\|_{\mathcal{C}(\mathcal{Y})}}{t} = 0.$$

By functional delta theorem, in supremum norm,

$$n^r(b_{\hat{\mu}} - b_\mu) = b'[\mu](n^r(\hat{\mu} - \mu)) + o_p(1).$$

Because $b'[\mu]$ is a tight continuous operator, $b'[\mu](n^r(\hat{\mu} - \mu))$ is tight, and therefore $\|(b_{\hat{\mu}} - b_\mu)\|_{\mathcal{C}(\mathcal{Y})} = \mathcal{O}_p(n^{-r})$.

- **Controlling $\left\|\mathfrak{g}(M_{b_{\hat{\mu}}}\gamma) - \mathfrak{g}(M_{b_\mu}\gamma)\right\|_{\mathcal{G}}$.** By Kokot & Luedtke (2025, Cor. 23), $\|b_{\hat{\mu}} - b_\mu\|_{\mathcal{C}(\mathcal{Y})} = \mathcal{O}_p(n^{-r})$ immediately implies that $\left\|M_{b_{\hat{\mu}}} - M_{b_\mu}\right\|_{\ell^\infty(\mathcal{G}_1) \to \ell^\infty(\mathcal{G}_1)} = \mathcal{O}_p(n^{-r})$. As a consequence, $\left\|\mathfrak{g}(M_{b_{\hat{\mu}}}\gamma) - \mathfrak{g}(M_{b_\mu}\gamma)\right\|_{\mathcal{G}} = \mathcal{O}_p(n^{-r})\|\gamma\|_{\ell^\infty(\mathcal{G}_1)}$.

Altogether, term (i) is $\mathcal{O}_p(n^{-r})\|\gamma\|_{\ell^\infty(\mathcal{G}_1)}$.

**Step 2: Control (ii).** We bound $\|H_\mu\|_{\mathcal{M}_0(\mathcal{Y}) \to \mathcal{C}(\mathcal{Y})}$ as following. For any $\gamma \in \mathcal{M}_0(\mathcal{Y}) \subset \ell^\infty(\mathcal{G}_1)$,

$$\langle \gamma, H_\mu \gamma \rangle = \|M_{b_\mu}\gamma\|_{\ell^\infty(\mathcal{G}_1)}^2 \leq C_1^2\|\gamma\|_{\ell^\infty(\mathcal{G}_1)}^2.$$

Therefore, the operator norm must be uniformly bounded. For the difference term, we consider the following identity

$$\begin{aligned}
\left\|(I - T_{\hat{\mu}}^2)^{-1} - (I - T_\mu^2)^{-1}\right\| &= \left\|(I - T_{\hat{\mu}}^2)^{-1}\left[T_\mu^2 - T_{\hat{\mu}}^2\right](I - T_\mu^2)^{-1}\right\| \\
&\leq \left\|(I - T_{\hat{\mu}}^2)^{-1}\right\|\left\|(I - T_\mu^2)^{-1}\right\|\left\|T_{\hat{\mu}}^2 - T_\mu^2\right\| \\
&\leq \left\|(I - T_{\hat{\mu}}^2)^{-1}\right\|\left\|(I - T_\mu^2)^{-1}\right\|(\|T_{\hat{\mu}}\| + \|T_\mu\|)\|T_{\hat{\mu}} - T_\mu\|,
\end{aligned}$$

where all operator norms are on the common function space $\mathcal{C}(\mathcal{Y})/\mathbb{R}$. As in Step 1, the operator norm of $(I - T_{\hat{\mu}}^2)^{-1}$ is uniformly bounded on $\mathcal{C}(\mathcal{Y})/\mathbb{R}$. Lavenant et al. (2024, Prop. 3.7) show that the operator norm of $T_\nu$ is uniformly bounded by 1 for all $\nu \in \mathcal{P}(\mathcal{Y})$. Since centering is a contraction, $\|T_{\hat{\mu}} - T_\mu\|_{\mathcal{C}(\mathcal{Y})/\mathbb{R} \to \mathcal{C}(\mathcal{Y})/\mathbb{R}} \leq \|T_{\hat{\mu}} - T_\mu\|_{\mathcal{C}(\mathcal{Y}) \to \mathcal{C}(\mathcal{Y})}$. For any $f$

inside a unit ball in $\mathcal{C}(\mathcal{Y})$,

$$
\begin{aligned}
\|(T_{\hat{\mu}} - T_\mu)f\|_{\mathcal{C}(\mathcal{Y})} &\leq \|H_{\hat{\mu}} M_f \hat{\mu} - H_\mu M_f \mu\|_{\mathcal{C}(\mathcal{Y})} \\
&\leq \|(H_{\hat{\mu}} - H_\mu)M_f \hat{\mu} + H_\mu M_f(\hat{\mu} - \mu)\|_{\mathcal{C}(\mathcal{Y})} \\
&\leq \|H_{\hat{\mu}} - H_\mu\|_{\mathcal{M}(\mathcal{Y}) \to \mathcal{C}(\mathcal{Y})} \|M_f \hat{\mu}\|_{\ell^\infty(\mathcal{G}_1)} + \|H_\mu\|_{\mathcal{M}(\mathcal{Y}) \to \mathcal{C}(\mathcal{Y})} \|M_f(\hat{\mu} - \mu)\|_{\ell^\infty(\mathcal{G}_1)}.
\end{aligned}
$$

Since $f$ belongs to a compact subset in $\mathcal{C}(\mathcal{Y})$, again by Kokot & Luedtke (2025, Lem. 22), $\|M_f \gamma\|_{\ell^\infty(\mathcal{G}_1)} \leq C_1 \|\gamma\|_{\ell^\infty(\mathcal{G}_1)}$. Using this and the fact that $\|H_\mu\|_{\mathcal{M}_0(\mathcal{Y}) \to \mathcal{C}(\mathcal{Y})} \leq C_1^2$,

$$
\|(T_{\hat{\mu}} - T_\mu)f\|_{\mathcal{C}(\mathcal{Y})} \lesssim \|H_{\hat{\mu}} - H_\mu\|_{\mathcal{M}(\mathcal{Y}) \to \mathcal{C}(\mathcal{Y})} + \|\hat{\mu} - \mu\|_{\ell^\infty(\mathcal{G}_1)} = \mathcal{O}_p(n^{-r}).
$$

Together, terms (i) and (ii) are $\mathcal{O}_p(n^{-r})$ and this completes the proof. $\qquad\square$

**Lemma E.5** (Rate for second-order drift term). *Let $P^* \in \mathcal{H}_0$ and assume that $\|n^r(\hat{\mu} - \mu)\|_{\ell^\infty(\mathcal{H}_1)} = \mathcal{O}_p(1)$ for some $r > 1/4$. Further, assume that the estimated nuisance parameters satisfy*

$$
P_X^* \left[ x \mapsto \left( \frac{e_{\widehat{P}}(a \mid x)}{e_{P^*}(a \mid x)} - 1 \right) \left( \widehat{P}_{Y \mid a, x} - P_{Y \mid a, x}^* \right) k_{\widehat{P}_1}(\cdot, y) \right] = o_p(n^{-1/2}),
$$

*uniformly in $y \in \mathcal{Y}$ and $a \in \{1, 0\}$. Then the drift term satisfies*

$$
\mathcal{D}_n = (P_n - P^*)\left( \dot{\mathbb{S}}_{\widehat{P}}(\cdot) + \int \ddot{\mathbb{S}}_{\widehat{P}}(z, \cdot) \, dP^*(z) \right) = o_p(n^{-1}).
$$

*Proof.* Since $\widehat{P}$ and $P_n$ are constructed from independent samples, the random function $\dot{\mathbb{S}}_{\widehat{P}}(\cdot) + \int \ddot{\mathbb{S}}_{\widehat{P}}(\cdot, z) \, dP^*(z)$ is independent of $P_n$. By Lem. E.1, it suffices to show that $\left\|\dot{\mathbb{S}}_{\widehat{P}}(\cdot) + \int \ddot{\mathbb{S}}_{\widehat{P}}(\cdot, z) \, dP^*(z)\right\|_{L^2(P^*)} = o_p(n^{-1/2})$ to show that $\mathcal{D}_n = o_p(n^{-1})$. Recall from (8) and (10) the formulations of $\dot{\mathbb{S}}_{\widehat{P}}$ and $\ddot{\mathbb{S}}_{\widehat{P}}$. Using $Q$ operator notation in (37), at evaluation points $z_j = (x_j, a_j, y_j)$ for $j = 1, 2$ in $\mathcal{Z}$, we have that

$$
\begin{aligned}
\dot{\mathbb{S}}_{\widehat{P}} &= Q_{\widehat{P}} f_{\widehat{P}}, \quad f_{\widehat{P}}(z_1) = \frac{a_1}{\hat{e}(1 \mid x_1)} \widehat{v}_1(y_1) + \frac{1 - a_1}{\hat{e}(0 \mid x_1)} \widehat{v}_0(y_1), \\
\ddot{\mathbb{S}}_{\widehat{P}} &= Q_{\widehat{P}}^{\otimes 2} g_{\widehat{P}}, \quad g_{\widehat{P}}(z_1, z_2) = \omega_{\widehat{P}}(a_1, x_1) \, \omega_{\widehat{P}}(a_2, x_2) \, k_{\widehat{P}_1}(y_1, y_2),
\end{aligned}
$$

where $\omega_{\widehat{P}}(a, x) = \frac{a}{\hat{e}(1 \mid x)} - \frac{1 - a}{\hat{e}(0 \mid x)}$. Note that, with $k_{\widehat{P}_1}(y, \cdot) \otimes \omega_{\widehat{P}} : (y', a', x') \mapsto k_{\widehat{P}_1}(y, y') \otimes \omega_{\widehat{P}}(a', x')$,

$$
\begin{aligned}
\dot{\mathbb{S}}_{\widehat{P}}(\cdot) + \int \ddot{\mathbb{S}}_{\widehat{P}}(z, \cdot) \, dP^*(z) &= Q_{\widehat{P}} f_{\widehat{P}} + (Q_{\widehat{P}} \otimes Q_{P^*, \widehat{P}}) g_{\widehat{P}} \\
&= Q_{\widehat{P}} \left[ (y, a, x) \mapsto \frac{a}{\hat{e}(1 \mid x)} \widehat{v}_1(y) + \frac{1 - a}{\hat{e}(0 \mid x)} \widehat{v}_0(y) + \omega_{\widehat{P}}(a, x) \, Q_{P^*, \widehat{P}}\left( k_{\widehat{P}_1}(y, \cdot) \otimes \omega_{\widehat{P}} \right) \right] \\
&= Q_{\widehat{P}} \left[ (y, a, x) \mapsto \frac{a}{\hat{e}(1 \mid x)} \widehat{v}_1(y) + \frac{1 - a}{\hat{e}(0 \mid x)} \widehat{v}_0(y) + \left( \frac{a}{\hat{e}(1 \mid x)} - \frac{1 - a}{\hat{e}(0 \mid x)} \right) Q_{P^*, \widehat{P}}\left( k_{\widehat{P}_1}(y, \cdot) \otimes \omega_{\widehat{P}} \right) \right] \\
&= Q_{\widehat{P}} \left[ (y, a, x) \mapsto \frac{a}{\hat{e}(1 \mid x)} \zeta_1(y) \right] + Q_{\widehat{P}} \left[ (y, a, x) \mapsto \frac{1 - a}{\hat{e}(0 \mid x)} \zeta_2(y) \right],
\end{aligned}
$$

where

$$
\zeta_1(y) := \widehat{v}_1(y) + Q_{P^*, \widehat{P}}\left( k_{\widehat{P}_1}(y, \cdot) \otimes \omega_{\widehat{P}} \right) \quad \text{and} \quad \zeta_2(y) := \widehat{v}_0(y) - Q_{P^*, \widehat{P}}\left( k_{\widehat{P}_1}(y, \cdot) \otimes \omega_{\widehat{P}} \right).
$$

Expanding the form of $Q_{\widehat{P}}$, we have that the above is equal to

$$
\begin{aligned}
\dot{\mathcal{S}}_{\widehat{P}}(z) + \int \ddot{\mathcal{S}}_{\widehat{P}}(z', z)\, dP^*(z') = {} & \frac{a}{\hat{e}(1\,|\,x)}(I - \widehat{P}_{Y\,|\,A,X})\,[(y', a', x') \mapsto \zeta_1(y')]\,(z) \\
& + \frac{1-a}{\hat{e}(0\,|\,x)}(I - \widehat{P}_{Y\,|\,A,X})\,[(y', a', x') \mapsto \zeta_2(y')]\,(z) \\
& + (I - \widehat{P}_X)\widehat{P}_{Y\,|\,1,X}\,[(y', a', x') \mapsto \zeta_1(y')]\,(z) \\
& + (I - \widehat{P}_X)\widehat{P}_{Y\,|\,0,X}\,[(y', a', x') \mapsto \zeta_2(y')]\,(z).
\end{aligned}
$$

Note that if $\|\zeta_1\|_\infty = \sup_{y\in\mathcal{Y}} |\zeta_1(y)| = o_p(n^{-1/2})$, then

$$
\begin{aligned}
& \left\| z \mapsto \frac{a}{\hat{e}(1\,|\,x)}(I - \widehat{P}_{Y\,|\,A,X})\,[z' \mapsto \zeta_1(y')]\,(z) \right\|_{L^2(P^*)}^2 \\
& = \int \frac{a^2}{\hat{e}(1\,|\,x)^2}\left( \zeta_1(y) - \int \zeta_1(y')\hat{p}_{Y\,|\,A,X}(y'\,|\,a, x)\, dy' \right)^2 dP^*(z) \\
& \le 2\|\zeta_1\|_\infty^2 \int \frac{a^2}{\hat{e}(1\,|\,x)^2}\, dP^*(x, a, y) \le \frac{2}{E}\|\zeta_1\|_\infty^2,
\end{aligned}
$$

and

$$
\left\| (I - \widehat{P}_X)\widehat{P}_{Y\,|\,1,X}\,[z' \mapsto \zeta_1(y')] \right\|_{L^2(P^*)} \le \left\| \widehat{P}_{Y\,|\,1,X}\,[z' \mapsto \zeta_1(y')] \right\|_{L^2(P^*)} + \left\| \widehat{P}_X\widehat{P}_{Y\,|\,1,X}\,[z' \mapsto \zeta_1(y')] \right\|_{L^2(P^*)}
$$

$$
\le 2\|\zeta_1\|_\infty.
$$

Similarly, the same upperbounds hold for terms involving $\zeta_2$, and therefore,

$$
\left\| \dot{\mathcal{S}}_{\widehat{P}}(\cdot) + \int \ddot{\mathcal{S}}_{\widehat{P}}(z', \cdot)\, dP^*(z') \right\|_{L^2(P^*)} \lesssim \|\zeta_1\|_\infty + \|\zeta_2\|_\infty.
$$

It remains to show that both $\|\zeta_1\|_\infty$ and $\|\zeta_2\|_\infty$ are $o_p(n^{-1/2})$. For any $y$, $\zeta_1(y)$ can be simplified as follows:

$$
\begin{aligned}
\zeta_1(y) = {} & \widehat{v}_1(y) \\
& + \left( P^* - P^*_{A',X'}\widehat{P}_{Y'\,|\,A',X'} + P^*_{X'}\widehat{P}_{Y',A'\,|\,X'} - \widehat{P} \right)\left[ (y', a', x') \mapsto \left( \frac{a'}{\hat{e}(1\,|\,x')} - \frac{1-a'}{\hat{e}(0\,|\,x')} \right) k_{\widehat{P}_1}(y, y') \right] \\
= {} & \widehat{v}_1(y) + P^*_{X'}\left[ x' \mapsto \frac{e^*(1\,|\,x')}{\hat{e}(1\,|\,x')} P^*_{Y'\,|\,1,x'} k_{\widehat{P}_1}(y, \cdot) - \frac{e^*(0\,|\,x')}{\hat{e}(0\,|\,x')} P^*_{Y'\,|\,0,x'} k_{\widehat{P}_1}(y, \cdot) \right] \\
& - P^*_{X'}\left[ x' \mapsto \frac{e^*(1\,|\,x')}{\hat{e}(1\,|\,x')} \widehat{P}_{Y'\,|\,1,x'} k_{\widehat{P}_1}(y, \cdot) - \frac{e^*(0\,|\,x')}{\hat{e}(0\,|\,x')} \widehat{P}_{Y'\,|\,0,x'} k_{\widehat{P}_1}(y, \cdot) \right] \\
& + P^*_{X'}\left[ x' \mapsto \widehat{P}_{Y'\,|\,1,x'} k_{\widehat{P}_1}(y, \cdot) - \widehat{P}_{Y'\,|\,0,x'} k_{\widehat{P}_1}(y, \cdot) \right] - (\widehat{P}_1 - \widehat{P}_0)\, k_{\widehat{P}_1}(y, \cdot) \\
= {} & \widehat{v}_1(y) - K_{\widehat{P}_1}(\widehat{P}_1 - \widehat{P}_0)(y) \\
& + P^*_{X'}\left[ x' \mapsto \left( \frac{e^*(1\,|\,x')}{\hat{e}(1\,|\,x')} - 1 \right) P^*_{Y'\,|\,1,x'} k_{\widehat{P}_1}(y, \cdot) - \left( \frac{e^*(0\,|\,x')}{\hat{e}(0\,|\,x')} - 1 \right) P^*_{Y'\,|\,0,x'} k_{\widehat{P}_1}(y, \cdot) \right] \\
& - P^*_{X'}\left[ x' \mapsto \left( \frac{e^*(1\,|\,x')}{\hat{e}(1\,|\,x')} - 1 \right) \widehat{P}_{Y'\,|\,1,x'} k_{\widehat{P}_1}(y, \cdot) - \left( \frac{e^*(0\,|\,x')}{\hat{e}(0\,|\,x')} - 1 \right) \widehat{P}_{Y'\,|\,0,x'} k_{\widehat{P}_1}(y, \cdot) \right] \\
& + (P^*_1 - P^*_0)k_{\widehat{P}_1}(y, \cdot)
\end{aligned}
$$

Since the null holds by assumption, $P_1^* = P_0^*$, and so the last additive term is zero. Combining the terms above

$$\zeta_1(y) = \widehat{v}_1(y) - K_{\widehat{P}_1}(\widehat{P}_1 - \widehat{P}_0)(y)$$
$$+ P_{X'}^* \left[ x' \mapsto \left( \frac{e^*(1 \mid x')}{\widehat{e}(1 \mid x')} - 1 \right) \left( P_{Y' \mid 1, x'}^* - \widehat{P}_{Y' \mid 1, x'} \right) k_{\widehat{P}_1}(y, \cdot) \right]$$
$$- P_{X'}^* \left[ x' \mapsto \left( \frac{e^*(0 \mid x')}{\widehat{e}(0 \mid x')} - 1 \right) \left( P_{Y' \mid 0, x'}^* - \widehat{P}_{Y' \mid 0, x'} \right) k_{\widehat{P}_1}(y, \cdot) \right].$$

A similar calculation yields

$$\zeta_2(y) = \widehat{v}_0(y) + K_{\widehat{P}_1}(\widehat{P}_1 - \widehat{P}_0)(y)$$
$$- P_{X'}^* \left[ x' \mapsto \left( \frac{e^*(1 \mid x')}{\widehat{e}(1 \mid x')} - 1 \right) \left( P_{Y' \mid 1, x'}^* - \widehat{P}_{Y' \mid 1, x'} \right) k_{\widehat{P}_1}(y, \cdot) \right]$$
$$+ P_{X'}^* \left[ x' \mapsto \left( \frac{e^*(0 \mid x')}{\widehat{e}(0 \mid x')} - 1 \right) \left( P_{Y' \mid 0, x'}^* - \widehat{P}_{Y' \mid 0, X'} \right) k_{\widehat{P}_1}(y, \cdot) \right].$$

The last two terms in both $\zeta_1$ and $\zeta_2$ involve products of estimation errors for propensity score $e_{P^*}$ and the outcome regression $P_{Y \mid A, X}^*$. By assumption, these terms are $o_p(n^{-1/2})$ uniformly over $y \in \mathcal{Y}$, and hence negligible. It therefore remains to control the leading components: $\widehat{v}_1 - K_{\widehat{P}_1}(\widehat{P}_1 - \widehat{P}_0)$ and $\widehat{v}_0 + K_{\widehat{P}_1}(\widehat{P}_1 - \widehat{P}_0)$. Consider the breakdown

$$\widehat{v}_1 - K_{\widehat{P}_1}(\widehat{P}_1 - \widehat{P}_0) = \widehat{v}_1 - K_{P_1^*}(\widehat{P}_1 - \widehat{P}_0) + (K_{P_1^*} - K_{\widehat{P}_1})(\widehat{P}_1 - P_1^*) - (K_{P_1^*} - K_{\widehat{P}_1})(\widehat{P}_0 - P_0^*),$$

and

$$\widehat{v}_0 + K_{\widehat{P}_1}(\widehat{P}_1 - \widehat{P}_0) = \widehat{v}_0 + K_{P_1^*}(\widehat{P}_1 - \widehat{P}_0) - (K_{P_1^*} - K_{\widehat{P}_1})(\widehat{P}_1 - P_1^*) + (K_{P_1^*} - K_{\widehat{P}_1})(\widehat{P}_0 - P_0^*).$$

From Lem. E.4, we have that for each $i, j \in \{1, 0\}$

$$\left\| (K_{\widehat{P}_i} - K_{P_i^*})(\widehat{P}_j - P_j^*) \right\|_{\mathcal{C}(\mathcal{Y})} \leq \left\| K_{\widehat{P}_i} - K_{P_i^*} \right\|_{\mathcal{M}_0(\mathcal{Y}) \to \mathcal{C}(\mathcal{Y})/\mathbb{R}} \left\| \widehat{P}_j - P_j^* \right\|_{\ell^\infty(\mathcal{H})} = \mathcal{O}_p(n^{-2r}).$$

Since $r > 1/4$, we have that $(K_{\widehat{P}_1} - K_{P_1^*})(\widehat{P}_a - P_a^*)$ is $o_p(n^{-1/2})$ in the $L^2(P^*)$ sense. Finally, by Cor. C.7, the quantities $\widehat{v}_1 - K_{P_1^*}(\widehat{P}_1 - \widehat{P}_0)$ and $\widehat{v}_0 + K_{P_1^*}(\widehat{P}_1 - \widehat{P}_0)$ constitute the second-order remainder terms in the Hadamard expansion of the centered entropic potentials. From assumption, $\|n^r(\widehat{P}_1 - P_1^*)\|_{\ell^\infty(\mathcal{H}_1)}$ and $\|n^r(\widehat{P}_0 - P_0^*)\|_{\ell^\infty(\mathcal{H}_1)}$ are $\mathcal{O}_p(1)$ for some $r > 1/4$. Then by functional delta theorem and second-order Hadamard differentiability of entropic potentials (Goldfeld et al., 2024, Theorem 4), these remainders are $\mathcal{O}_p(n^{-2r})$ in $\mathcal{C}(\mathcal{Y})$ topology, and hence $o_p(n^{-1/2})$ in $L^2(P^*)$ norm. $\qquad \square$

**Lemma E.6** (Rate for second-order U-process term). *If* $\|C_{P^*}\ddot{\mathcal{S}}_{\widehat{P}} - C_{P^*}\ddot{\mathcal{S}}_{P^*}\|_{L^2(P^* \otimes P^*)} \xrightarrow{p} 0$, *then the U-process term*

$$\mathcal{U}_n = \frac{1}{2}\mathbb{U}_n \left( C_{P^*}\ddot{\mathcal{S}}_{\widehat{P}} - C_{P^*}\ddot{\mathcal{S}}_{P^*} \right) = o_p(n^{-1}).$$

*Proof.* Denote $f_n := C_{P^*}\ddot{\mathcal{S}}_{\widehat{P}} - C_{P^*}\ddot{\mathcal{S}}_{P^*}$. We will show that for every $\delta > 0$,

$$\mathbb{P}\left( |n\mathbb{U}_n f_n| > \delta \right) \leq \frac{C}{\delta^2} \|f_n\|_{L^2(P^* \otimes P^*)}^2.$$

Because $C_{P^*}\ddot{\mathcal{S}}_{\widehat{P}}$ is $P^*$-degenerate, $\mathbb{E}_{P^*}\left[ U_n f_n \mid \widehat{P} \right] = 0$. Let $Z_1, \ldots, Z_n$ be the iid samples from $P^*$ used to construct $\widehat{P}$.

By Chebyshev's inequality,

$$\mathbb{P}\left(|n\mathbb{U}_n f_n| > \delta \,|\, \widehat{P}\right) \leq \frac{\mathbb{E}\left[(n\mathbb{U}_n f_n)^2 \,|\, \widehat{P}\right]}{\delta^2}$$

$$= \frac{n^2}{\delta^2}\mathbb{E}\left[\left(\binom{n}{2}^{-1}\sum_{i<j} f_n(Z_i, Z_j)\right)^2 \Big|\, \widehat{P}\right]$$

$$= \frac{2n}{\delta^2(n-1)}\mathbb{E}\left[f_n(Z_1, Z_2)^2 \,|\, \widehat{P}\right] \leq \frac{C}{\delta^2}\|f_n\|_{L^2(P^*\otimes P^*)}^2 \quad \text{a.s. for some } C > 0.$$

Taking expectation on both sides,

$$\mathbb{P}\left(|n\mathbb{U}_n f_n| > \delta\right) = \mathbb{E}\left[\mathbb{P}\left(|n\mathbb{U}_n f_n| > \delta \,|\, \widehat{P}\right)\right] \leq \mathbb{E}\left[\min\left(1, \frac{C}{\delta^2}\|f_n\|_{L^2(P^*\otimes P^*)}^2\right)\right].$$

Since $\|f_n\|_{L^2(P^*\otimes P^*)}^2 \xrightarrow{p} 0$, the integrand on the right converges to $0$ in probability. Since it is also dominated by the constant function $1$, by the dominated convergence theorem, the right-hand side converges to $0$ as $n \to \infty$. $\qquad\square$

**Lemma E.7** (Second-order remainder for the Sinkhorn divergence). *Consider paths $\mu_t = \mu + t\gamma_t^1$ and $\nu_t = \mu + t\gamma_t^2$ in $\mathcal{P}_\mu$ such that $\gamma_t^i \to \gamma^i$ in $\ell^\infty(\mathcal{H}_1)$, for $i = 1, 2$, with $\gamma^1, \gamma^2 \in \mathcal{M}_{0,\mu}$. Suppose that $S_\varepsilon : (\mu, \nu) \mapsto S_\varepsilon(\mu, \nu)$ is twice differentiable along the path $t \mapsto (\mu + t\gamma_t^1, \mu + t\gamma_t^2)$ for $t$ small enough, and its second derivative satisfies the local Lipschitz bound*

$$|S_\varepsilon''[\mu_t, \nu_t](h_1, h_2) - S_\varepsilon''[\mu_t, \mu_t](h_1, h_2)| \leq C_1\|h_1 - h_2\|_{\ell^\infty(\mathcal{H}_1)}^2\|\nu_t - \mu_t\|_{\ell^\infty(\mathcal{H}_1)}$$

*for all $h_1, h_2$ in $\ell^\infty(\mathcal{H}_1)$ and some constant $C_1 > 0$. Then*

$$S_\varepsilon(\mu_t, \nu_t) - \frac{t^2}{2}(\gamma_t^1 - \gamma_t^2)K_\mu(\gamma_t^1 - \gamma_t^2) = O(t^3).$$

*Proof.* Define $g_t(s) = S_\varepsilon(\mu + st\gamma_t^1, \mu + st\gamma_t^2)$, where $\mu + ts\gamma_t^1$ and $\mu + ts\gamma_t^2$ belong to $\mathcal{P}(\mathcal{Y})$ for any $s \in [0, 1]$ because $\mu + ts\gamma_t^1 = (1-s)\mu + s\mu_t$ and same argument for $\mu + st\gamma_t^2$. Then by the first and second order Hadamard differentiability of $S_\varepsilon : (\mu, \nu) \mapsto S_\varepsilon(\mu, \nu)$,

$$g_t'(s) = S_\varepsilon'[\mu + st\gamma_t^1, \mu + st\gamma_t^2](t\gamma_t^1, t\gamma_t^2) \quad \text{and} \quad g_t''(s) = S_\varepsilon''[\mu + st\gamma_t^1, \mu + st\gamma_t^2](t\gamma_t^1, t\gamma_t^2).$$

Since $(\mu, \nu) \mapsto S_\varepsilon''[\mu, \nu]$ is continuous in operator norm, this implies that $g_t''$ is a continuous function. Therefore, the path $(g_s : s \in [0, 1])$ is $C^2$ and

$$g_t(1) - g_t(0) - g_t'(0) - \frac{1}{2}g_t''(0) = \frac{1}{2}\int_0^1 (1-s)(g''(s) - g''(0))\, ds.$$

Plugging in the value of $g, g'$, and $g''$,

$$S_\varepsilon(\mu_t, \nu_t) - S_\varepsilon(\mu, \mu) - tS_\varepsilon'[\mu, \mu](\gamma_t^1, \gamma_t^2) - \frac{t^2}{2}S_\varepsilon''[\mu, \mu](\gamma_t^1, \gamma_t^2) = \frac{1}{2}\int_0^1 (1-s)(g''(s) - g''(0))\, ds.$$

Anchoring $g''(s) - g''(0)$ at the null distribution near $(\mu + st\gamma_t^1, \mu + st\gamma_t^2)$

$$
\begin{aligned}
|g''(s) - g''(0)| &= \left| S_\varepsilon''[\mu + st\gamma_t^1, \mu + st\gamma_t^2](t\gamma_t^1, t\gamma_t^2) - S_\varepsilon''[\mu, \mu](t\gamma_t^1, t\gamma_t^2) \right| \\
&\leq \left| S_\varepsilon''[\mu + st\gamma_t^1, \mu + st\gamma_t^2](t\gamma_t^1, t\gamma_t^2) - S_\varepsilon''[\mu + st\gamma_t^1, \mu + st\gamma_t^1](t\gamma_t^1, t\gamma_t^2) \right| \\
&\quad + \left| S_\varepsilon''[\mu + st\gamma_t^1, \mu + st\gamma_t^1](t\gamma_t^1, t\gamma_t^2) - S_\varepsilon''[\mu, \mu](t\gamma_t^1, t\gamma_t^2) \right| \\
&= \left| S_\varepsilon''[\mu + st\gamma_t^1, \mu + st\gamma_t^2](t\gamma_t^1, t\gamma_t^2) - S_\varepsilon''[\mu + st\gamma_t^1, \mu + st\gamma_t^1](t\gamma_t^1, t\gamma_t^2) \right| \\
&\quad + t^2 \left| \left\langle \gamma_t^1 - \gamma_t^2, (K_{\mu + st\gamma_t^1} - K_\mu)\gamma_t^1 - \gamma_t^2 \right\rangle \right|.
\end{aligned}
$$

From assumption, the first additive term on the right hand side is $O\left( st^3 \|\gamma_t^1 - \gamma_t^2\|_{\ell^\infty(\mathcal{H}_1)}^3 \right)$. Applying the local Lipschitz continuity of $\mu \mapsto K_\mu$ from Lem. E.4, we have that

$$
\begin{aligned}
t^2 \left| \left\langle \gamma_t^1 - \gamma_t^2, (K_{\mu + st\gamma_t} - K_\mu)\gamma_t^1 - \gamma_t^2 \right\rangle \right| &\leq t^2 \|\gamma_t^1 - \gamma_t^2\|_{\ell^\infty(\mathcal{H}_1)}^2 \|K_{\mu + st\gamma_t} - K_\mu\|_{\ell^\infty(\mathcal{H}_1) \to \mathcal{H}} \\
&\leq st^3 C_2 \|\gamma_t^1 - \gamma_t^2\|_{\ell^\infty(\mathcal{H}_1)}^3.
\end{aligned}
$$

Therefore,

$$
\left| \frac{1}{2} \int_0^1 (1-s)(g''(s) - g''(0))\, ds \right| \leq O\left( t^3 \|\gamma_t^1 - \gamma_t^2\|_{\ell^\infty(\mathcal{H}_1)}^3 \right).
$$

This completes the proof.

$\square$

**Lemma E.8** (Rate for second-order remainder term). *Let $P^* \in \mathcal{H}_0$ and its initial estimator $\widehat{P}$ satisfy $\|n^r(\widehat{P}_a - P_a^*)\|_{\ell^\infty(\mathcal{H}_1)} = \mathcal{O}_p(1)$, $a \in \{1, 0\}$, for some $r > 1/3$. Suppose $S_\varepsilon : (\mu, \nu) \mapsto S_\varepsilon(\mu, \nu)$ is second-order Hadamard differentiable along the path $s \mapsto (\widehat{P}_{1s}, \widehat{P}_{0s})$, where $\widehat{P}_{a\,s} = (1-s)P_a^* + s\widehat{P}_a$, $a \in \{1, 0\}$ and $s \in [0, 1]$. Further, assume that the estimated nuisance parameters satisfy the following conditions, uniformly across $a, a' \in \{1, 0\}$,*

1. $P_X^* \left[ x \mapsto \left( \frac{e_{P^*}(a \mid x)}{e_{\widehat{P}}(a \mid x)} - 1 \right) \left( P_{Y \mid a,x}^* - \widehat{P}_{Y \mid a,x} \right) \left( v_a^{(\widehat{P}_1, \widehat{P}_0)} - v_a^{(P_1^*, P_0^*)} \right) \right] = o_p(n^{-1})$;

2. $P_X^* \left[ x \mapsto \left( \frac{e_{P^*}(a \mid x)}{e_{\widehat{P}}(a \mid x)} - 1 \right) \left( P_{Y \mid a,x}^* - \widehat{P}_{Y \mid a,x} \right) K_{\widehat{P}_1}[\widehat{P}_1 - \widehat{P}_0] \right] = o_p(n^{-1})$;

3. $(P_X^* \otimes P_X^*) \left[ (x, x') \mapsto \left( \frac{e_{P^*}(a \mid x)}{e_{\widehat{P}}(a \mid x)} - 1 \right) \left( \frac{e_{P^*}(a' \mid x')}{e_{\widehat{P}}(a' \mid x')} - 1 \right) \left[ \left( P_{Y \mid a,x}^* - \widehat{P}_{Y \mid a,x} \right) \otimes \left( P_{Y \mid a',x'}^* - \widehat{P}_{Y \mid a',x'} \right) \right] k_{\widehat{P}_1} \right] = o_p(n^{-1})$;

4. $\left| S_\varepsilon''[\widehat{P}_{1s}, \widehat{P}_{0s}](\gamma^1, \gamma^2) - S_\varepsilon''[\widehat{P}_{1s}, \widehat{P}_{1s}](\gamma^1, \gamma^2) \right| = o_p(n^{-1/3})s\|\gamma^1 - \gamma^2\|_{\ell^\infty(\mathcal{H}_1)}^2$ *for any $s \in [0, 1]$ and $(\gamma^1, \gamma^2) \in \mathcal{M}_{0,\mu} \times \mathcal{M}_{0,\mu}$,*

*Then, the remainder term satisfies*

$$
\mathcal{R}_n = S(\widehat{P}) + (P^* - \widehat{P})\dot{S}_{\widehat{P}} + \frac{1}{2}(P^* - \widehat{P})^2 \ddot{S}_{\widehat{P}} - S(P^*) = o_p(n^{-1}).
$$

Applying the generalized Hölder inequality, it can be seen that the first, second, and fourth conditions amount to $o_p(n^{-1/3})$ conditions on the nuisance functions therein, while the third amounts to an $o_p(n^{-1/4})$ condition.

*Proof.* Here we prove that the remainder from the second-order von Mises expansion is $o_p(n^{-1})$. Under the null, the remainder term can be written as

$$
\mathcal{R}_n = S(\widehat{P}) + P^* \dot{S}_{\widehat{P}} + \frac{1}{2}(P^*)^{\otimes 2} \ddot{S}_{\widehat{P}}
$$

Recall the expectation operator notations from (7), (33), and (34). Consider the first two additive terms $\mathcal{S}(\widehat{P}) + P^*\dot{\mathcal{S}}_{\widehat{P}}$ first:

$$= \widehat{P}_1\widehat{\upsilon}_1 + \widehat{P}_0\widehat{\upsilon}_0 + Q_{\widehat{P},P^*}\left((y,a,x) \mapsto \frac{\widehat{\upsilon}_1(y)}{\hat{e}(1\,|\,x)} + \frac{\widehat{\upsilon}_0(y)}{\hat{e}(0\,|\,x)}\right)$$

$$= \left(P^* - P^*_{A,X}\widehat{P}_{Y\,|\,A,X} + P^*_X\widehat{P}_{Y,A\,|\,X}\right)\left((y,a,x) \mapsto \frac{\widehat{\upsilon}_1(y)}{\hat{e}(1\,|\,x)} + \frac{\widehat{\upsilon}_0(y)}{\hat{e}(0\,|\,x)}\right)$$

$$= P^*_X\left(x \mapsto \frac{e^*(1\,|\,x)}{\hat{e}(1\,|\,x)}P^*_{Y\,|\,1,x}\,\widehat{\upsilon}_1 + \frac{e^*(0\,|\,x)}{\hat{e}(0\,|\,x)}P^*_{Y\,|\,0,x}\,\widehat{\upsilon}_0\right)$$

$$\quad - P^*_X\left(x \mapsto \frac{e^*(1\,|\,x)}{\hat{e}(1\,|\,x)}\widehat{P}_{Y\,|\,1,x}\,\widehat{\upsilon}_1 + \frac{e^*(0\,|\,x)}{\hat{e}(0\,|\,x)}\widehat{P}_{Y\,|\,0,x}\,\widehat{\upsilon}_0\right)$$

$$\quad + P^*_X\left(x \mapsto \widehat{P}_{Y\,|\,1,x}\,\widehat{\upsilon}_1 + \widehat{P}_{Y\,|\,0,x}\,\widehat{\upsilon}_0\right)$$

$$= P^*_X\left(x \mapsto \frac{e^*(1\,|\,x)}{\hat{e}(1\,|\,x)}(P^*_{Y\,|\,1,x} - \widehat{P}_{Y\,|\,1,x})\,\widehat{\upsilon}_1 + \frac{e^*(0\,|\,x)}{\hat{e}(0\,|\,x)}(P^*_{Y\,|\,0,x} - \widehat{P}_{Y\,|\,0,x})\,\widehat{\upsilon}_0\right)$$

$$\quad - P^*_X\left(x \mapsto (P^*_{Y\,|\,1,x} - \widehat{P}_{Y\,|\,1,x})\widehat{\upsilon}_1 + (P^*_{Y\,|\,0,x} - \widehat{P}_{Y\,|\,0,x})\,\widehat{\upsilon}_0\right) + P^*_1\,\widehat{\upsilon}_1 + P^*_0\,\widehat{\upsilon}_0$$

$$= P^*_1\,\widehat{\upsilon}_1 + P^*_0\,\widehat{\upsilon}_0$$

$$\quad + P^*_X\left(x \mapsto \left(\frac{e^*(1\,|\,x)}{\hat{e}(1\,|\,x)} - 1\right)(P^*_{Y\,|\,1,x} - \widehat{P}_{Y\,|\,1,x})\,\widehat{\upsilon}_1 + \left(\frac{e^*(0\,|\,x)}{\hat{e}(0\,|\,x)} - 1\right)(P^*_{Y\,|\,0,x} - \widehat{P}_{Y\,|\,0,x})\,\widehat{\upsilon}_0\right)$$

From the first assumption, the last additive term is $o_p(n^{-1})$. The assumption applies because, under the null, $\upsilon_1^* = 0$ and $\upsilon_0^* = 0$, $P^*_1$-a.e. It remains to show

$$P^*_1\widehat{\upsilon}_1 + P^*_0\widehat{\upsilon}_0 + \frac{1}{2}(P^*)^{\otimes 2}\ddot{\mathcal{S}}_{\widehat{P}} = o_p(n^{-1}).$$

Let us focus on computing the form of $(P^*)^2\ddot{\mathcal{S}}_{\widehat{P}}$, where we have that

$$(P^*)^{\otimes 2}\ddot{\mathcal{S}}_{\widehat{P}} = \left(Q_{\widehat{P},P^*}\right)^{\otimes 2}\left((\omega_{\widehat{P}} \otimes \omega_{\widehat{P}}) \odot k_{\widehat{P}_1}\right).$$

Instead of working with the tensor product of the operator, to simplify the computation, let us characterize the action of operator $\left(P^* - P^*_{A,X}\widehat{P}_{Y\,|\,A,X} + P^*_X\widehat{P}_{Y,A\,|\,X} - \widehat{P}\right)$ on $\omega_{\widehat{P}} \odot f$ for a generic function $f : \mathcal{Y} \to \mathbb{R}$. This is equal to

$$Q_{\widehat{P},P^*}(\omega_{\widehat{P}} \odot f) = P^*_X\left[x \mapsto \frac{e^*(1\,|\,x)}{\hat{e}(1\,|\,x)}\left(P^*_{Y\,|\,1,x} - \widehat{P}_{Y\,|\,1,x}\right)f - \frac{e^*(0\,|\,x)}{\hat{e}(0\,|\,x)}\left(P^*_{Y\,|\,0,x} - \widehat{P}_{Y\,|\,0,x}\right)f\right]$$

$$= \quad + \left(P^*_X - \widehat{P}_X\right)\left[x \mapsto \left(\widehat{P}_{Y\,|\,1,x} - \widehat{P}_{Y\,|\,0,x}\right)f\right]$$

$$= P^*_X\left[x \mapsto \left(\frac{e^*(1\,|\,x)}{\hat{e}(1\,|\,x)} - 1\right)\left(P^*_{Y\,|\,1,x} - \widehat{P}_{Y\,|\,1,x}\right)f\right]$$

$$\quad - P^*_X\left[x \mapsto \left(\frac{e^*(0\,|\,x)}{\hat{e}(0\,|\,x)} - 1\right)\left(P^*_{Y\,|\,0,x} - \widehat{P}_{Y\,|\,0,x}\right)f\right] - \left(\widehat{P}_1 - \widehat{P}_0\right)f$$

Now replacing $f$ by $k_{\widehat{P}_1}(\cdot, y')$ for any $y' \in \mathcal{Y}$, and applying the operator $Q_{\widehat{P},P^*}$ along the second axis, we note that all cross terms involving the first two additive terms of the expression above are $o_p(n^{-1})$ by the second and third assumptions. Therefore, the dominant term from above expression is

$$\frac{1}{2}(P^*)^{\otimes 2}\ddot{\mathcal{S}}_{\widehat{P}} = \frac{1}{2}(\widehat{P}_1 - \widehat{P}_0)^{\otimes 2}k_{\widehat{P}_1} + o_p(n^{-1}).$$

Combining everything we have shown above yields

$$
\begin{aligned}
\mathcal{R}_n &= P_1^* v_1^{(\widehat{P}_1, \widehat{P}_0)} + P_1^* v_0^{(\widehat{P}_1, \widehat{P}_0)} + \frac{1}{2}(\widehat{P}_1 - \widehat{P}_0)^2 k_{\widehat{P}_1} + o_p(n^{-1}) \\
&= \left[ P_1^* v_1^{(\widehat{P}_1, \widehat{P}_0)} + P_1^* v_0^{(\widehat{P}_1, \widehat{P}_0)} + \frac{1}{2}(\widehat{P}_1 - \widehat{P}_0)^2 k_{P_1^*} \right] + \left[ \frac{1}{2}(\widehat{P}_1 - \widehat{P}_0)^2 (k_{\widehat{P}_1} - k_{P_1^*}) \right] + o_p(n^{-1}) \\
&= \left[ P_1^* v_1^{(\widehat{P}_1, \widehat{P}_0)} + P_1^* v_0^{(\widehat{P}_1, \widehat{P}_0)} + \frac{1}{2}(\widehat{P}_1 - \widehat{P}_0) K_{P_1^*}(\widehat{P}_1 - \widehat{P}_0) \right] + \left[ \frac{1}{2}(\widehat{P}_1 - \widehat{P}_0)(K_{\widehat{P}_1} - K_{P_1^*})(\widehat{P}_1 - \widehat{P}_0) \right] + o_p(n^{-1}).
\end{aligned}
$$

From Lem. E.4, the second additive term in the above expression is $\mathcal{O}_p(n^{-3r})$, given $\|n^r(\widehat{P}_a - P_a^*)\|_{\ell^\infty(\mathcal{H}_1)} = \mathcal{O}_p(1)$ for $a \in \{1, 0\}$. Now we focus on showing that the first additive term $P_1^* v_1^{(\widehat{P}_1, \widehat{P}_0)} + P_1^* v_0^{(\widehat{P}_1, \widehat{P}_0)} + \frac{1}{2}(\widehat{P}_1 - \widehat{P}_0)^2 k_{P_1^*} = o_p(n^{-1})$. Consider the following breakdown

$$
\begin{aligned}
(P_1^* &- \widehat{P}_1) v_1^{(\widehat{P}_1, \widehat{P}_0)} + (P_1^* - \widehat{P}_0) v_0^{(\widehat{P}_1, \widehat{P}_0)} + S_\varepsilon(\widehat{P}_1, \widehat{P}_0) + \frac{1}{2}(\widehat{P}_1 - \widehat{P}_0)^2 k_{P_1^*} \\
&= -(\widehat{P}_1 - P_1^*) \left( v_1^{(\widehat{P}_1, \widehat{P}_0)} - K_{P_1^*}(\widehat{P}_1 - \widehat{P}_0) - \varepsilon \rho \mathbf{1} \right) \\
&\quad - (\widehat{P}_0 - P_1^*) \left( v_0^{(\widehat{P}_1, \widehat{P}_0)} + K_{P_1^*}(\widehat{P}_1 - \widehat{P}_0) + \varepsilon \rho \mathbf{1} \right) \\
&\quad + S_\varepsilon(\widehat{P}_1, \widehat{P}_0) - \frac{1}{2}(\widehat{P}_1 - \widehat{P}_0) K_{P_1^*}(\widehat{P}_1 - \widehat{P}_0).
\end{aligned}
$$

From Cor. C.7, the quantities

$$
v_1^{(\widehat{P}_1, \widehat{P}_0)} - K_{P_1^*}(\widehat{P}_1 - \widehat{P}_0) - \varepsilon \rho \mathbf{1} \quad \text{and} \quad v_0^{(\widehat{P}_1, \widehat{P}_0)} + K_{P_1^*}(\widehat{P}_1 - \widehat{P}_0) + \varepsilon \rho \mathbf{1}
$$

are remainder from the first-order Hadamard expansion of the maps $(\mu, \nu) \mapsto v_1^{(\mu, \nu)}$ and $(\mu, \nu) \mapsto v_0^{(\mu, \nu)}$. Therefore, the first two additive terms are $\mathcal{O}_p\left( \|\widehat{P}_a - P_a^*\|_{\ell^\infty(\mathcal{H}_1)}^3 \right) = \mathcal{O}_p(n^{-3r}) = o_p(n^{-1})$ for $r > 1/3$. Finally, the fourth condition allows application of Lem. E.7 to obtain the rate

$$
\begin{aligned}
S_\varepsilon(\widehat{P}_1, \widehat{P}_0) - \frac{1}{2}(\widehat{P}_1 - \widehat{P}_0) K_{P_1^*}(\widehat{P}_1 - \widehat{P}_0) &= S_\varepsilon(\widehat{P}_1, \widehat{P}_0) - \frac{1}{2} S_\varepsilon''[P_1^*, P_1^*](\widehat{P}_1 - P_1^*, \widehat{P}_0 - P_0^*) \\
&= \mathcal{O}_p(\|\widehat{P}_1 - \widehat{P}_0\|_{\ell^\infty(\mathcal{H}_1)}^3) = \mathcal{O}_p(n^{-3r}).
\end{aligned}
$$

$\square$

**Lemma E.9** (Consistency under fixed alternative). *Let $P^* \notin \mathcal{H}_0$ and let $(\lambda_j, j > 0)$ are eigenvalues of the integral operator $f \mapsto \int \ddot{S}_{P^*}(\cdot, z) f(z) dP^*(y)$ repeated according to their multiplicity. Suppose conditions of Appx. E.1 hold, $\ddot{S}_{P^*} \in L^2(P^* \otimes P^*)$, and further $\|C_{P^*} \ddot{S}_{\widehat{P}} - C_{P^*} \ddot{S}_{P^*}\|_{L^2(P^* \otimes P^*)} \xrightarrow{p} 0$. Then,*

$$
P^*(n\overline{S} > q_{1-\alpha}) \to 1, \quad \text{as } n \to \infty,
$$

*where $q_{1-\alpha}$ is the $(1 - \alpha)$th quantile of $\sum_{j=1}^n \lambda_j(N_j^2 - 1)$.*

*Proof.* For $\overline{S} = \widehat{S} + \frac{1}{2} \mathbb{U}_n \ddot{S}_{\widehat{P}}$, consider the breakdown

$$
\sqrt{n}\,\overline{S} = \sqrt{n}\,S^* + \sqrt{n}(\widehat{S} - S^*) + \frac{\sqrt{n}}{2} \mathbb{U}_n(C_{P^*} \ddot{S}_{\widehat{P}} - C_{P^*} \ddot{S}_{P^*}) + \frac{\sqrt{n}}{2} \mathbb{U}_n(\ddot{S}_{\widehat{P}} - C_{P^*} \ddot{S}_{\widehat{P}}) + \frac{\sqrt{n}}{2} \mathbb{U}_n \ddot{S}_{P^*}.
$$

Under the alternative ($P^* \notin \mathcal{H}_0$), we have that $S^* > 0$ and therefore, $\sqrt{n}\, S^*$ diverges to infinity. Under the sufficient conditions presented in Lem. E.2 and Lem. E.3, the second additive term converges weakly to a mean-zero normal distribution (11). Since $\|C_{P^*} \ddot{S}_{\widehat{P}} - C_{P^*} \ddot{S}_{P^*}\|_{L^2(P^* \otimes P^*)} = o_p(1)$ by assumption, the third term is $o_p(n^{-1/2})$ by Chebyshev's inequality

for U-statistics (see Lem. E.6). The degeneracy of $\ddot{\mathcal{S}}_{\widehat{P}} - C_{P^*}\ddot{\mathcal{S}}_{\widehat{P}}$ and $\ddot{\mathcal{S}}_{P^*}$, together with Chebyshev's inequality for U-statistics, implies that the fourth and fifth terms are $o_p(1)$. Putting the four terms together, we have that $n^{1/2}\,\overline{\mathcal{S}}$ diverges to infinity in probability as $n \to \infty$, and so the test statistic $n\overline{\mathcal{S}}$ does as well.

Because $\ddot{\mathcal{S}}_{P^*} \in L^2(P^* \otimes P^*)$, the Hilbert-Schmidt norm $\sum_{j=1}^{\infty} \lambda_j^2 < \infty$. Consequently, all quantiles of $\sum_{j=1}^{\infty} \lambda_j(N_j^2 - 1)$ are finite. Hence, $P(n\overline{\mathcal{S}} > q_{1-\alpha}) \to 1$ as $n \to \infty$ for any $\alpha \in (0, 1)$. $\qquad\square$

## F. One-Step Estimator of MMD Distribution Treatment Effect

As a benchmark, we also compute first- and second-order one-step estimators for the maximum mean discrepancy between counterfactual outcome distributions; we call this the MMD treatment effect (MTE). The corresponding EIFs are derived using the same framework as for the Sinkhorn divergence; however, smoothness with respect to the underlying kernel mean embeddings is immediate for MMD due to its quadratic form.

The (squared) MMD functional is defined as

$$\mathcal{M} : P \in \mathcal{P} \mapsto \frac{1}{2}\mathrm{MMD}^2\left(\psi^1(P)\right), \psi^0(P)) = \frac{1}{2}\left\|\psi^1(P)) - \psi^0(P))\right\|_{\mathcal{G}}^2, \tag{38}$$

where recall that $\mathcal{G}$ is the Gaussian RKHS with kernel $g_\varepsilon(y_1, y_2) = \exp(-\|y_1 - y_2\|^2/2\varepsilon)$ and KME operator $\mathfrak{g}(\mu) = \int g_\varepsilon(\cdot, y)d\mu(y)$.

**Lemma F.1** (MMD first-order local parameter, Ex. 3 in Luedtke & Chung (2024))**.** *The parameter $\mathcal{M} : \mathcal{P} \to \mathbb{R}$ is pathwise differentiable at $P \in \mathcal{P}$ with first-order EIF given by*

$$\dot{\mathcal{M}}_P = (I - (I - (I - P_X)P_{A\,|\,X})P_{Y\,|\,A,X})f,$$

*where $f(x, a, y) = \frac{2a-1}{e_P(a\,|\,x)}\left(\mathfrak{g}(P_1)(y) - \mathfrak{g}(P_0)(y)\right)$.*

*Proof.* This lemma follows by applying the automatic differentiation algorithm in Thm. 1 of Luedtke (2026) with the composition of two primitives from that work: the kernel mean embedding in Lem. S11 in Appx. C.2.5 and the squared (RKHS) norm in Appx. C.4.3. $\qquad\square$

**Lemma F.2** (MMD second-order local parameter)**.** *The parameter $\mathcal{M} : \mathcal{P} \to \mathbb{R}$ is second- order pathwise differentiable at all $P \in \mathcal{H}_0$ with second- order EIF, denoted by $\ddot{\mathcal{M}}_P \in L^2(P^{\otimes 2})$, and equal to*

$$\ddot{\mathcal{M}}_P = (I - (I - (I - P_X)P_{A\,|\,X})P_{Y\,|\,A,X})^{\otimes 2}g,$$

*where $g = (\omega_P \otimes \omega_P) \odot g_\varepsilon$, and*

$$\omega_P(a, x) = \left(\frac{a}{e_P(1\,|\,x)} - \frac{1-a}{e_P(0\,|\,x)}\right).$$

*Proof.* Consider a submodel $(P_t : t \in [0, \delta)]) \in \mathfrak{P}(P, \mathcal{P}, s)$ starting at $P_0 = P$. Then, the second-order local parameter of $\mathcal{M}$ can be derived as

$$\frac{d^2}{dt^2}\mathcal{M}(P_t)\Big|_{t=0} = \left\|\frac{d}{dt}\left(\psi^1(P_t) - \psi^0(P_t)\right)\right\|_{\mathcal{H}}^2\Big|_{t=0} + \left\langle\psi^1(P_t) - \psi^0(P_t), \frac{d^2}{dt^2}(\psi^1(P_t) - \psi^0(P_t))\right\rangle_{\mathcal{H}}\Big|_{t=0}$$

$$= \left\|D\psi_P^1(s) - D\psi_P^0(s)\right\|_{\mathcal{H}}^2 + 0.$$

The last equality follows from the fact that, under the null, $\psi^1(P) = \psi^0(P)$. Using the form of $D\psi_P^1$ and $D\psi_P^0$ from

Lem. D.2, we have that $\frac{d^2}{dt^2}\mathcal{M}(P_t)\big|_{t=0}$ is equal to

$$\int\int (\mathcal{E}_z - (\mathcal{E}_{a,x} - (\mathcal{E}_x - P_X)P_{A\mid X})P_{Y\mid A,X})(\mathcal{E}'_z - (\mathcal{E}'_{a',x'} - (\mathcal{E}'_x - P_X)P_{A\mid X})P_{Y\mid A,X})$$
$$\times\, g(z,z')\, s(z)\, s(z')\, dP(z)\, dP(z'),$$

where $g(z,z') = \omega_P(a,x)\,\omega_P(a',x')\,g_\varepsilon(y,y')$. Since the above expression is bilinear in $s$, it is equal to $D^2\mathcal{M}_P(s)$. The proof concludes by noting that (i) $D^2\mathcal{M}_P(s) = \int\langle\ddot{\mathcal{M}}_P(z,\cdot),s\rangle_{L^2(P)}\,s(z)\,dP(z)$ and (ii) $\ddot{\mathcal{M}}_P$ is $P^{\otimes 2}$-square integrable since the kernel is bounded and strong positivity holds. $\qquad\square$

Lem. F.1 and Lem. F.2 yield the first and second order EIF used to construct the one-step estimator of $\mathcal{M}(P^*)$ and the test statistic for testing the null hypothesis. The one-step estimator of $\mathcal{M}(P^*)$ is

$$\widehat{\mathcal{M}} = \mathcal{M}(\widehat{P}) + P_n\dot{\mathcal{M}}_{\widehat{P}}.$$

The corresponding test statistic is

$$\overline{\mathcal{M}} = \mathcal{M}(\widehat{P}) + P_n\dot{\mathcal{M}}_{\widehat{P}} + \frac{\mathbb{U}_n}{2}\ddot{\mathcal{M}}_{\widehat{P}},$$

where $\mathbb{U}_n$ denotes the U-statistic operator constructed using the same sample split as $P_n$.

**Finite sample implementation.** From a computational perspective, the first-order EIF of MMD is obtained by applying Alg. 1 with $\mathbf{U}^1 = \mathbf{G}(\mathbf{P}^1 - \mathbf{P}^0)$ and $\mathbf{U}^2 = -\mathbf{G}(\mathbf{P}^1 - \mathbf{P}^0)$. The second-order EIF is computed via Alg. 2 by omitting the intermediate operator application (Step 3) and replacing $\mathbf{K}$ with $\mathbf{G}$ in Step 4.

## G. Algorithms for Finite-Sample Implementation

Here we present the algorithms for debiasing the STE estimator using first-order EIF computations in Alg. 1 and second-order EIF computations in Alg. 2. First, refer to the finite-sample setup in Sec. 5 and precomputations below.

**Precomputations.** The nuisance quantities—propensity score, outcome regression, centered entropic potentials (consequently self-transport entropic potential)—are estimated as outlined in Sec. 5. We denote the resulting estimators by $\hat{e}, \widehat{P}_{Y\mid A,X}$, and $(\hat{v}_1,\hat{v}_0)$ (consequently $\hat{\varphi}_1$). Define the data projection operator $\mathbf{D}_{A,X}$ acting on tensors $\mathbf{A}\in\mathbb{R}^{2\times n,p}$ by $(\mathbf{D}_{AX}\mathbf{A})_{[i,:]} = \mathbf{A}_{[2-a_i,i,:]}$. Let $\mathbf{G}$ be the $n\times n$ Gram matrix of $\mathcal{D}_n^1$. For $i\in[n]$ and $a\in\{1,0\}$, define the nuisance evaluations

$$\mathbf{E}_i = \hat{e}(x_i),\ \ \mathbf{U}_i^a = \hat{v}_a(y_i),\ \ \mathbf{F}_i = \hat{\varphi}_1(y_i).$$

Let $\mathbf{P}\in\mathbb{R}^{n\times 2\times n}$ collect evaluations of $\widehat{P}_{Y\mid A,X}$, so $\mathbf{P}_{i,j,k} = \widehat{P}_{Y\mid A,X}(y_i\mid 2-j,x_k)$, and define the marginal $\mathbf{P}^1$ (resp. $\mathbf{P}^0$) by averaging the treatment (resp. control) slice of $\mathbf{P}$ along along the third axis; e.g., $\mathbf{P}_i^1 = \frac{1}{n}\sum_{k=1}^n\mathbf{P}_{i,1,k}$. In both Algs. 1 and 2, multiplication by matrix $\mathbf{P}$ is done along the $Y$ (first) axis of both matrices.

## H. Max-Aggregated Test

Recall that $\mathcal{T}_{n,\varepsilon} = n\overline{\mathcal{S}}$ is the test statistic for a fixed $\varepsilon > 0$ with explicit dependence on $n$ and $\varepsilon$. Under the null, from Thm. 4.1, we have that $\mathcal{T}_{n,\varepsilon} = n\mathbb{U}_n h_\varepsilon/2 + o_p(1)$, where $h_\varepsilon$ is the symmetric, one-degenerate kernel $h_\varepsilon = \ddot{\mathcal{S}}_{P^*}\in L^2(P^*\otimes P^*)$. For each $\varepsilon$, define the Hilbert-Schmidt operator $(T_\varepsilon f)(z) = \int h_\varepsilon(z,z')f(z')\,dP^*(z')$. Since $h_\varepsilon$ is symmetric, $T_\varepsilon$ is compact and self-adjoint on $L^2(P^*)$. The spectral theorem yields that there exist real eigenvalues $(\lambda_{j,\varepsilon})_{j\geq 1}$ and an orthonormal family $(\phi_{j,\varepsilon})_{j\geq 1}$ such that $h_\varepsilon(z,z') = \sum_{j=1}^\infty \lambda_{j,\varepsilon}\phi_{j,\varepsilon}(z)\phi_{j,\varepsilon}(z')$ in $L^2(P^*\otimes P^*)$ and $\sum_{j=1}^\infty \lambda_{j,\varepsilon}^2 < \infty$. Because $h_\varepsilon$ is one-degenerate, every eigenfunction corresponding to a nonzero eigenvalue belongs to $L_0^2(P^*)$.

---

**Algorithm 1** First-order EIF evaluations

---

1: **Input:** $\mathbf{U}^1, \mathbf{U}^0, \mathbf{P}, \mathcal{D}_n^1$
2: **Output:** $\mathbf{I}^1 \in \mathbb{R}^n$
3: $\mathbf{W}^1 \leftarrow a_i/\mathbf{E}_i$ and $\mathbf{W}^0 \leftarrow (1 - a_i)/(1 - \mathbf{E}_i)$
4: $T_1 \leftarrow \mathbf{W}^1 \odot \mathbf{U}^1 + \mathbf{W}^0 \odot \mathbf{U}^0$
5: $T_2 \leftarrow \mathbf{W}^1 \odot \mathbf{D}_{A,X}(\mathbf{P}\mathbf{U}^1) + \mathbf{W}^0 \odot \mathbf{D}_{A,X}(\mathbf{P}\mathbf{U}^0)$
6: $T_3 \leftarrow \mathbf{D}_{1,X}(\mathbf{P}\mathbf{U}^1) + \mathbf{D}_{0,X}(\mathbf{P}\mathbf{U}^0)$
7: $T_4 \leftarrow (1_n^\top T_3/n)1_n$
8: $\mathbf{I}^1 \leftarrow T_1 - T_2 + T_3 - T_4$

---

**Algorithm 2** Second-order EIF evaluations

---

1: **Input:** $\mathbf{P}, \mathbf{P}^1, \mathbf{E}, \mathbf{F}, \mathbf{G}$
2: **Output:** $\mathbf{I}^2 \in \mathbb{R}^{n \times n}$
3: $\mathbf{X}_{ij} \leftarrow \exp((\mathbf{F}_i + \mathbf{F}_j - \mathbf{G}_{ij})/\varepsilon)$
4: $\mathbf{\Omega}_{ij} \leftarrow 1/[\mathbb{1}(i = 1)\mathbf{E}_j - \mathbb{1}(i = 2)(1 - \mathbf{E}_j)]$
5: $\mathbf{M} \leftarrow (I - (\mathbf{P}^1 \odot \mathbf{X})^2)^{-1}(\mathbf{X} - \mathbf{1}_{n \times n}/n)$
6: **for** $i = 1$ **to** 2 **do**
7: $\quad T_1 \leftarrow (\mathbf{D}_{A,X}\mathbf{\Omega} \otimes \mathbf{D}_{A,X}\mathbf{\Omega}) \odot \mathbf{M}$
8: $\quad T_2 \leftarrow (\mathbf{D}_{A,X}\mathbf{\Omega} \otimes \mathbf{D}_{A,X}\mathbf{\Omega}) \odot \mathbf{D}_{A,X}(\mathbf{P}\mathbf{M})$
9: $\quad T_3 \leftarrow ((\mathbf{P}\mathbf{M})_{[1,:,:]} - (\mathbf{P}\mathbf{M})_{[2,:,:]})$
10: $\quad T_4 \leftarrow \mathbf{1}_n^\top T_3/n$
11: $\quad \mathbf{M} \leftarrow (T_1 - T_2 + T_3 - T_4)^\top$
12: **end for**
13: Set $\mathbf{I}^2 = \mathbf{M}$

---

Let $G = \{G(f) : f \in L_0^2(P^*)\}$ be an isonormal Gaussian process on $L_0^2(P^*)$. The Gaussian coordinates $(G(\phi_{j,\varepsilon}))_{j \geq 1}$ are i.i.d. standard normal random variables. Let $I_2$ denote the second-order Wiener–Itô integral with respect to $G$. Then for a tensor product $f \otimes g$, with $f, g \in L_0^2(P^*)$, we have that $I_2(f \otimes g) = G(f)G(g) - \langle f, g \rangle_{L^2(P^*)}$. Therefore,

$$I_2(h_\varepsilon) = I_2\left(\sum_{j=1}^\infty \lambda_{j,\varepsilon}(\phi_{j,\varepsilon} \otimes \phi_{j,\varepsilon})\right) = \sum_{j=1}^\infty \lambda_{j,\varepsilon}I_2(\phi_{j,\varepsilon} \otimes \phi_{j,\varepsilon}) = \sum_{j=1}^\infty \lambda_{j,\varepsilon}\left(G(\phi_{j,\varepsilon})^2 - 1\right). \tag{39}$$

The above is the spectral representation of the so-called Gaussian chaos random variable (Janson, 1997, Chap. 2) associated with the symmetric first-order degenerate kernel $h_\varepsilon$.

We now provide results we use to establish the asymptotic distribution of the STEAgg test statistic. For each fixed $\varepsilon > 0$, Thm. 4.1 proves that

$$\mathcal{T}_{n,\varepsilon} = \frac{n}{2}U_n h_\varepsilon + o_p(1) \xrightarrow{d} \frac{1}{2}\sum_{j=1}^\infty \lambda_{j,\varepsilon}(N_j^2 - 1),$$

where $(N_j)_{j \geq 1}$ are i.i.d. standard normal random variables. The distributional limit follows from standard U-statistics limit theory (Leucht & Neumann, 2013, Thm. 1). Using the fact that $(G(\phi_{j,\varepsilon}))_{j=1}^\infty$ are independent normal random variables and (39), we have that $\mathcal{T}_{n,\varepsilon} \xrightarrow{d} W(\varepsilon) := \frac{I_2(h_\varepsilon)}{2}$.

**Lemma H.1** (Joint null distribution of test statistics). *Suppose the conditions of Thm. 4.1 hold for each $\varepsilon \in \Xi$, then*

$$(\mathcal{T}_{n,\varepsilon_1}, \ldots, \mathcal{T}_{n,\varepsilon_m}) \xrightarrow{d} (W(\varepsilon_1), \ldots, W(\varepsilon_m)), \quad in \; \mathbb{R}^m. \tag{40}$$

*The limit vector is a jointly defined second-order Gaussian-chaos vector. Explicitly, for every $a = (a_1, \ldots, a_m) \in \mathbb{R}^m$,*

$$a^\top \left( W(\varepsilon_1), \ldots, W(\varepsilon_m) \right) = \frac{1}{2} I_2 \left( \sum_{r=1}^m a_r h_{\varepsilon_r} \right).$$

*Particularly, $\mathrm{Cov}(W(\varepsilon_i), W(\varepsilon_j)) = \frac{1}{2} \langle h_{\varepsilon_i}, h_{\varepsilon_j} \rangle_{L^2(P^* \otimes P^*)}$. Each marginal $W(\varepsilon_r)$ admits the expansion*

$$W(\varepsilon_r) = \sum_{j=1}^\infty \frac{\lambda_{j,\varepsilon_r}}{2} \left( G(\phi_{j,\varepsilon_r})^2 - 1 \right),$$

*with dependence across $r$ induced by the common isonormal process $G$.*

*Proof.* Since $\Xi$ is finite and for each $\varepsilon \in \Xi$, $\mathcal{T}_{n,\varepsilon} = \frac{n}{2} \mathbb{U}_n h_\varepsilon + r_{n,\varepsilon}$ such that $r_{n,\varepsilon} = o_p(1)$, we have that $\sup_{\varepsilon \in \Xi} |r_{n,\varepsilon}| = o_p(1)$. Thus it suffices to prove joint convergence of the leading $U$-statistic vector $(\mathbb{U}_n h_{\varepsilon_1}, \ldots, \mathbb{U}_n h_{\varepsilon_m})$. By the Cramér-Wold device (van der Vaart, 2000, Thm. 2.1), it is enough to show that for every fixed $a = (a_1, \ldots, a_m) \in \mathbb{R}^m$,

$$\sum_{r=1}^m a_r \mathcal{T}_{n,\varepsilon_r} = \frac{n}{2} \mathbb{U}_n \left( \sum_{r=1}^m a_r h_{\varepsilon_r} \right) \xrightarrow{d} \sum_{r=1}^m a_r W(\varepsilon_r).$$

Set $h_a := \sum_{r=1}^m a_r h_{\varepsilon_r}$. Because the grid is finite and each $h_{\varepsilon_r}$ is measurable, symmetric, and square-integrable, the same properties hold for $h_a$. Likewise, one-degeneracy is preserved under finite linear combinations. Hence, using the same asymptotic result for U-statistic, associated with kernel $h_a$, we have that

$$\frac{n}{2} \mathbb{U}_n h_a \xrightarrow{d} \frac{1}{2} I_2(h_a) = \frac{1}{2} I_2 \left( \sum_{r=1}^m a_r h_{\varepsilon_r} \right).$$

This proves the desired joint convergence. The covariance identity follows because of the isometry property of multiple Wiener–Itô integrals (Nualart & Etheridge, 2019, Def. 3.1)

$$\mathrm{Cov}(W(\varepsilon_i), W(\varepsilon_j)) = \frac{1}{4} \mathrm{Cov}(I_2(h_{\varepsilon_i}), I_2(h_{\varepsilon_j})) = \frac{1}{4} \mathbb{E} \left[ I_2(h_{\varepsilon_i}) I_2(h_{\varepsilon_j}) \right] = \frac{1}{2} \langle h_{\varepsilon_i}, h_{\varepsilon_j} \rangle_{L^2(P^* \otimes P^*)}.$$

$\square$

For a fixed $0 < \beta < 1$ and for each $r$, let $F_r$ denote the distribution function of $W(\varepsilon_r)$, and let $q_{\varepsilon_r} := F_r^{-1}(1 - \beta)$. If $h_{\varepsilon_r} \not\equiv 0$, then $W(\varepsilon_r)$ is a non-degenerate second-order Gaussian-chaos random variable and $F_r$ is continuous.

*Proof Thm. 4.2.* Suppose the null holds. Under the assumption of the theorem, we have that $q_{n,\varepsilon_r} \xrightarrow{P} q_{\varepsilon_r}$ for all $r \in \{1, \ldots, m\}$ and $(q_{n,\varepsilon_1}, \ldots, q_{n,\varepsilon_m}) \to (q_{\varepsilon_1}, \ldots, q_{\varepsilon_m})$ in $(0, \infty)^m$. By Slutsky's theorem,

$$\left( \frac{\mathcal{T}_{n,\varepsilon_1}}{q_{n,\varepsilon_1}}, \ldots, \frac{\mathcal{T}_{n,\varepsilon_m}}{q_{n,\varepsilon_m}} \right) \xrightarrow{d} \left( \frac{W(\varepsilon_1)}{q_{\varepsilon_1}}, \ldots, \frac{W(\varepsilon_m)}{q_{\varepsilon_m}} \right), \quad \text{in } \mathbb{R}^m.$$

Applying the continuous mapping theorem to $\max(\cdot) : \mathbb{R}^m \to \mathbb{R}$ gives the first result.

Now consider the fixed alternative $P^* \notin \mathcal{H}_0$. By the assumptions of Lem. E.2 and Lem. E.3, together with the displayed $L^2$ convergence assumption, the same argument as in Lem. E.9 implies that for each $\varepsilon \in \Xi$, $\mathcal{T}_{n,\varepsilon} \xrightarrow{P} \infty$. On the other hand, since $\ddot{\mathbb{S}}_{\varepsilon,P^*} \in L^2(P^* \otimes P^*)$ for each $\varepsilon \in \Xi$, the associated Hilbert–Schmidt operator has square-summable eigenvalues $\sum_{j=1}^\infty \lambda_{\varepsilon,j}^2 < \infty$. Therefore $W(\varepsilon) = \sum_{j=1}^\infty \lambda_{\varepsilon,j}(N_j^2 - 1)$ has finite quantiles. Since $q_{n,\varepsilon} \xrightarrow{p} q_\varepsilon$ with $q_\varepsilon > 0$, we have that

$q_{n,\varepsilon} = O_p(1)$ and $\frac{1}{q_{n,\varepsilon}} = O_p(1)$ for each $\varepsilon \in \Xi$. It follows that $\frac{\mathcal{T}_{n,\varepsilon}}{q_{n,\varepsilon}} \xrightarrow{p} \infty$ for all $\varepsilon \in \Xi$. Hence,

$$\mathcal{T}_n^{\mathrm{agg}} = \max_{\varepsilon \in \Xi} \frac{\mathcal{T}_{n,\varepsilon}}{q_{n,\varepsilon}} \geq \frac{\mathcal{T}_{n,\varepsilon_1}}{q_{n,\varepsilon_1}} \xrightarrow{P} \infty.$$

Finally, because $\Xi$ is finite and each $W(\varepsilon)/q_\varepsilon$ is tight, the random variable $W^{\mathrm{agg}}$ has finite $(1-\alpha)$-quantile. Therefore,

$$\Pr(\mathcal{T}_n^{\mathrm{agg}} > q_{1-\alpha}(W^{\mathrm{agg}})) \to 1,$$

so the proposed test rejects with probability tending to 1. $\qquad\square$

## I. Experimental Details

For our experiments, we use a sample splitting approach in which an effective sample of size $2n$ is divided evenly. The first $n$ observations are used to construct an initial estimator $\widehat{P}$ of $P^*$, and the remaining $n$ observations are used to form the empirical distribution $P_n$.

### I.1. Datasets

#### I.1.1. SIMULATIONS.

The covariates $X \sim \mathcal{N}(\mathbf{0}_3, I_3)$, and the treatment assignment model is logistic such that,

$$A \,|\, X = x \sim \mathrm{Bernoulli}(\mathrm{expit}(\mathbf{1}_3^\top x))).$$

Throughout Exp (i) and (ii), we maintain a fixed covariance matrix $\Sigma$. For Exp (i) (mean difference), the outcome $Y$ given $(A, X)$ is sampled as

$$Y \,|\, (A = a, X = x) \sim \begin{cases} \mathcal{N}(\mathbf{0}_2, \Sigma) & \text{if } a = 0, \\ \mathcal{N}(\theta \mathbf{1}_2, \Sigma) & \text{if } a = 1. \end{cases},$$

For Exp (ii) (covariance difference), the outcome $Y$ given $(A, X)$ is sampled as

$$Y \,|\, (A = a, X = x) \sim \begin{cases} \mathcal{N}(\mathbf{0}_2, \Sigma) & \text{if } a = 0, \\ \mathcal{N}(\mathbf{0}_2, \Sigma + \theta(uv^\top + vu^\top)) & \text{if } a = 1, \end{cases}$$

where $u$ and $v$ are the eigenvectors of $\Sigma$. Figure 3 illustrates the resulting $95\%$ ellipsoids for varying $\theta$.

**Simulations for aggregated test.** We replicate the simulation setup of Exp (ii), but now evaluate the tests over a grid of $\varepsilon$ values of the form $\varepsilon = \eta m$, where $m$ is the median heuristic and $\eta \in \{0.25, 0.5, 1, 2, 4\}$. For each fixed $\varepsilon$, we compute the corresponding MTE- and STE-based test statistics and p-values, and we additionally evaluate the aggregated procedures MTEAgg and STEAgg. The results are shown in Fig. 4 under both the null (Exp (ii) with $\theta = 0.0$) and an alternative (Exp (ii) with $\theta = 0.8$). Under the null, both MTE- and STE-based tests exhibit lower type-I error as $\varepsilon$ increases. Under the alternative, for STE, power initially improves with $\varepsilon$, but eventually deteriorates when $\varepsilon$ becomes too large, reflecting oversmoothing and a loss of sensitivity to distributional differences; in this regime, Sinkhorn divergence begins to behave more like MMD (Feydy et al., 2019). Interestingly, for STE power is also lower when $\varepsilon$ is too small, which is consistent with the numerical instability of Sinkhorn iterations at low regularization. Overall, this experiment recovers the expected bias-variance tradeoff in the choice of $\varepsilon$ for STE. Among the values considered, the strongest performance is attained near the median heuristic, which we therefore recommend as a practical default.

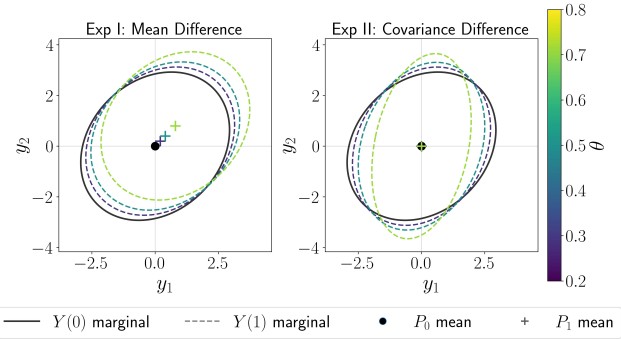

*Figure 3.* Mean and covariance ellipsoids (95%) of counterfactual outcome distributions under varying gap between $P_0$ and $P_1$, parametrized by $\theta$. Exp (i): Mean difference experiment $P_0 = \mathbb{N}(\mathbf{0}_2, \Sigma)$ and $P_1 = \mathbb{N}(\theta\mathbf{1}_2, \Sigma)$. Exp (ii): Covariance difference experiment $P_0 = \mathbb{N}(\mathbf{0}_2, \Sigma)$ and $P_1 = \mathbb{N}(\mathbf{0}_2, \Sigma + \theta\Delta)$.

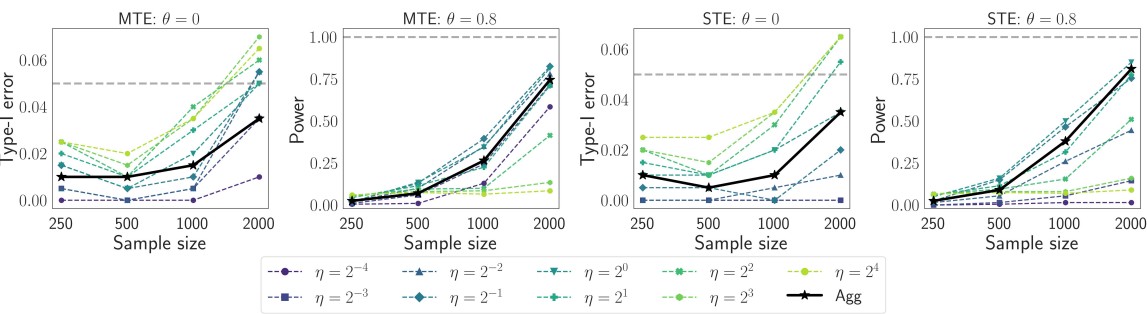

*Figure 4.* Type-I error and power in Exp (ii) for the aggregated procedures MTEAgg and STEAgg, together with the corresponding MTE- and STE-based tests evaluated on a finite grid of kernel bandwidth parameters $\varepsilon = \eta m$; $m =$ median heuristic

### I.1.2. PCAM DATASET

We provide here the exact data-generating mechanism used for the image-outcome experiments. For each unit, covariates are generated as $X \sim \mathcal{N}(0, I_5)$. Conditional on $X = x$, a latent disease status $S \in \{0, 1\}$ is sampled from a Bernoulli distribution with success probability $\text{expit}(\pi_0 + \pi_1^\top x)$, where $S = 1$ denotes metastatic tissue. Treatment assignment $A \in \{0, 1\}$ is generated independently given $X = x$ from a Bernoulli distribution with success probability $\text{expit}(\gamma_0 + \gamma_1^\top x)$, inducing confounding through shared dependence on $X$.

Outcome images $Y$ are sampled from the empirical PCam image distributions conditional on $(S, A, X)$. Specifically, if $S = 0$, outcomes are drawn from the empirical distribution supported on $\mathcal{D}_0$. If $S = 1$ and $A = 0$, outcomes are drawn from $\mathcal{D}_1$. If $S = 1$ and $A = 1$, outcomes are drawn from $\mathcal{D}_0$ with probability $q(x) = \text{expit}(\beta_0 + \beta_1^\top x)$ and from $\mathcal{D}_1$ otherwise. This construction induces heterogeneous, covariate-dependent treatment effects on the counterfactual distributions while preserving overlap and positivity.

### I.2. Compute details

The code was written in Python 3 and we use PyTorch for automatic differentiation. All our experiments were conducted on a CUDA-enabled machine with 12GB GPU memory, 64GB RAM, and 24 vCPUs. Although the experiments were run on a GPU, we observe similar complexity trends for CPU implementations. Fig. 6 plots the average wall-clock time

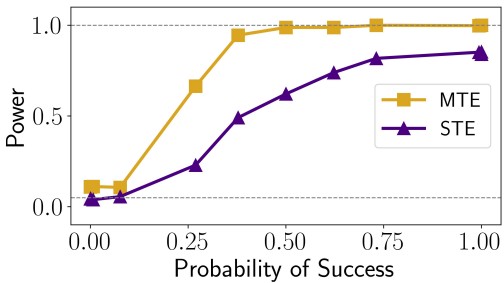

*Figure 5.* Type 1 error (far left point) and power (all other points) of STE and MTE as a function of the treatment success probability for the PCam dataset.

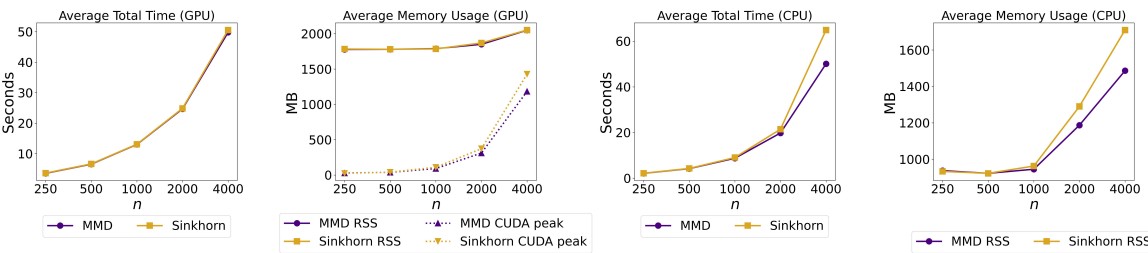

*Figure 6.* Wall-clock runtime (in seconds) and memory usage (in MB) for the simulation setup across increasing sample sizes $n$, averaged over 20 Monte Carlo simulations for both GPU and CPU implementations.

(averaged over 20 Monte Carlo simulations) and memory usage across increasing sample sizes $n$ for both GPU and CPU implementations.

### I.3. Acceleration recommendations

We now make the computational bottlenecks of the second-order one-step STE estimator explicit and summarize practical acceleration strategies. From Sec. 5, the total computational complexity is $O(n^2(n+t))$, where $n$ is the sample size and $t$ is the number of Sinkhorn iterations. This cost is driven primarily by the Sinkhorn algorithm, which contributes $O(n^2t)$, and Algorithm 2, which contributes $O(n^3)$. In addition, once the Gram matrix of the second-order EIF has been formed, the second-order bias-correction term requires a U-statistic computation, which incurs an additional quadratic computational load. We now discuss standard acceleration techniques that can be incorporated directly into our framework. While we expect these strategies to yield empirical gains similar to those reported in the existing literature, a full asymptotic analysis of the resulting accelerated procedures is beyond the scope of the present paper.

**Sinkhorn algorithm.**    To accelerate the Sinkhorn step, several complementary strategies are available. Greenkhorn (Altschuler et al., 2017) replaces full alternating row/column normalizations by greedy coordinate updates, which often improves practical performance while retaining the same dense-kernel representation. Batch Greenkhorn methods (Kostic et al., 2022) extend this idea by updating batches of coordinates and provide refined convergence guarantees. These approaches aim to reduce the amount of scaling work required for convergence, but they do not by themselves eliminate the $O(n^2)$ cost of storing and multiplying by a dense Gibbs kernel. To address this bottleneck, Nyström-Sinkhorn (Altschuler et al., 2019) approximates the Gibbs kernel $K_{ij} = e^{-C_{ij}/\varepsilon}$ using a low-rank Nyström factorization, reducing per-iteration

kernel-vector multiplication costs and memory usage from $O(n^2)$ to $O(nr)$, where $r$ is the approximation rank. These approaches can be incorporated into our one-step STE estimator to reduce the per-iteration computational cost.

**Algorithm 2 acceleration.**  A natural way to reduce the $O(n^3)$ cost of Alg. 2 is to replace the dense discretized self-transport kernel $\mathbf{X}$, defined in line 3 by

$$\mathbf{X}_{ij} = \exp\left\{\frac{\mathbf{F}_i + \mathbf{F}_j - \mathbf{G}_{ij}}{\varepsilon}\right\},$$

with a rank-$r$ Nyström approximation $\tilde{\mathbf{X}} = CW^\dagger C^\top = UV^\top$ with $r \ll n$. This is well aligned with prior work on scalable entropic OT, where low-rank Nyström approximations are used to compress the Gibbs kernel while preserving the Sinkhorn scaling structure (Altschuler et al., 2019). In our setting, the cubic bottleneck arises because line 5 requires solving a dense $n \times n$ linear system,

$$\mathbf{M} = \left(I - (\mathbf{P}^1 \odot \mathbf{X})^2\right)^{-1} \left(\mathbf{X} - \mathbf{1}\mathbf{1}^\top/n\right),$$

and because the subsequent contractions $\mathbf{PM}$ in lines 8-10 amount to repeated dense matrix multiplications, each of overall order $n^3$. Replacing $\mathbf{X}$ by $\tilde{\mathbf{X}}$ lowers both costs: first, if $A := \mathbf{P}^1 \odot \tilde{\mathbf{X}}$ is represented in low-rank form as $A = \tilde{U}V^\top$, then

$$A^2 = (\tilde{U}V^\top)(\tilde{U}V^\top) = \tilde{U}BV^\top, \qquad B := V^\top\tilde{U} \in \mathbb{R}^{r \times r},$$

so $\left(I - A^2\right)^{-1}$ can be applied via the Woodbury identity using only an $r \times r$ inner solve, reducing line 5 from $O(n^3)$ to $O(nr^2 + r^3)$ after the low-rank factors are formed. Second, every multiplication by an $n \times n$ matrix generated from $\mathbf{X}$ can be evaluated through the factorization $UV^\top$, costing $O(nr)$ per vector and $O(n^2r)$ per matrix block, so the contractions $\mathbf{PM}$ drop from $O(n^3)$ to $O(n^2r)$. The remaining operations in the algorithm: forming $\mathbf{X}$, applying $\mathbf{D}_{A,X}$, and elementwise products are at most quadratic, so under a Nyström representation the total complexity of the second-order routine is reduced from $O(n^3)$ to roughly $O(n^2r + nr^2 + r^3)$, with memory reduced from $O(n^2)$ to $O(nr)$. Therefore, a Nyström approximation to the discretized Sinkhorn self-transport kernel is a direct and computationally efficient way to scale the second-order correction while retaining the kernel structure induced by the entropic potentials.

**$U$-statistic computation.**  The $O(n^2)$ complexity of the U-statistic, standard in kernel two-sample testing, can be mitigated using linear-time approximations (Gretton et al., 2012) or incomplete U-statistics (Schrab et al., 2022). Both can be directly substituted into the second-order bias-correction term of the STE one-step estimator.

