# OpenReview forum: "Sinkhorn Treatment Effects: A Causal Optimal Transport Measure"
_ICML.cc/2026/Conference — ICML 2026 regular_

### Official Review · Reviewer_x3BU · 2026-03-10

**Soundness:** 3
**Presentation:** 3
**Significance:** 3
**Originality:** 3
**Overall Recommendation:** 5
**Confidence:** 3

**Summary:**

Classical causal inference focuses on the average treatment effect, which fails to capture distributional shifts (e.g., changes in variance or shape) that do not affect the mean. While distributional treatment effects exist, current measures like MMD saturate when distributions are far apart, and f-divergences require restrictive absolute continuity. The paper introduces the Sinkhorn treatment effect (STE), which uses entropic optimal transport to measure the divergence between counterfactual distributions. The authors characterize STE as a statistical functional of counterfactual mean embeddings, enabling the use of semiparametric theory.

**Compliance With Llm Reviewing Policy:**

Affirmed.

**Final Justification:**

I will keep my score.

**Key Questions For Authors:**

# Questions

**Q1**: How does the performance of the STE test change with the choice of the regularization parameter $\epsilon$? Does a very small $\epsilon$ (closer to pure OT) lead to numerical instability in the second-order EIF calculation? Could you provide guidance on selecting $\epsilon$ in practice (e.g., cross-validation, median heuristic)?

**Q2:** The image experiment uses simulated treatments on the PCam dataset. Could the method be applied to real observational data with known propensity scores? What additional challenges would arise?

**Q3:** The image experiment uses a synthetic treatment effect construction. Have you considered applying this to real observational datasets where ground truth is unknown, and if so, how would you validate the approach?

**Limitations:**

yes

**Strengths And Weaknesses:**

# Strengths

- **S1:** The connection between entropic OT and semiparametric influence functions for causal inference is a significant and non-trivial advancement.

- **S2:** The use of second-order EIFs to handle the degeneracy of the test under the null is a sophisticated solution to a known problem in distributional testing.

- **S3:** The proposed method demonstrates superior geometric sensitivity; unlike MMD, it continues to grow as distributions move further apart, providing more informative effect sizes.

- **S4:** The paper is well-structured, clearly motivating why a new DTE measure is needed and walking through the technical requirements for its construction.


# Weaknesses

- **W1:** The empirical evaluation, while compelling, is somewhat limited in scope. Only one synthetic data-generating process is considered, and the image experiment uses simulated treatments rather than real causal structure. Additional experiments with different data-generating mechanisms (e.g., non-Gaussian outcomes, nonlinear propensity models) would strengthen the empirical claims.
- **W2:** The theoretical framework relies on the compactness of the outcome space $\mathcal{Y}$. While common in OT literature, it might limit direct application to unbounded outcomes without further modification.
- **W3:** The computational complexity analysis notes O(n³) for the second-order estimator due to matrix operations. For large n (e.g., n > 10⁴), this may become prohibitive. The paper could discuss potential approximations (e.g., Nyström methods, minibatching) to improve scalability.
- **W4:** The paper focuses primarily on Gaussian kernels. While these are characteristic, a brief discussion on how the choice of kernel or the entropic regularization $\epsilon$ impacts the test's power would be beneficial.

---

> ### Author Rebuttal · Authors · 2026-03-31
>
> We thank the reviewer for these helpful comments. They led us to broaden both the empirical and practical discussion. Due to space constraints, details are in the [rebuttal document](https://limewire.com/d/CyIaX#ve9BKM9Dmr).
>
> > **[W1] Empirical scope / real observational data**
>
> First, in response to similar suggestion reflected by Reviewers vUbg and WXjt, we expanded the simulation study to include alternatives that decouple mean and covariance differences between $P_0$ and $P_1$. This setup shows that both MTE- and STE tests perform well for mean shifts, while the STE-based test has clearer power gains for structured covariance perturbations. Second, we added an example on real-world causal data from the LaLonde NSW earnings study ([Section 2](https://limewire.com/d/CyIaX#ve9BKM9Dmr)).
>
> > **[W2] Compact outcome-space assumption**
>
> We agree that compactness limits direct coverage of unbounded outcomes. In the current paper, it is a technical assumption used to invoke existing statistical theory for Sinkhorn divergence. We will clarify that this is mainly for theoretical tractability, and that extending the results to unbounded outcomes under suitable tail conditions is an important direction for future work.
>
> > **[W3] Computation and scalability**
>
> We make the computational limitations of the STE estimator explicit and discuss acceleration methods that can be integrated in our framework ([Section 5](https://limewire.com/d/CyIaX#ve9BKM9Dmr)). The cost is dominated by Sinkhorn iterations and dense matrix operations in Algo. 2 in the main paper, yielding $O(n^2(n+t))$ complexity for the second-order correction, where $t$ is the number of Sinkhorn iterations. We discuss standard acceleration methods: Greenkhorn Sinkhorn for faster Sinkhorn optimization, Nyström or kernel-compression methods and stochastic OT to reduce the per-iteration $O(n^2))$ bottleneck in Sinkhorn algorithm, linear-time or incomplete $U$-statistic approximations, and Nyström approximations for Algo. 2 that reduce the cubic cost to $O(n^2 r)$. See Section 4 in the rebuttal document.
>
> > **[W4] Kernel choice / bandwidth**
>
> For STE, the Gaussian RKHS is primarily a theoretical device used to place the counterfactual distributions in a normed linear space where pathwise differentiability can be established. Unlike the MTE test, STE does not require explicitly computing KME in that RKHS. Thus, the effect of kernel choice on power is more direct for MTE than for STE, while for STE the more practically relevant tuning parameter is the entropic regularization. To reduce sensitivity to any single bandwidth choice for MTE, we propose the aggregated test **MTEAgg** (similar to STEAgg), which combines evidence across a finite grid of bandwidth values, taken in practice as stable multiples of the median heuristic.
>
> > **[Q1] Sensitivity to the entropic regularization**
>
> The regularization parameter $\varepsilon$ induces a bias-variance tradeoff: large values oversmooths the coupling, while very small values can lead to numerical instability in Sinkhorn iterations. In our experiments, we use the median heuristic based on median pairwise cost. To reduce sensitivity to any single value, we propose **STEAgg**, which aggregates evidence over a finite grid of numerically stable $\varepsilon$ values while retaining asymptotic type-I control. Empirically, STEAgg is more robust, though slightly more conservative; see [Section 2](https://limewire.com/d/CyIaX#ve9BKM9Dmr).
>
> > **[Q2] Applicability to observational data**
>
> The method is applicable to real-world observational data under the usual identifying assumptions, The main challenges are standard observational causal inference ones: possible unmeasured confounding, poor overlap, and nuisance-estimation error. Sinkhorn divergence introduces an additional practical consideration: the entropic regularization parameter $\varepsilon$ creates a bias-stability tradeoff. As a special case, if propensity scores are known, our method would simplify by using the oracle propensity scores, essentially satisfying the doubly robust product condition in Thm. 4.1 of the main paper.
>
> > **[Q3] Validation when ground truth is unknown**
>
> In the NSW study, prior work (Firpo, 2007) claims that overlap is poor and unconfoundedness is questionable. We run our hypothesis tests on this dataset and as expected, overlap is poor (Figure 5). With careful nuisance training, our point estimates reproduce the qualitative pattern noted by Firpo—positive effects below the median and negative effects in the upper tail—but neither the MMD- nor Sinkhorn-based tests provide enough evidence to reject the null with confidence. From one representative subsample, the p-values of (MTE, STE) are (0.33,0 0.35). Across 100 such balanced subsamples, all p-values remain above 0.3. We view this empirical study as a validation exercise in a realistic observational setting where ground truth is unknown but the literature provides a well-understood stress test.

---

> > ### Author Rebuttal · Reviewer_x3BU · 2026-04-03
> >
> > I will keep my score.

---

### Official Review · Reviewer_yyki · 2026-03-11

**Soundness:** 3
**Presentation:** 2
**Significance:** 3
**Originality:** 3
**Overall Recommendation:** 4
**Confidence:** 3

**Summary:**

The authors introduce the Sinkhorn Treatment Effect (STE) as an optimal transport metric to measure the discrepancy between counterfactual distributions. They prove the first-order and second-order path differentiability of this statistical functional under general conditions. Furthermore, the authors construct debiased estimators and propose an asymptotically efficient hypothesis testing method for the Distributional Treatment Effect (DTE). Experiments on simulated data and high-dimensional image data demonstrate the advantages of the proposed method over Maximum Mean Discrepancy (MMD)-based approaches.

**Compliance With Llm Reviewing Policy:**

Affirmed.

**Final Justification:**

After considering the authors’ rebuttal, I decide to maintain my score.

**Key Questions For Authors:**

See above cons.

**Strengths And Weaknesses:**

Strengths And Weaknesses:

Pros:

1. The authors propose the Sinkhorn Treatment Effect as an optimal transport metric to gauge the discrepancy between counterfactual distributions, which is an interesting and highly meaningful research topic.

2. The authors derive the first-order and second-order efficient influence functions, further constructing debiased estimators and an asymptotically efficient hypothesis test based on U-statistics.

Cons:

1. The proposed method involves solving optimal transport problems and computing second-order U-statistics, which can be highly computationally expensive. Particularly when processing large datasets, the algorithm requires substantial computational resources (e.g., high-performance GPUs). Its feasibility in real-world applications remains questionable, and specific computational time (wall-clock time) and memory usage are not reported.

2. The entropy term introduced in the Sinkhorn regularization may cause the transport plan to become overly smooth, thereby blurring the true matching structure. How sensitive is the method to the entropy coefficient? Have the authors considered implementing measures to prevent the transport plan from degenerating into a nearly uniform coupling?

3. The authors primarily compare their method against MTE. Considering that MMD-based methods also involve various kernel choices and bandwidth selection strategies, the authors should explicitly clarify whether the baseline methods underwent sufficient hyperparameter tuning to ensure a fair comparison.

---

> ### Author Rebuttal · Authors · 2026-03-31
>
> We thank the reviewer for these helpful comments. We have revised the manuscript to address all three points; full details are provided in the [rebuttal document](https://limewire.com/d/CyIaX#ve9BKM9Dmr). In particular:
>
> > **Computation and scalability**.
>
> We now report wall-clock time and memory usage for STE and MTE across a range of sample sizes, using a CUDA-enabled machine with 12 GB GPU memory, 64 GB RAM, and 24 vCPUs. Although these timings are from a GPU implementation, we observe the same qualitative scaling trends on CPU; a detailed CPU/GPU comparison is included in [Section 5](https://limewire.com/d/CyIaX#ve9BKM9Dmr).
>
> | Method | Metric                | n=250 | n=500 | n=1000 | n=2000 | n=4000 |
> | ------ | --------------------- | ----: | ----: | -----: | -----: | -----: |
> | MTE    | Wall-clock time (sec) |   1.4 |   2.6 |    6.1 |   16.7 |   58.1 |
> | MTE    | Memory usage (MB)     |  1442 |  1494 |   1428 |   1426 |   1909 |
> | STE    | Wall-clock time (sec) |   1.5 |   2.8 |    6.6 |   19.5 |   74.4 |
> | STE    | Memory usage (MB)     |  1443 |  1500 |   1439 |   1316 |   1983 |
>
> For the sample sizes studied in the paper, the method is computationally feasible, but we agree that the current implementation is not yet optimized for very large $n$. Both per-iteration Sinkhorn step and the second-order $U$-statistic computation scale quadratically in the naive implementation. We make this limitation explicit in [Section 5](https://limewire.com/d/CyIaX#ve9BKM9Dmr) and discuss standard accelerations that can be incorporated directly into our framework.
>
> 1. **Per-iteration Sinkhorn cost**: To mitigate this, we can use (1) kernel-compression methods such as Nyström-Sinkhorn, that run Sinkhorn algorithm on the Nÿstrom approximation of the kernel matrix $K_{ij} = e^{-C_{ij} / \varepsilon}$, where $C$ is the cost matrix, or (2) stochastic OT methods that estimate entropic potentials via minibatch stochastic gradient descent. Both of these methods get rid of the $O(n^2)$ bottleneck and can be directly plugged in the one-step STE estimator.
>
> 2. **$U$-statistic computation cost**: To mitigate this, we can use linear-time approximations ([Gretton et al., 2012](https://www.jmlr.org/papers/volume13/gretton12a/gretton12a.pdf)) or incomplete U-statistics ([Schrab et al., 2022](https://neurips.cc/media/neurips-2022/Slides/54933.pdf)). Both can be plugged directly into the second-order bias-correction term.
>
> In addition to above considerations, we expand the discussion to acceleration of the number of Sinkhorn iterations and Algorithm 2 (from the main paper) in [Section 5](https://limewire.com/d/CyIaX#ve9BKM9Dmr). While we expect similar empirical gains as observed in these papers, the asymptotic analysis of its incorporation in our test is outside the scope of our paper.
>
> > **Sensitivity to the entropic regularization**
>
> The entropic regularization parameter $\varepsilon$ induces a bias-stability tradeoff: large $\varepsilon$ oversmooths the transport plan, while very small $\varepsilon$ can lead to numerical instability in the Sinkhorn iterations, especially when $\varepsilon$ is small relative to the scale of the cost matrix. In the paper, we use a data-dependent heuristic based on the median pairwise cost.
>
> To reduce sensitivity to any single choice of $\varepsilon$, we propose **STEAgg**, which aggregates evidence over a finite grid of numerically stable $\varepsilon$ values, while retaining asymptotic Type-I control; see [Section 2](https://limewire.com/d/CyIaX#ve9BKM9Dmr). Concretely, letting $m_0$ denote the median pairwise cost, we consider
> $\varepsilon_i = \eta_i m_0, i = 1, \dots, m,$
> where $\eta_i$ are fixed multipliers. If $T_{n, \varepsilon_i}$ denotes the test statistic computed at $\varepsilon_i$, then STEAgg is
> $\max_{i \in [m]} T_{n, \varepsilon_i}$.
> This method is inspired by aggregated kernel tests ([Schrab et al. (2023)](https://www.jmlr.org/papers/volume24/21-1289/21-1289.pdf)) that combine MMD statistics across bandwidths while controlling Type-I error. Empirically, single-scale STE can lose power when $\varepsilon$ is chosen too small or too large, whereas STEAgg is more robust, albeit slightly more conservative. Our simulations are consistent with this bias-stability tradeoff, with the strongest performance attained near the median-based default, that is $\eta = 1.0$.
>
> > **Sensitivity to the kernel bandwidth**
>
> In our experiments, MTE uses a Gaussian kernel with bandwidth selected by the median heuristic, while STE uses an analogous median-based scale for the transport cost. Thus, neither method relies on aggressive tuning. To further reduce dependence on a single bandwidth choice, we now also introduce an aggregated version of MTE, **MTEAgg**, parallel to STEAgg. Like STEAgg, MTEAgg avoids reliance on a single heuristic bandwidth at the cost of evaluating the statistic over a small grid.

---

> > ### Author Rebuttal · Reviewer_yyki · 2026-04-02
> >
> > Thank you to the authors for their detailed responses to my questions.
> >
> > However, could the authors further provide sensitivity analyses for the entropic regularization parameter to validate the proposed STEAgg?
> > Do STEAgg and MTEAgg come with complete theoretical guarantees?
> > The authors evaluate the method on real data; however, for real image data such as PCam, increasing dimensionality can lead to more extreme propensity scores [1]. Would this affect the first- and second-order EIFs? This issue is particularly important for the stability of the proposed method. Yet the paper only uses the first 10 principal components, which seems insufficient to validate the method’s effectiveness on real-world data.
> > [1]Melnychuk, Valentyn, et al. "Overlap-Adaptive Regularization for Conditional Average Treatment Effect Estimation." arXiv preprint arXiv:2509.24962 (2025).

---

> > > ### Author Response · Authors · 2026-04-07
> > >
> > > We thank the reviewer for these questions. They helped us strengthen both the empirical validation and the theoretical presentation of the aggregated procedures, and clarify the role of positivity in our framework.
> > >
> > > Please refer to the [rebuttal document](https://limewire.com/d/tT6qm#f2WgtpwYfZ) for all results.
> > >
> > > > **Sensitivity analysis for regularization parameter**
> > >
> > > We now include an explicit sensitivity analysis for the regularization/bandwidth parameter for both STE and MTE, together with the aggregated versions STEAgg and MTEAgg (Fig. 1). We evaluate the fixed-$\varepsilon$ tests over the grid
> > >
> > > $$ \varepsilon_i = \eta_i m, \quad \eta_i \in \lbrace 2^{-4}, 2^{-3}, 2^{-2}, 2^{-1}, 2^0, 2^1, 2^2, 2^3, 2^4 \rbrace,$$
> > > where $m$ is the median pairwise distance. The simulation demonstrates tuning sensitivity for both families of fixed-$\varepsilon$ tests. Under the null, type-I error varies across the grid, with slight inflation at small values of $\varepsilon$ and conservative type-I error at large values. Under the alternative, power is nonuniform, with very small and very large values of $\varepsilon$ leading to power loss. In contrast, the aggregation strategies STEAgg and MTEAgg appear robust to the inclusion of overly large or small values of $\varepsilon$ in the grid, having power that is almost as high as that of the best-performing values of $\varepsilon$ while still controlling type-I error. This is precisely the motivation for aggregation: it protects against tuning misspecification without requiring a single heuristic choice of $\varepsilon$.
> > >
> > > These results also recover the expected bias-variance trade-off in $\varepsilon$. For STE, very small values are numerically unstable, whereas very large values oversmooth the discrepancy and reduce power. In our simulations, among the fixed choices, performance is typically strongest near the median heuristic, making it a reasonable default for non-aggregated testing.
> > >
> > > > **Theoretical guarantees for STEAgg and MTEAgg**
> > >
> > > Yes. In the rebuttal document, we add a theorem (Thm. 1) establishing the asymptotic null distribution of the aggregated statistic. With this result, our proposed test achieves nominal type-I error asymptotically (see Eq. (4)). We prove the theorem in the document appendix.
> > >
> > > > **Low Overlap and high-dimensionality issues**
> > >
> > > Indeed, low overlap inflates inverse-propensity weights and can destabilize both the first- and second-order EIF terms. Our theory, like most IPW-based methods, is developed under a strong positivity assumption. In practice, however, positivity violations are common: they can arise even in low-dimensional settings when certain covariates almost deterministically receive one treatment (for example, due to clinical practice), and they may become more pronounced as the covariate dimension increases.
> > >
> > > This issue is well studied in causal inference. Common remedies include truncating estimated propensity scores away from 0 and 1 [[1]](https://pubmed.ncbi.nlm.nih.gov/21030422/), trimming units with extreme propensity scores [[2]](https://academic.oup.com/biomet/article/96/1/187/235329?guestAccessKey=), restricting inference to better-overlap subpopulations, and using overlap or other stabilized weights [[3]](https://projecteuclid.org/journals/statistical-science/volume-35/issue-3/Comment--Stabilizing-the-Doubly-Robust-Estimators-of-the-Average/10.1214/20-STS774.full). In the revision, we now state explicitly that estimated propensity scores are truncated in our experiments, and we add the following discussion:
> > >
> > > *“Stabilizing the one-step estimator for STE under practical positivity violations is an important direction for future work. In particular, adapting trimming, overlap weighting, or other stabilized weighting strategies developed for ATE estimation [[4]](https://www.tandfonline.com/doi/full/10.1080/01621459.2016.1260466) to the STE setting is a natural next step.”*
> > >
> > > Regarding the reviewer’s point on PCam, we clarify that our current dimension reduction is applied to the outcome $Y$, not to the covariates $X$ used in the propensity model. Thus, since the outcome is not involved in propensity estimation, the use of 10 principal components should not be interpreted as addressing high-dimensional propensity estimation. More broadly, however, handling high-dimensional covariates and outcomes remains an important direction for future work. On the covariate side, a promising direction is to use dimension reduction methods that construct lower-dimensional balancing representations while preserving the identifiability of causal effects [[5]](https://pmc.ncbi.nlm.nih.gov/articles/PMC6588012/). On the outcome side, we add the following to the discussion:
> > >
> > > *“For high-dimensional outcomes, the discrepancy between $P_0$ and $P_1$ may concentrate on lower-dimensional subspaces. An interesting direction for future work is to explore sliced variants of optimal transport metrics that aggregate one-dimensional discrepancies.”*

---

### Official Review · Reviewer_WXjt · 2026-03-11

**Soundness:** 4
**Presentation:** 3
**Significance:** 4
**Originality:** 4
**Overall Recommendation:** 5
**Confidence:** 4

**Summary:**

In a causal model, a divergence measure between counterfactual distributions is introduced and studied. The divergence, termed Sinkhorn Treatment Effects (STE), is based on the Sinkhorn divergence between kernel mean embeddings of the counterfactuals. An estimator of the STE is constructed by a second order one-step estimator using an initial estimator constructed from an independent sample (sample splitting). This estimator is then used to construct a statistical test for equality of the two counterfactual distributions. The limit distribution of the test statistic under the null-hypothesis is derived. The main technical ingredient here is to establish second order pathwise differentiability of the STE. The manuscript also discusses computational issues and conducts numerical experiments to illustrate the performance of the test.

**Compliance With Llm Reviewing Policy:**

Affirmed.

**Key Questions For Authors:**

- I was a little bit surprised about the fact that the proposed test shows quite different behavior under the null in the two scenarios considered. Do you have any insights on this?
- I was also wondering in how much the estimation of the nuisance parameters influences the type 1 error. It would not be surprising if there were some bias issues. Have you thought about this? (And if yes, can this explain the different behavior of the type I error in the two scenarios?)
- In your second example, the two counterfactuals differ in both mean and variance.  Perhaps it would be insightful to decouple these differences (i.e. consider only a change in mean, and/or only a change in variance)?

**Limitations:**

yes.

**Strengths And Weaknesses:**

STRENGTHS: The manuscript is addressing an interesting and timely problem. It is nicely written demonstrating both methodological and theoretical competency. The main technical challenge is the derivation of the second order differentiability of the STE, which then leads to the form of the asymptotic distribution of the proposed test statistic that also involves estimates of nuisance parameters. On a technical level this is a non-trivial and interesting application of classical statistical theory for semi-parametric efficiency.

The provided discussions nicely connect the contributions to the related literature, and existing results that are being used in the technical derivations appear appropriately cited.

The presented numerical experiments illustrate feasibility of conducting the proposed test statistics and provides some insights about its performance.

WEAKNESS:

I did not spot significant weaknesses, but I do have a few questions about the numerical studies/finite sample behavior of the proposed test (see below).

I also did spot a few typos:

In the main text:
- In (12), shouldn’t the ‘$\le \alpha$’ be an ‘$= \alpha$’?
- page 3, equation (2): I guess the index “$Y$” should be lower case.
- page 4, first formula in (6): $P(dy’|A=a,X=x)$ (the ‘prime’ is missing)

In the Background section B:
- equation (13): It seems that it should be ‘$D\Phi_S(s)$’ The index ’$S$’ seems missing)

References:
The references seem to need some more attention.
- Please make sure that names, such as Riemann, Wasserstein or Sinkhorn, are capitalized.
- Double-check capitalization, such as in
 	- “Mathematical Sciences” in Kantorovitvh (2006)
	- ‘Statistical’ (capitalized) in Robins et al. (1994) and in Rubin (2005)
	- ‘Educational’ in Rubin (1974)
	- ‘Volume’ in Villani (2009)
	- check inconsistencies in capitalizing the outlets for instance in Bareinboim et al. (2022), Berliner  and Thomas-Agnan (2011), Bickel et al. (1993)… and many others
- Details are missing for Ramdas et a. (2015)
- Correct `Schrödinger’ in Levenant et al. (2024)

---

> ### Author Rebuttal · Authors · 2026-03-31
>
> We thank the reviewer for the careful reading and for identifying these typographical and formatting issues. We have revised the manuscript to correct typographical issues and improve clarity throughout. These changes will be reflected in the revised version of the manuscript. We now address your questions directly.
>
> > **[Q1 + Q2] I was a little bit surprised about the fact that the proposed test shows quite different behavior under the null in the two scenarios considered... I was also wondering in how much the estimation of the nuisance parameters influences the type 1 error.**
>
> Under our theoretical assumptions, the proposed test achieves nominal Type-1 error asymptotically, so in large samples we do not expect systematic differences. The differences observed in finite samples in our experiments, albeit similar trends, are primarily due to nuisance estimation (as suggested in your [Q2]). Indeed, the Sinkhorn-based procedure involves additional nuisance estimation (of entropic dual potentials), on top of the shared nuisance components: propensity score $(A | X)$ and outcome regression $(Y | A, X)$ model. This can introduce slightly higher variability in finite samples, which manifests as differences in empirical Type-I error.
>
> > **[Q2] It would not be surprising if there were some bias issues. Have you thought about this?**
>
> Our one-step construction is designed so that, under the stated regularity conditions, the estimator is asymptotically unbiased. In that sense, the procedure is bias-corrected at first order, and under the null the remainder terms are of higher order. However, in finite samples, nuisance estimation affects Type-1 error through two primary channels:
>
> 1) residual error in the estimated nuisance components themselves; and
> 2) error in estimating the asymptotic variance used for calibration.
>
> Our view is therefore that nuisance estimation is an important contributor to the finite-sample differences in Type-1 error.
>
> > **[Q3] Perhaps it would be insightful to decouple these differences (i.e. consider only a change in mean, and/or only a change in variance)?**
>
> We appreciate this suggestion and agree that it makes the empirical comparison substantially more informative. We have extended the simulation study to decouple the differences between mean and covariances (see [Section 1](https://limewire.com/d/CyIaX#ve9BKM9Dmr)). Specifically, we consider two alternative regimes:
>
> 1. **Exp I (mean shift)**: $P_1$ and  $P_0$ have different means but identical covariance, and
> 2. **Exp II (covariance shift)**: $P_1$ and  $P_0$ have identical means but differ via symmetric rank-2 perturbation of the covariance matrix, supported on the span of its two leading eigenvectors.
>
> These regimes are visualized in [Figure 1](https://limewire.com/d/CyIaX#ve9BKM9Dmr) and explain the behavior seen in the original experiments. In Exp I, both the MMD-based and Sinkhorn-based tests gain power quickly, which is consistent with the fact that both methods are sensitive to mean differences. In Exp II, however, the Sinkhorn-based test is consistently more powerful in our experiments, maintaining the advantage even as the distributional gap increases. This is because Exp II perturbations induce structured mass transport while preserving many low-order kernel features, making the advantage of Sinkhorn-based test over the MMD-based test clear in this setting.
>
> Finally, we agree with the reviewer that in real data applications (like the second example), the treatment and control distributions may differ in ways that are not cleanly decomposable into mean and covariance differences alone. This is precisely part of the motivation for using a fully distributional estimand and test, whose advantage is established via the updated simulation study now.

---

> > ### Author Rebuttal · Reviewer_WXjt · 2026-04-03
> >
> > Thanks for the nice and insightful work. I maintain my positive rating.

---

### Official Review · Reviewer_vUbg · 2026-03-13

**Soundness:** 3
**Presentation:** 3
**Significance:** 2
**Originality:** 3
**Overall Recommendation:** 4
**Confidence:** 3

**Summary:**

The paper introduces the Sinkhorn treatment effect (STE) as the Sinkhorn divergence between counterfactual outcome distributions, shows first-order pathwise differentiability in general and second-order pathwise differentiability under the null, and then develops debiased estimators and tests based on these smoothness properties.

**Compliance With Llm Reviewing Policy:**

Affirmed.

**Final Justification:**

I thank the authors for the detailed rebuttal and their efforts in revising the manuscript. I remain positive about the submission.

**Key Questions For Authors:**

See strengths and concerns.

**Limitations:**

See strengths and concerns.

**Strengths And Weaknesses:**

Strengths.

Overall, the research's central aspect is the attempt to study treatment effects at the level of full distributions rather than only through means. This is practically relevant, and the paper clearly explains that mean-based summaries can miss important changes away from the center of the distribution.

Another strength is the second-order theory. The paper addresses the degeneracy of first-order inference under the null, derives a second-order efficient influence function, and establishes the null limiting distribution of the resulting test statistic. This part is less common in semi-paramteric theory papers.

Concerns.

1. The estimand could be better justified. Using the STE as a tool for detecting whether the two distributions differ is reasonable. However, the STE itself seems less directly interpretable as a causal estimand than more standard quantities such as the ATE. In particular, while it serves as a measure of distributional discrepancy, its scientific meaning may be less clear in applications.

2. My second concern is the empirical evaluation. In Figure 2, Panel I shows that the type I error rates of the MMD- and STE-based methods are comparable. In Panel II, however, the MMD-based method appears to be slightly more powerful. This makes it less clear whether the proposed method offers a substantial empirical advantage in this setting.

---

> ### Author Rebuttal · Authors · 2026-03-31
>
> We thank the reviewer for their insightful comments. The comment on empirical experiments is also reflected by Reviewer x3BU, and has helped us to strengthen the scope of our empirical investigations. More specifically,
>
> > **Scientific meaning of Sinkhorn treatment effect**
>
> We acknowledge the gap in scientific grounding of our causal estimand – the Sinkhorn treatment effect (STE). We have revised the introduction to better position STE as a distribution-level causal estimand that complements, standard summaries such as the ATE. Concretely, we now clarify that:
>
> "*Formally, STE quantifies the minimum cost of reallocating probability mass from $P_0$ to $P_1$, where the cost of moving mass between two outcome values is defined by a chosen ground metric reflecting domain-specific notions of discrepancy between outcomes.*"
>
> The ATE answers the question: *how much does treatment change the average outcome?* In contrast, STE answers the question: *how different are the treated and untreated outcome distributions, when differences are measured through a chosen notion of distance on the outcome space?*
>
> From an applied perspective, we also emphasize that the choice of cost should reflect domain knowledge about what constitutes a meaningful discrepancy between outcomes. For example, in the new LaLonde NSW study ([Section 5](https://limewire.com/d/CyIaX#ve9BKM9Dmr)) on earnings data, a quadratic cost is natural because larger income differences are more consequential and should therefore incur larger transport penalties. More generally, outside causal inference, OT applications routinely tailor the cost to the geometry of the data. For images, costs are chosen to respect spatial bin geometry of pixels ([Scetborn & Cuturi, 2020](https://proceedings.neurips.cc/paper/2020/hash/9bde76f262285bb1eaeb7b40c758b53e-Abstract.html)), while in single-cell biology OT is used with costs that induce meaningful alignment of cell populations ([Samaran et al., 2024](https://www.nature.com/articles/s41467-024-51382-x)).
>
> > **Extended simulation study to showcase advantage of STE over MTE**
>
> We thank the reviewer for a careful reading of Figure 2. Our interpretation is that, in Figure 2, the comparable performance of MTE and STE is largely because the original alternative mixes mean and covariance changes in a way that remains well aligned with kernel-based features. In such a setting, it is not surprising that MMD performs competitively. Under our regularity conditions, both MTE and STE admit local second-order expansions near the null, so when the alternative is dominated by features that are already well captured by the kernel, similar local sensitivity is expected.
>
> Combined with suggestion from Reviewer WXjt, we have extended the simulation study to decouple the differences between mean and covariances for constructing alternatives. We present results for two regimes:
>
> 1. **Exp I (mean shift)**: $P_0$ and $P_1$ have identical covariance but different means, and
> 2. **Exp II (covariance shift)**: $P_0$ and $P_1$ have identical means but differ via symmetric rank-2 perturbation of the covariance matrix, supported on the span of its two leading eigenvectors.
>
> This revised design ([Section 1](https://limewire.com/d/CyIaX#ve9BKM9Dmr)) makes the comparison more informative. In Exp I, both methods show increasing power, which is consistent with the fact that both kernel-based and transport-based discrepancies are sensitive to mean differences. The perturbations in Exp II induce structured mass transport while preserving many low-order kernel features, making the advantage of STE over MTE clear in this setting. In Exp II, STE-based test is consistently more powerful than the MTE-based test in our experiments, maintaining the advantage even as the distributional gap increases.

---

> > ### Author Rebuttal · Reviewer_vUbg · 2026-04-03
> >
> > Thank you for the reply!
> >
> > “STE quantifies the minimum cost of reallocating probability mass from $P_0$ to $P_1$” may not be perfectly precise. According to Eq.(5), the Sinkhorn divergence depends on the regularization parameter $\epsilon$ of the entropic penalty. The interpretation as the minimum transport cost corresponds to the unregularized case $\epsilon=0$, namely OT. When $\epsilon=0$, both the statistical and computational aspects become substantially more challenging, and the results in the manuscript do not directly apply.

---

> > > ### Author Response · Authors · 2026-04-07
> > >
> > > Thank you for this comment - we agree that our previous wording was imprecise. The phrase “minimum cost of reallocating probability mass from $P_0$ to $P_1$” corresponds most directly to the unregularized OT problem ($\varepsilon = 0$). In contrast, the estimand in Eq. (5), and the one we consider throughout our paper, is the Sinkhorn divergence, which is the centered version of entropic OT (EOT), and depends on the regularization parameter $\varepsilon > 0$. Accordingly, STE should be described as a *regularized minimum transport cost* between $P_0$ and $P_1$.
> > >
> > > STE is a transport-based distributional discrepancy between $P_0$ and $P_1$, induced by a ground cost $c$, together with the regularization parameter $\varepsilon$. Here, the ground cost $c$ encodes which discrepancies between outcome values are scientifically meaningful, while $\varepsilon$ controls the degree of entropic smoothing in the comparison of counterfactual outcome distributions. As $\varepsilon \to 0$, the Sinkhorn divergence approaches the unregularized OT cost, which admits the classical minimum-cost transport interpretation. However, our paper does not study this unregularized limit. As stated in **Section 3** of our main paper, all of our differentiability results and limit theorems are developed for a fixed $\varepsilon > 0$ case. This regularized setting offers important statistical and computational advantages over classical OT: EOT can be computed efficiently for empirical distributions using the celebrated Sinkhorn algorithm [1]. The Sinkhorn divergence used in our paper is the debiased, centered version of this EOT cost. While STE is scientifically meaningful as a causal estimand in its own right, it also provides a natural basis for testing the causal null. Our paper explores both of these characteristics.
> > >
> > > We have revised the manuscript to make the scientific interpretation of STE as a causal estimand clearer. We have removed the earlier sentence suggesting that STE itself is the minimum transport cost, and replaced it with language that is accurate for the regularized setting considered in the paper.
> > >
> > > *“Formally, STE quantifies how different $P_0$ and $P_1$ are through a geometric transport-based discrepancy, defined by a ground cost $c$ and a regularization level $\varepsilon$. Here $c$ reflects the cost of moving mass between two outcome values, thereby encoding domain-specific notions of discrepancy between outcomes, while $\varepsilon$ controls the degree of entropic smoothing. As $\varepsilon \to 0$, STE approaches the corresponding unregularized OT cost, and as $\varepsilon \to \infty$, it approaches the MMD distance corresponding to the Gibbs kernel $e^{-c / \varepsilon}$. More broadly, Sinkhorn divergence interpolates between the OT cost and MMD [2], balancing the appealing geometric properties of OT and computational feasibility of MMD.”*
> > >
> > > ---
> > >
> > > [1] Cuturi, Marco. Sinkhorn distances: Lightspeed computation of optimal transport. Advances in neural information processing systems 26 (2013).
> > >
> > > [2] Feydy, Jean, et al. "Interpolating between optimal transport and MMD using Sinkhorn divergences." The 22nd international conference on artificial intelligence and statistics. PMLR, 2019.

---

### Decision · Program_Chairs · 2026-04-30

**Decision:**

Accept (regular)

**Comment:**

The submission introduces Sinkhorn treatment effects (STE), a measure of the divergence between the distribution of the potential outcomes of interventions (counterfactuals). Unlike traditional effects, defined as the expected difference between outcomes, this approach captures differences across entire distributions.

On the whole, the paper was supported by all reviewers, who mentioned, among others, the practical relevance and timeliness of the problem, the new connection between entropic OT and semiparametric influence functions, and the well-structured manuscript as strengths.

No severe weaknesses or errors were pointed out by reviewers, but a few concerns were raised and discussed with the authors, for example, the low interpretability of the STE, compared to simple expected contrast functions, and the limited empirical evidence for the power of the proposed effect measure. Regarding the latter, one reviewer pointed out that, in the original submission, the proposed method did not show stronger power than an effect measure based on the MMD, pitched as a worse alternative early in the paper. In the rebuttal, the authors did provide new experiments to complete this picture. Mostly, reviewers were content with the rebuttals but left a few remaining questions for the authors which were addressed in follow-up comments. Following extensive rebuttals and discussion, all reviewers left their recommendations on the side of accepting the paper.